# Information Rates for Channels with Fading, Side Information and Adaptive Codewords

**DOI:** 10.3390/e25050728

**Published:** 2023-04-27

**Authors:** Gerhard Kramer

**Affiliations:** School of Computation, Information and Technology, Technical University of Munich (TUM), 80333 Munich, Germany; gerhard.kramer@tum.de

**Keywords:** capacity, channel state information, directed information, fading, feedback, generalized mutual information, side information

## Abstract

Generalized mutual information (GMI) is used to compute achievable rates for fading channels with various types of channel state information at the transmitter (CSIT) and receiver (CSIR). The GMI is based on variations of auxiliary channel models with additive white Gaussian noise (AWGN) and circularly-symmetric complex Gaussian inputs. One variation uses reverse channel models with minimum mean square error (MMSE) estimates that give the largest rates but are challenging to optimize. A second variation uses forward channel models with linear MMSE estimates that are easier to optimize. Both model classes are applied to channels where the receiver is unaware of the CSIT and for which adaptive codewords achieve capacity. The forward model inputs are chosen as linear functions of the adaptive codeword’s entries to simplify the analysis. For scalar channels, the maximum GMI is then achieved by a conventional codebook, where the amplitude and phase of each channel symbol are modified based on the CSIT. The GMI increases by partitioning the channel output alphabet and using a different auxiliary model for each partition subset. The partitioning also helps to determine the capacity scaling at high and low signal-to-noise ratios. A class of power control policies is described for partial CSIR, including a MMSE policy for full CSIT. Several examples of fading channels with AWGN illustrate the theory, focusing on on-off fading and Rayleigh fading. The capacity results generalize to block fading channels with in-block feedback, including capacity expressions in terms of mutual and directed information.

## 1. Introduction

The capacity of fading channels is a topic of interest in wireless communications [1,2,3,4]. Fading refers to model variations over time, frequency, and space. A common approach to track fading is to insert pilot symbols into transmit symbol strings, have receivers estimate fading parameters via the pilot symbols, and have the receivers share their estimated channel state information (CSI) with the transmitters. The CSI available at the receiver (CSIR) and transmitter (CSIT) may be different and imperfect.

Information-theoretic studies on fading channels distinguish between average (ergodic) and outage capacity, causal and non-causal CSI, symbol and rate-limited CSI, and different qualities of CSIR and CSIT that are coarsely categorized as no, perfect, or partial. We refer to [5] for a review of the literature up to 2008. We here focus exclusively on average capacity and causal CSIT as introduced in [6]. Codes for such CSIT, or more generally for noisy feedback [7], are based on Shannon strategies, also called codetrees ([8], Chapter 9.4), or adaptive codewords ([9], Section 4.1). (The term “adaptive codeword” was suggested to the author by J. L. Massey.) Adaptive codewords are usually implemented by a conventional codebook and by modifying the codeword symbols as a function of the CSIT. This approach is optimal for some channels [10] and will be our main interest.

### 1.1. Block Fading

A model that accounts for the different time scales of data transmission (e.g., nanoseconds) and channel variations (e.g., milliseconds) is block fading [11,12]. Such fading has the channel parameters constant within blocks of *L* symbols and varying across blocks. A basic setup is as follows.
The fading is described by a state process $S_{H1},S_{H2},\ldots$ independent of the transmitter messages and channel noise. The subscript “*H*” emphasizes that the states $S_{Hi}$ may be hidden from the transceivers.Each receiver sees a state process $S_{R1},S_{R2},\ldots$ where $S_{Ri}$ is a noisy function of $S_{Hi}$ for all *i*.Each transmitter sees a state process $S_{T1},S_{T2},\ldots$ where $S_{Ti}$ is a noisy function of $S_{Hi}$ for all *i*.
The state processes may be modeled as memoryless [11,12] or governed by a Markov chain [13,14,15,16,17,18,19,20,21]. The memoryless models are particular cases of Shannon’s model [6]. For scalar channels, $S_{Hi}$ is usually a complex number $H_{i}$. Similarly, for vector or multi-input, multi-output (MIMO) channels with *M*- and *N*-dimensional inputs and outputs, respectively, $S_{Hi}$ is a N×M matrix $H_{i}$.

Consider, for example, a point-to-point channel with block-fading and complex-alphabet inputs $X_{i\ell }$ and outputs
(1)$$\begin{array}{c} Y_{i\ell }=H_{i}X_{i\ell }+Z_{i\ell } \end{array}$$
where the index *i*, i=1,…,n, enumerates the blocks and the index *ℓ*, ℓ=1,…,L, enumerates the symbols of each block. The additive white Gaussian noise (AWGN) $Z_{11},Z_{12},\ldots$ is a sequence of independent and identically distributed (i.i.d.) random variables that have a common circularly-symmetric complex Gaussian (CSCG) distribution.

### 1.2. CSI and In-Block Feedback

The motivation for modeling CSI as independent of the messages is simplicity. If one uses only pilot symbols to estimate the $H_{i}$ in (Equation 1), for example, then the independence is valid, and the capacity analysis may be tractable. However, to improve performance, one can implement data and parameter estimation jointly, and one can actively adjust the transmit symbols $X_{i\ell }$ using past received symbols $Y_{ik}$, k=1,…,ℓ−1, if in-block feedback is available. (Across-block feedback does not increase capacity if the state processes are memoryless; see ([22], Remark 16).) An information theory for such feedback was developed in [22], where a challenge is that code design is based on adaptive codewords that are more sophisticated than conventional codewords.

For example, suppose the CSIR is $S_{Ri}=H_{i}$. Then, one might expect that CSCG signaling is optimal, and the capacity is an average of log(1+SNR) terms, where SNR is a signal-to-noise ratio. However, this simplification is based on constraints, e.g., that the CSIT is a function of the CSIR and that the $X_{i\ell }$ cannot influence the CSIT. The former constraint can be realistic, e.g., if the receiver quantizes a pilot-based estimate of $H_{i}$ and sends the quantization bits to the transmitter via a low-latency and reliable feedback link. On the other hand, the latter constraint is unrealistic in general.

### 1.3. Auxiliary Models

This paper’s primary motivation is to further develop information theory for adaptive codewords. To gain insight, it is helpful to have achievable rates with log(1+SNR) terms. A common approach to obtain such expressions is to lower bound the channel mutual information I(X;Y) as follows.

Suppose *X* is continuous and consider two conditional densities: the density p(x|y) and an auxiliary density q(x|y). We will refer to such densities as reverse models; similarly, p(y|x) and q(y|x) are called forward models. One may write the differential entropy of *X* given *Y* as
(2)$$\begin{array}{cc} h(X|Y) & =E\left[-\log p(X|Y)\right]=\underset{\mathrm{average}\mathrm{cross}-\mathrm{entropy}}{\underbrace{E\left[-\log q(X|Y)\right]}}-\underset{\mathrm{average}\mathrm{divergence}\geq 0}{\underbrace{E\left[\log\frac{p(X|Y)}{q(X|Y)}\right]}} \end{array}$$
where the first expectation in (Equation 2) is an average cross-entropy, and the second is an average informational divergence, which is non-negative. Several criteria affect the choice of q(x|y): the cross-entropy should be simple enough to admit theoretical or numerical analysis, e.g., by Monte Carlo simulation; the cross-entropy should be close to h(X|Y); and the cross-entropy should suggest suitable transmitter and receiver structures.

We illustrate how reverse and forward auxiliary models have been applied to bound mutual information. Assume that $E\left[X\right]=E\left[Y\right]=0$ for simplicity.

*Reverse Model:* Consider the reverse density that models X,Y as jointly CSCG:(3)$$\begin{array}{c} q\left(x|y\right)=\frac{1}{\pi \sigma_{L}^{2}}\exp\left(-{\left|x-{\hat{x}}_{L}\right|}^{2}/\sigma_{L}^{2}\right) \end{array}$$
where ${\hat{X}}_{L}=\left(E\left[X\;Y^{*}\right]/E\left[{\left|Y\right|}^{2}\right]\right)Y$ and
(4)$$\begin{array}{c} \sigma_{L}^{2}=E\left[{\left|X-{\hat{X}}_{L}\right|}^{2}\right]=E\left[{\left|X\right|}^{2}\right]-\frac{|E\left[XY^{*}\right]{|}^{2}}{E\left[{\left|Y\right|}^{2}\right]} \end{array}$$
is the mean square error (MSE) of the estimate ${\hat{X}}_{L}$. In fact, ${\hat{X}}_{L}$ is the linear estimate with the minimum MSE (MMSE), and $\sigma_{L}^{2}$ is the linear MMSE (LMMSE) which is independent of Y=y; see Section 2.5. The bound in (Equation 2) gives
(5)$$\begin{array}{c} h\left(X|Y\right)\leq \log\left(\pi e\;\sigma_{L}^{2}\right). \end{array}$$
Thus, if *X* is CSCG, then we have the desired form
(6)$$\begin{array}{c} I\left(X;Y\right)=h\left(X\right)-h\left(X|Y\right)\geq \log\left(1+\frac{{\left|h\right|}^{2}E\left[{\left|X\right|}^{2}\right]}{\sigma^{2}}\right) \end{array}$$
where the parameters *h* and $\sigma^{2}$ are
(7)$$\begin{array}{c} h=\frac{E\left[YX^{*}\right]}{E\left[{\left|X\right|}^{2}\right]},\;\sigma^{2}=E\left[{\left|Y-hX\right|}^{2}\right]. \end{array}$$
The bound (Equation 6) is apparently due to Pinsker [23,24,25] and is widely used in the literature; see e.g., [18,26,27,28,29,30,31,32,33,34,35,36,37,38]. The bound is usually related to channels p(y|x) with additive noise but (Equation 2)–(Equation 6) show that it applies generally. The extension to vector channels is given in Section 2.7 below.

*Forward Model:* A more flexible approach is to choose the reverse density as
(8)$$\begin{array}{c} q\left(x|y\right)=\frac{p\left(x\right)q{\left(y|x\right)}^{s}}{q(y)} \end{array}$$
where q(y|x) is a forward auxiliary model (not necessarily a density), s≥0 is a parameter to be optimized, and
(9)$$\begin{array}{c} q\left(y\right)=\int_{C}p\left(x\right)\;q{\left(y|x\right)}^{s}\;dx. \end{array}$$
Inserting (Equation 8) into (Equation 2) we compute
(10)$$\begin{array}{c} I\left(X;Y\right)\geq \max_{s\geq 0}E\left[\log\frac{q{\left(Y|X\right)}^{s}}{q(Y)}\right]. \end{array}$$
The right-hand side (RHS) of (Equation 10) is called a generalized mutual information (GMI) [39,40] and has been applied to problems in information theory [41], wireless communications [42,43,44,45,46,47,48,49,50,51], and fiber-optic communications [52,53,54,55,56,57,58,59,60,61]. For example, the bounds (Equation 6) and (Equation 10) are the same if s=1 and
(11)$$\begin{array}{c} q\left(y|x\right)=\exp\left({-|y-hx|}^{2}/\sigma^{2}\right) \end{array}$$
where *h* and $\sigma^{2}$ are given by (Equation 7). Note that (Equation 11) is not a density unless $\sigma^{2}=1/\pi$ but q(x|y) is a density. (We require q(x|y) to be a density to apply the divergence bound in (Equation 2).)

We compare the two approaches. The bound (Equation 5) is simple to apply and works well since the choices (Equation 7) give the maximal GMI for CSCG *X*; see Proposition 1 below. However, there are limitations: one must use continuous *X*, the auxiliary model q(y|x) is fixed as (Equation 11), and the bound does not show how to design the receiver. Instead, the GMI applies to continuous/discrete/mixed *X* and has an operational interpretation: the receiver uses q(y|x) rather than p(y|x) to decode. The framework of such mismatched receivers appeared in ([62], Exercise 5.22); see also [63].

### 1.4. Refined Auxiliary Models

The two approaches above can be refined in several ways, and we review selected variations in the literature.

*Reverse Models:* The model q(x|y) can be different for each Y=y, e.g., on may choose *X* as Gaussian with mean $E\left[X|Y=y\right]$ and variance
(12)$$\begin{array}{c} \mathrm{Var}\left[X|Y=y\right]=E\left[{\left|X\right|}^{2}|Y=y\right]-|E\left[X|Y=y\right]{|}^{2} \end{array}$$
and where
(13)$$\begin{array}{c} q\left(x|y\right)=\frac{1}{\pi \mathrm{Var}\left[X|Y=y\right]}\exp\left(-\frac{{\left|x-E\left[X|Y=y\right]\right|}^{2}}{\mathrm{Var}\left[X|Y=y\right]}\right). \end{array}$$
Inserting (Equation 13) in (Equation 2) we have the bound
(14)$$\begin{array}{c} h\left(X|Y\right)\leq E\left[\log\left(\pi e\;\mathrm{Var}\left[X|Y\right]\right)\right] \end{array}$$
which improves (Equation 5) in general, since $\mathrm{Var}\left[X|Y=y\right]$ is the MMSE of *X* given the event Y=y. In other words, we have $\mathrm{Var}\left[X|Y=y\right]\leq \sigma_{L}^{2}$ for all Y=y and the following bound improves (Equation 6) for CSCG *X*:(15)$$\begin{array}{c} I\left(X;Y\right)\geq E\left[\log\frac{E\left[{\left|X\right|}^{2}\right]}{\mathrm{Var}\left[X|Y\right]}\right]. \end{array}$$

In fact, the bound (Equation 15) was derived in ([50], Section III.B) by optimizing the GMI in (Equation 10) over all forward models of the form
(16)$$\begin{array}{c} q\left(y|x\right)=\exp\left(-{\left|{\tilde{g}}_{y}-{\tilde{f}}_{y}\;x\right|}^{2}\right) \end{array}$$
where ${\tilde{f}}_{y}$, ${\tilde{g}}_{y}$ depend on *y*; see also [47,48,49]. We provide a simple proof. By inserting (Equation 16) into (Equation 8) and (Equation 9), absorbing the *s* parameter in ${\tilde{f}}_{y}$ and ${\tilde{g}}_{y}$, and completing squares, one can equivalently optimize over all reverse densities of the form
(17)$$\begin{array}{cc} \; & q\left(x|y\right)=\exp\left(-{\left|g_{y}-f_{y}\;x\right|}^{2}+h_{y}\right) \end{array}$$
where $|f_{y}{|}^{2}=\pi e^{h_{y}}$ so that q(x|y) is a density. We next bound the cross-entropy as
(18)$$\begin{array}{cc} E\left[-\log q(X|Y=y)\right] & =E\left[{\left|g_{y}/f_{y}-X\right|}^{2}\right]{\left|f_{y}\right|}^{2}-h_{y}\\ \; & \geq \mathrm{Var}\left[X|Y=y\right]\pi e^{h_{y}}-h_{y} \end{array}$$
with equality if $g_{y}/f_{y}=E\left[X|Y=y\right]$; see Section 2.5. The RHS of (Equation 18) is minimized by $\mathrm{Var}\left[X|Y=y\right]\pi e^{h_{y}}=1$, so the best choice for $f_{y}$, $g_{y}$, $h_{y}$ gives the bound (Equation 14).

**Remark** **1.**
*The model (Equation 16) uses generalized nearest-neighbor decoding, improving the rules proposed in [42,43,44]. The authors of [50] pointed out that (Equation 6) and (Equation 15) use the LMMSE and MMSE, respectively; see ([50], Equation (Equation 87)).*


**Remark** **2.**
*A corresponding forward model can be based on (Equation 8) and (Equation 13), namely*

(19)
$$\begin{array}{c} q{\left(y|x\right)}^{s}=\frac{q(x|y)}{p(x)}\;\Rightarrow \;q\left(y\right)=1. \end{array}$$



**Remark** **3.**
*The RHS of (Equation 15) has a more complicated form than the RHS of (Equation 6) due to the outer expectation and conditional variance, and this makes optimizing X challenging when there is CSIR and CSIT. Also, if p(y|x) is known, then it seems sensible to numerically compute p(y) and I(X;Y) directly, e.g., via Monte Carlo or numerical integration.*


**Remark** **4.**
*Decoding rules for discrete X can be based on decision theory as well as estimation theory; see ([64], Equation (Equation 11)).*


*Forward Models:* Refinements of (Equation 11) appear in the optical fiber literature where the non-linear Schrödinger equation describes wave propagation [52]. Such channels exhibit complicated interactions of attenuation, dispersion, nonlinearity, and noise, and the channel density is too challenging to compute. One thus resorts to capacity lower bounds based on GMI and Monte Carlo simulation. The simplest models are memoryless, and they work well if chosen carefully. For example, the paper [52] used auxiliary models of the form
(20)$$\begin{array}{c} q\left(y|x\right)=\exp\left({-|y-hx|}^{2}/\sigma_{\left|x\right|}^{2}\right) \end{array}$$
where *h* accounts for attenuation and self-phase modulation, and where the noise variance $\sigma_{\left|x\right|}^{2}$ depends on |x|. Also, *X* was chosen to have concentric rings rather than a CSCG density. Subsequent papers applied progressively more sophisticated models with memory to better approximate the actual channel; see [53,54,55,56,57,58,59]. However, the rate gains over the model (Equation 20) are minor (≈12%) for 1000 km links, and the newer models do not suggest practical receiver structures.

A related application is short-reach fiber-optic systems that use direct detection (DD) receivers [65] with photodiodes. The paper [60] showed that sampling faster than the symbol rate increases the DD capacity. However, spectrally efficient filtering gives the channel a long memory, motivating auxiliary models q(y|x) with reduced memory to simplify GMI computations [61,66]. More generally, one may use channel-shortening filters [67,68,69] to increase the GMI.

**Remark** **5.**
*The ultimate GMI is I(X;Y), and one can compute this quantity numerically for the channels considered in this paper. We are motivated to focus on forward auxiliary models q(y|x) to understand how to improve information rates for more complex channels. For instance, simple q(y|x) let one understand properties of optimal codes, see Lemma 3, and they suggest explicit power control policies, see Theorem 2.*


**Remark** **6.**
*The paper [37] (see also ([2], Equation (3.3.45)) and ([70], Equation (Equation 6))) derives two capacity lower bounds for massive MIMO channels. These bounds are designed for problems where the fading parameters have small variance so that, in effect, $\sigma^{2}$ in (Equation 7) is small. We will instead encounter cases where $\sigma^{2}$ grows in proportion to $E\left[{\left|X\right|}^{2}\right]$ and the RHS of (Equation 6) quickly saturates as $E\left[{\left|X\right|}^{2}\right]$ grows; see Remark 20.*


### 1.5. Organization

This paper is organized as follows. Section 2 defines notation and reviews basic results. Section 3 develops two results for the GMI of scalar auxiliary models with AWGN:Proposition 1 in Section 3.1 states a known result, namely that the RHS of (Equation 6) is the maximum GMI for the AWGN auxiliary model (Equation 11) and a CSCG *X*.Lemma 1 in Section 3.2 generalizes Proposition 1 by partitioning the channel output alphabet into *K* subsets, K≥1. We use K=2 to establish capacity properties at high and low SNR.
Section 4 and Section 5 apply the GMI to channels with CSIT and CSIR.

Section 4.3 treats adaptive codewords and develops structural properties of their optimal distribution.Lemma 2 in Section 4.4 generalizes Proposition 1 to MIMO channels and adaptive codewords. The receiver models each transmit symbol as a weighted sum of the entries of the corresponding adaptive symbol.Lemma 3 in Section 4.5 states that the maximum GMI for scalar channels, an AWGN auxiliary model, adaptive codewords with jointly CSCG entries, and K=1 is achieved by using a conventional codebook where each symbol is modified based on the CSIT.Lemma 4 in Section 4.6 extends Lemma 3 to MIMO channels, including diagonal or parallel channels.Theorem 1 in Section 5.1 generalizes Lemma 3 to include CSIR; we use this result several times in Section 6.Lemma 5 in Section 5.3 generalizes Lemmas 1 and 2 by partitioning the channel output alphabet.

Section 6, Section 7 and Section 8 apply the GMI to fading channels with AWGN and illustrate the theory for on-off and Rayleigh fading.

Lemma 6 in Section 6 gives a general capacity upper bound.Section 6.5 introduces a class of power control policies for full CSIT. Theorem 2 develops the optimal policy with an MMSE form.Theorem 3 in Section 6.6 provides a quadratic waterfilling expression for the GMI with partial CSIR.

Section 9 develops theory for block fading channels with in-block feedback (or in-block CSIT) that is a function of the CSIR and past channel inputs and outputs.

Theorem 4 in Section 9.2 generalizes Lemma 4 to MIMO block fading channels;Section 9.3 develops capacity expressions in terms of directed information;Section 9.4 specializes the capacity to fading channels with AWGN and delayed CSIR;Proposition 3 generalizes Proposition 2 to channels with special CSIR and CSIT.

Section 10 concludes the paper. Finally, Appendix A, Appendix B, Appendix C, Appendix D, Appendix E, Appendix F and Appendix G provide results on special functions, GMI calculations, and proofs.

## 2. Preliminaries

### 2.1. Basic Notation

Let 1(·) be the indicator function that takes on the value 1 if its argument is true and 0 otherwise. Let δ(.) be the Dirac generalized function with $\int_{X}\delta \left(x\right)f\left(x\right)dx=f\left(0\right)\cdot 1\left(0\in X\right)$. For x∈R, define ${\left(x\right)}^{+}=\max\left(0,x\right)$. The complex-conjugate, absolute value, and phase of x∈C are written as $x^{*}$, |x|, and arg(x), respectively. We write $j=\sqrt{-1}$ and $\bar{\epsilon }=1-\epsilon$.

Sets are written with calligraphic font, e.g., S={1,…,n} and the cardinality of S is |S|. The complement of S in T is $S^{c}$ where T is understood from the context.

### 2.2. Vectors and Matrices

Column vectors are written as $\underline{x}={\left[x_{1},\ldots ,x_{M}\right]}^{T}$ where *M* is the dimension, and *T* denotes transposition. The complex-conjugate transpose (or Hermitian) of $\underline{x}$ is written as ${\underline{x}}^{†}$. The Euclidean norm of $\underline{x}$ is $∥\underline{x}∥$. Matrices are written with bold letters such as A. The letter I denotes the identity matrix. The determinant and trace of a square matrix A are written as detA and trA, respectively.

A singular value decomposition (SVD) is $A=U\Sigma V^{†}$ where U and V are unitary matrices and Σ is a rectangular diagonal matrix with the singular values of A on the diagonal. The square matrix A is positive semi-definite if ${\underline{x}}^{†}A\underline{x}\geq 0$ for all $\underline{x}$. The notation A⪯B means that B−A is positive semi-definite. Similarly, A is positive definite if ${\underline{x}}^{†}A\underline{x}>0$ for all $\underline{x}$, and we write A≺B if B−A is positive definite.

### 2.3. Random Variables

Random variables are written with uppercase letters, such as *X*, and their realizations with lowercase letters, such as *x*. We write the distribution of discrete *X* with alphabet X={0,…,n−1} as $P_{X}=\left[P_{X}\left(0\right),\ldots ,P_{X}\left(n-1\right)\right]$. The density of a real- or complex-valued *X* is written as $p_{X}$. Mixed discrete-continuous distributions are written using mixtures of densities and Dirac-δ functions.

Conditional distributions and densities are written as $P_{X|Y}$ and $p_{X|Y}$, respectively. We usually drop subscripts if the argument is a lowercase version of the random variable, e.g., we write p(y|x) for $p_{Y|X}\left(y|x\right)$. One exception is that we consistently write the distributions $P_{S_{R}}\left(.\right)$ and $P_{S_{T}}\left(.\right)$ of the CSIR and CSIT with the subscript to avoid confusion with power notation.

### 2.4. Second-Order Statistics

The expectation and variance of the complex-valued random variable *X* are $E\left[X\right]$ and $\mathrm{Var}\left[X\right]=E\left[|X-E\left[X\right]{|}^{2}\right]$, respectively. The correlation coefficient of $X_{1}$ and $X_{2}$ is $\rho =E\left[U_{1}U_{2}^{*}\right]$ where
$$\begin{array}{c} U_{i}=\left(X_{i}-E\left[X_{i}\right]\right)/\sqrt{\mathrm{Var}\left[X_{i}\right]} \end{array}$$
for i=1,2. We say that $X_{1}$ and $X_{2}$ are fully correlated if $\rho =e^{j\phi }$ for some real ϕ. Conditional expectation and variance are written as $E\left[X|A=a\right]$ and
$$\begin{array}{c} \mathrm{Var}\left[X|A=a\right]=E\left[\left(X-E\left[X\right]\right){\left(X-E\left[X\right]\right)}^{*}|A=a\right]. \end{array}$$
The expressions $E\left[X|A\right]$, $\mathrm{Var}\left[X|A\right]$ are random variables that take on the values $E\left[X|A=a\right]$, $\mathrm{Var}\left[X|A=a\right]$ if A=a.

The expectation and covariance matrix of the random column vector $\underline{X}={\left[X_{1},\ldots ,X_{M}\right]}^{T}$ are $E\left[\underline{X}\right]$ and $Q_{\underline{X}}=E\left[\left(\underline{X}-E\left[\underline{X}\right]\right){\left(\underline{X}-E\left[\underline{X}\right]\right)}^{†}\right]$, respectively. We write $Q_{\underline{X},\underline{Y}}$ for the covariance matrix of the stacked vector ${\left[{\underline{X}}^{T}{\underline{Y}}^{T}\right]}^{T}$. We write $Q_{\underline{X}|\underline{Y}=\underline{y}}$ for the covariance matrix of $\underline{X}$ conditioned on the event $\underline{Y}=\underline{y}$. $Q_{\underline{X}|\underline{Y}}$ is a random matrix that takes on the matrix value $Q_{\underline{X}|\underline{Y}=\underline{y}}$ when $\underline{Y}=\underline{y}$.

We often consider CSCG random variables and vectors. A CSCG $\underline{X}$ has density
$$\begin{array}{c} p\left(\underline{x}\right)=\frac{\exp\left(-{\underline{x}}^{†}\;Q_{\underline{X}}^{-1}\;\underline{x}\right)}{\pi^{M}\det Q_{\underline{X}}} \end{array}$$
and we write $\underline{X}∼\mathrm{CN}\left(\underline{0},Q_{\underline{X}}\right)$.

### 2.5. MMSE and LMMSE Estimation

Assume that $E\left[\underline{X}\right]=E\left[\underline{Y}\right]=\underline{0}$. The MMSE estimate of $\underline{X}$ given the event $\underline{Y}=\underline{y}$ is the vector $\hat{\underline{X}}\left(\underline{y}\right)$ that minimizes
$$\begin{array}{c} E\left[\left.{∥\underline{X}-\hat{\underline{X}}\left(\underline{y}\right)∥}^{2}\right|\underline{Y}=\underline{y}\right]. \end{array}$$Direct analysis gives ([71], Chapter 4)
(21)$$\begin{array}{cc} \; & \hat{\underline{X}}\left(\underline{y}\right)=E\left[\underline{X}|\underline{Y}=\underline{y}\right] \end{array}$$
(22)$$\begin{array}{cc} \; & E\left[∥\underline{X}-\hat{\underline{X}}{∥}^{2}\right]=E\left[∥\underline{X}{∥}^{2}\right]-E\left[∥\hat{\underline{X}}{∥}^{2}\right] \end{array}$$
(23)$$\begin{array}{cc} \; & Q_{\underline{X}-\hat{\underline{X}}}=Q_{\underline{X}}-Q_{\hat{\underline{X}}} \end{array}$$
(24)$$\begin{array}{cc} \; & E\left[\left(\underline{X}-\hat{\underline{X}}\right)\;{\underline{Y}}^{†}\right]=0 \end{array}$$
where the last identity is called the orthogonality principle.

The LMMSE estimate of $\underline{X}$ given $\underline{Y}$ with invertible $Q_{\underline{Y}}$ is the vector ${\hat{\underline{X}}}_{L}=C\;\underline{Y}$ where C is chosen to minimize $E\left[∥\underline{X}-{\hat{\underline{X}}}_{L}{∥}^{2}\right]$. We compute
(25)$$\begin{array}{cc} \; & {\hat{\underline{X}}}_{L}=E\left[\underline{X}\;{\underline{Y}}^{†}\right]Q_{\underline{Y}}^{-1}\;\underline{Y} \end{array}$$
and we also have the properties (22)–(24) with $\hat{\underline{X}}$ replaced by ${\hat{\underline{X}}}_{L}$. Moreover, if $\underline{X}$ and $\underline{Y}$ are jointly CSCG, then the MMSE and LMMSE estimators coincide, and the orthogonality principle (24) implies that the error $\underline{X}-\hat{\underline{X}}$ is independent of $\underline{Y}$, i.e., we have
(26)$$\begin{array}{cc} E\left[\left.\left(\underline{X}-\hat{\underline{X}}\right){\left(\underline{X}-\hat{\underline{X}}\right)}^{†}\right|\underline{Y}=\underline{y}\right] & =E\left[\left.\underline{X}\;{\underline{X}}^{†}\right|\underline{Y}=\underline{y}\right]-E\left[\underline{X}\;{\underline{Y}}^{†}\right]Q_{\underline{Y}}^{-1}\;\underline{y}\;{\underline{y}}^{†}Q_{\underline{Y}}^{-1}E{\left[\underline{X}\;{\underline{Y}}^{†}\right]}^{†}\\ \; & =Q_{\underline{X}}-Q_{\hat{\underline{X}}}. \end{array}$$

### 2.6. Entropy, Divergence, and Information

Entropies of random vectors with densities *p* are written as
$$\begin{array}{c} h\left(\underline{X}\right)=E\left[-\log p(\underline{X})\right],\;h\left(\underline{X}|\underline{Y}\right)=E\left[-\log p(\underline{X}|\underline{Y})\right] \end{array}$$
where we use logarithms to the base *e* for analysis. The informational divergence of the densities *p* and *q* is
$$\begin{array}{c} D\left(p∥q\right)=E\left[\log\frac{p(\underline{X})}{q(\underline{X})}\right] \end{array}$$
and D(p∥q)≥0 with equality if and only if p=q almost everywhere. The mutual information of $\underline{X}$ and $\underline{Y}$ is
$$\begin{array}{c} I\left(\underline{X};\underline{Y}\right)=D\left(p\left(\underline{X},\underline{Y}\right)\;∥\;p\left(\underline{X}\right)\;p\left(\underline{Y}\right)\right)=E\left[\log\frac{p(\underline{Y}|\underline{X})}{p(\underline{Y})}\right]. \end{array}$$
The average mutual information of $\underline{X}$ and $\underline{Y}$ conditioned on $\underline{Z}$ is $I(\underline{X};\underline{Y}|\underline{Z})$. We write strings as $X^{L}=\left(X_{1},X_{2},\ldots ,X_{L}\right)$ and use the directed information notation (see [9,72])
(27)$$\begin{array}{cc} \; & I\left(X^{L}\to Y^{L}|Z\right)=\sum_{\ell =1}^{L}I\left(X^{\ell };Y_{\ell }|Y^{\ell -1},Z\right) \end{array}$$
(28)$$\begin{array}{cc} \; & I(X^{L}\to Y^{L}∥Z^{L}\left|W\right)=\sum_{\ell =1}^{L}I\left(X^{\ell };Y_{\ell }|Y^{\ell -1},Z^{\ell },W\right) \end{array}$$
where $Y_{0}=0$.

### 2.7. Entropy and Information Bounds

The expression (Equation 2) applies to random vectors. Choosing $q(\underline{x}|\underline{y})$ as the conditional density where the $\underline{X},\underline{Y}$ are modeled as jointly CSCG we obtain a generalization of (Equation 5):(29)$$\begin{array}{cc} h(\underline{X}|\underline{Y}) & \leq \log\frac{\det\left(\pi e\;Q_{\underline{X},\underline{Y}}\right)}{\det\left(\pi e\;Q_{\underline{Y}}\right)}=\log\det\left(\pi e\left\{Q_{\underline{X}}-E\left[\underline{X}\;{\underline{Y}}^{†}\right]Q_{\underline{Y}}^{-1}E\left[\underline{Y}\;{\underline{X}}^{†}\right]\right\}\right). \end{array}$$
The vector generalization of (Equation 6) for CSCG $\underline{X}$ is
(30)$$\begin{array}{cc} I(\underline{X};\underline{Y}) & =h\left(\underline{X}\right)-h\left(\underline{X}|\underline{Y}\right)\\ \; & \geq \log\det\left({\left(Q_{\underline{X}}-E\left[\underline{X}\;{\underline{Y}}^{†}\right]Q_{\underline{Y}}^{-1}E\left[\underline{Y}\;{\underline{X}}^{†}\right]\right)}^{-1}Q_{\underline{X}}\right)\\ \; & \overset{\left(a\right)}{=}\log\det\left(I+Q_{\underline{Z}}^{-1}HQ_{\underline{X}}^{-1}H^{†}\right) \end{array}$$
where (cf. (Equation 7))
(31)$$\begin{array}{c} H=E\left[\underline{Y}\;{\underline{X}}^{†}\right]Q_{\underline{\bar{X}}}^{-1},\;Q_{\underline{Z}}=Q_{\underline{Y}}-HQ_{\underline{X}}H^{†} \end{array}$$
and step (a) in (Equation 30) follows by the Woodbury identity
(32)$$\begin{array}{c} {\left(A+BCD\right)}^{-1}=A^{-1}-A^{-1}B{\left(C^{-1}+DA^{-1}B\right)}^{-1}DA^{-1} \end{array}$$
and the Sylvester identity
(33)$$\begin{array}{cc} \; & \det\left(I+AB\right)=\det\left(I+BA\right). \end{array}$$
We also have vector generalizations of (Equation 14) and (Equation 15): (34)$$\begin{array}{cc} h(\underline{X}|\underline{Y}) & \leq E\left[\log\det\left(\pi e\;Q_{\underline{X}|\underline{Y}}\right)\right] \end{array}$$(35)$$\begin{array}{cc} I(\underline{X};\underline{Y}) & \geq E\left[\log\frac{\det Q_{\underline{X}}}{\det Q_{\underline{X}|\underline{Y}}}\right]\;\;\mathrm{for}\;\mathrm{CSC}G\;\underline{X}. \end{array}$$

### 2.8. Capacity and Wideband Rates

Consider the complex-alphabet AWGN channel with output Y=X+Z and noise Z∼CN(0,1). The capacity with the block power constraint $\frac{1}{n}\sum_{i=1}^{n}{\left|X_{i}\right|}^{2}\leq P$ is
(36)$$\begin{array}{c} C\left(P\right)=\max_{E\left[{\left|X\right|}^{2}\right]\leq P}I\left(X;Y\right)=\log\left(1+P\right). \end{array}$$

The low SNR regime (small *P*) is known as the wideband regime [73]. For well-behaved channels such as AWGN channels, the minimum $E_{b}/N_{0}$ and the slope *S* of the capacity vs. $E_{b}/N_{0}$ in bits/(3 dB) at the minimum $E_{b}/N_{0}$ are (see ([73], Equation (Equation 35)) and ([73], Theorem 9))
(37)$$\begin{array}{c} {\left.\frac{E_{b}}{N_{0}}\right|}_{\min}=\frac{\log2}{C^{\prime }\left(0\right)},\;S=\frac{2{\left[C^{\prime }\left(0\right)\right]}^{2}}{-C^{\prime \prime }\left(0\right)} \end{array}$$
where $C^{\prime }\left(P\right)$ and $C^{\prime \prime }\left(P\right)$ are the first and second derivatives of C(P) (measured in nats) with respect to *P*, respectively. For example, the wideband derivatives for (Equation 36) are $C^{\prime }\left(0\right)=1$ and $C^{\prime \prime }\left(0\right)=-1$ so that the wideband values (Equation 37) are
(38)$$\begin{array}{c} {\left.\frac{E_{b}}{N_{0}}\right|}_{\min}=\log2,\;S=2. \end{array}$$
The minimal $E_{b}/N_{0}$ is usually stated in decibels, for example $10\log_{10}\left(\log2\right)=-1.59$ dB. An extension of the theory to general channels is described in ([74], Section III).

**Remark** **7.**
*A useful method is flash signaling, where one sends with zero energy most of the time. In particular, we will consider the CSCG flash density*

(39)
$$\begin{array}{c} p\left(x\right)=\left(1-p\right)\;\delta \left(x\right)+p\;\frac{e^{-{\left|x\right|}^{2}/\left(P/p\right)}}{\pi (P/p)} \end{array}$$

*where 0<p≤1 so that the average power is $E\left[{\left|X\right|}^{2}\right]=P$. Note that flash signaling is defined in ([73], Definition 2) as a family of distributions satisfying a particular property as P→0. We use the terminology informally.*


### 2.9. Uniformly-Spaced Quantizer

Consider a uniformly-spaced scalar quantizer $q_{u}\left(.\right)$ with *B* bits, domain [0,∞), and reconstruction points
$$\begin{array}{c} s\in \{\Delta /2,3\Delta /2,\ldots ,\Delta /2+\left(2^{B}-1\right)\Delta \} \end{array}$$
where Δ>0. The quantization intervals are
$$\begin{array}{c} I\left(s\right)=\left\{\begin{array}{cc} \left[s-\frac{\Delta }{2},s+\frac{\Delta }{2}\right), & s\neq s_{\max}\\ \left[s-\frac{\Delta }{2},\infty \right), & s=s_{\max} \end{array}\right. \end{array}$$
where $s_{\max}=\Delta /2+\left(2^{B}-1\right)\Delta$. We will consider B=0,1,∞. For B=∞ we choose $q_{u}\left(x\right)=x$.

Suppose one applies the quantizer to the non-negative random variable *G* with density p(g) to obtain $S_{T}=q_{u}\left(G\right)$. Let $P_{S_{T}}$ and $P_{S_{T}|G}$ be the probability mass functions of $S_{T}$ without and with conditioning on *G*, respectively. We have
(40)$$\begin{array}{c} P_{S_{T}|G}\left(s|g\right)=1\left(g\in I(s)\right),\;P_{S_{T}}\left(s\right)=\int_{g\in I(s)}p\left(g\right)\;dg \end{array}$$
and using Bayes’ rule, we obtain
(41)$$\begin{array}{cc} p(g|s)\; & =\left\{\begin{array}{cc} p\left(g\right)/P_{S_{T}}\left(s\right), & g\in I(s)\\ 0, & \mathrm{else}. \end{array}\right. \end{array}$$

## 3. Generalized Mutual Information

We re-derive the GMI in the usual way, where one starts with the forward model q(y|x) rather than the reverse density q(x|y) in (Equation 8). Consider the joint density p(x,y) and define q(y) as in (Equation 9) for s≥0. Note that neither q(y|x) nor q(y) must be densities. The GMI is defined in [39] to be $\max_{s\geq 0}I_{s}\left(X;Y\right)$ where (see the RHS of (Equation 10))
(42)$$\begin{array}{c} I_{s}\left(X;Y\right)=E\left[\log\frac{q{\left(Y|X\right)}^{s}}{q(Y)}\right] \end{array}$$
and where the expectation is with respect to p(x,y). The GMI is a lower bound on the mutual information since
(43)$$\begin{array}{c} I_{s}\left(X;Y\right)=I\left(X;Y\right)-D\left(p_{X,Y}∥p_{Y}\;q_{X|Y}\right). \end{array}$$
Moreover, by using Gallager’s derivation of error exponents, but without modifying his “*s*” variable, the GMI $I_{s}\left(X;Y\right)$ is achievable with a mismatched decoder that uses q(y|x) for its decoding metric [39].

### 3.1. AWGN Forward Model with CSCG Inputs

A natural metric is based on the AWGN auxiliary channel $Y_{a}=hX+Z$ where *h* is a channel parameter and $Z∼\mathrm{CN}(0,\sigma^{2})$ is independent of *X*, i.e., we have the auxiliary model (here a density)
(44)$$\begin{array}{c} q\left(y|x\right)=\frac{1}{\pi \sigma^{2}}\exp\left({-|y-hx|}^{2}/\sigma^{2}\right) \end{array}$$
where *h* and $\sigma^{2}$ are to be optimized. A natural input is X∼CN(0,P) so that (Equation 9) is
(45)$$\begin{array}{c} q\left(y\right)=\frac{\pi \sigma^{2}/s}{{\left(\pi \sigma^{2}\right)}^{s}}\cdot \frac{\exp\left(\frac{-{\left|y\right|}^{2}}{\sigma^{2}/s+{\left|h\right|}^{2}P}\right)}{\pi (\sigma^{2}/s+|h{|}^{2}P)}. \end{array}$$
We have the following result, see [43] that considers channels of the form (Equation 1) and ([47], Proposition 1) that considers general p(y|x).

**Proposition** **1.**
*The maximum GMI (Equation 42) for the channel p(y|x), a CSCG input X with variance P>0, and the auxiliary model (Equation 44) with $\sigma^{2}>0$ is*

(46)
$$\begin{array}{c} I_{1}\left(X;Y\right)=\log\left(1+\frac{|\tilde{h}{|}^{2}P}{{\tilde{\sigma }}^{2}}\right) \end{array}$$

*where s=1 and (cf. (Equation 7))*

(47)
$$\begin{array}{cc} \tilde{h} & =E\left[YX^{*}\right]/P \end{array}$$


(48)
$$\begin{array}{cc} {\tilde{\sigma }}^{2} & =E\left[|Y-\tilde{h}{\;X|}^{2}\right]=E\left[{\left|Y\right|}^{2}\right]-{\left|\tilde{h}\right|}^{2}P. \end{array}$$

*The expectations are with respect to the actual density p(x,y).*


**Proof.** The GMI (Equation 42) for the model (Equation 44) is
(49)$$\begin{array}{cc} I_{s}\left(X;Y\right) & =\log\left(1+\frac{{\left|h\right|}^{2}P}{\sigma^{2}/s}\right)+\frac{E\left[{\left|Y\right|}^{2}\right]}{\sigma^{2}/s+{\left|h\right|}^{2}P}-\frac{E\left[{\left|Y-hX\right|}^{2}\right]}{\sigma^{2}/s}. \end{array}$$
Since (Equation 49) depends only on the ratio $\sigma^{2}/s$ one may as well set s=1. Thus, choosing $h=\tilde{h}$ and $\sigma^{2}={\tilde{\sigma }}^{2}$ gives (Equation 46).Next, consider $Y_{a}=\tilde{h}X+\tilde{Z}$ where $\tilde{Z}∼\mathrm{CN}\left(0,{\tilde{\sigma }}^{2}\right)$ is independent of *X*. We have
(50)$$\begin{array}{cc} \; & E\left[|Y_{a}{|}^{2}\right]=E\left[{\left|Y\right|}^{2}\right] \end{array}$$
(51)$$\begin{array}{cc} \; & E\left[|Y_{a}-\tilde{h}X{|}^{2}\right]=E\left[|Y-\tilde{h}X{|}^{2}\right]. \end{array}$$
In other words, the second-order statistics for the two channels with outputs *Y* (the actual channel output) and $Y_{a}$ are the same. But the GMI (Equation 46) is the mutual information $I(X;Y_{a})$. Using (Equation 43) and (Equation 49), for any *s*, *h* and $\sigma^{2}$ we have
(52)$$\begin{array}{c} I\left(X;Y_{a}\right)=\log\left(1+\frac{|\tilde{h}{|}^{2}P}{{\tilde{\sigma }}^{2}}\right)\geq I_{s}\left(X;Y_{a}\right)=I_{s}\left(X;Y\right) \end{array}$$
and equality holds if $h=\tilde{h}$ and $\sigma^{2}/s={\tilde{\sigma }}^{2}$. □

**Remark** **8.**
*The rate (Equation 46) is the same as the RHS of (Equation 6).*


**Remark** **9.**
*Proposition 1 generalizes to vector models and adaptive input symbols; see Section 4.4.*


**Remark** **10.**
*The estimate $\tilde{h}$ is the MMSE estimate of h:*

(53)
$$\begin{array}{c} \tilde{h}=\arg\min_{h}E\left[{\left|Y-hX\right|}^{2}\right] \end{array}$$

*and ${\tilde{\sigma }}^{2}$ is the variance of the error. To see this, expand*

(54)
$$\begin{array}{c} E\left[{\left|Y-hX\right|}^{2}\right]=E\left[|\left(Y-\tilde{h}X\right)+\left(\tilde{h}-h\right){X|}^{2}\right]={\tilde{\sigma }}^{2}+{\left|\tilde{h}-h\right|}^{2}P \end{array}$$

*where the final step follows by the definition of $\tilde{h}$ in (Equation 47).*


**Remark** **11.**
*Suppose that h is an estimate other than (Equation 53). Then if $E\left[{\left|Y\right|}^{2}\right]>E\left[{\left|Y-h\;X\right|}^{2}\right]$ we may choose*

(55)
$$\begin{array}{c} \sigma^{2}/s={\left|h\right|}^{2}P\cdot \frac{E\left[{\left|Y-h\;X\right|}^{2}\right]}{E\left[{\left|Y\right|}^{2}\right]-E\left[{\left|Y-h\;X\right|}^{2}\right]} \end{array}$$

*and the GMI (Equation 49) simplifies to*

(56)
$$\begin{array}{cc} I_{s}\left(X;Y\right) & =\log\left(\frac{E\left[{\left|Y\right|}^{2}\right]}{E\left[{\left|Y-h\;X\right|}^{2}\right]}\right). \end{array}$$



**Remark** **12.**
*The LM rate (for “lower bound to the mismatch capacity”) improves the GMI for some q(y|x) [40,75]. The LM rate replaces q(y|x) with $q\left(y|x\right)e^{t(x)/s}$ for some function t(.) and permits optimizing s and t(.); see ([41], Section 2.3.2). For example, if p(y|x) has the form $q{\left(y|x\right)}^{s}e^{t(x)}$ then the LM rate can be larger than the GMI; see [76,77].*


### 3.2. CSIR and *K*-Partitions

We consider two generalizations of Proposition 1. The first is for channels with a state $S_{R}$ known at the receiver but not at the transmitter. The second expands the class of CSCG auxiliary models. The motivation is to obtain more precise models under partial CSIR, especially to better deal with channels at high SNR and with high rates. We here consider discrete $S_{R}$ and later extend to continuous $S_{R}$.

*CSIR:* Consider the average GMI
(57)$$\begin{array}{c} I_{1}\left(X;Y|S_{R}\right)=\sum_{s_{R}}P_{S_{R}}\left(s_{R}\right)\;I_{1}\left(X;Y|S_{R}=s_{R}\right) \end{array}$$
where $I_{1}\left(X;Y|S_{R}=s_{R}\right)$ is the usual GMI where all densities are conditioned on $S_{R}=s_{R}$. The parameters (Equation 47) and (Equation 48) for the event $S_{R}=s_{R}$ are now
(58)$$\begin{array}{cc} \tilde{h}\left(s_{R}\right) & =\frac{E\left[\left.YX^{*}\right|S_{R}=s_{R}\right]}{E\left[\left.{\left|X\right|}^{2}\right|S_{R}=s_{R}\right]} \end{array}$$
(59)$$\begin{array}{cc} {\tilde{\sigma }}^{2}\left(s_{R}\right) & =E\left[\left.|Y-\tilde{h}\left(s_{R}\right){\;X|}^{2}\right|S_{R}=s_{R}\right]. \end{array}$$
The GMI (Equation 57) is thus
(60)$$\begin{array}{c} I_{1}\left(X;Y|S_{R}\right)=\sum_{s_{R}}P_{S_{R}}\left(s_{R}\right)\;\log\left(1+\frac{|\tilde{h}\left(s_{R}\right){|}^{2}P}{\tilde{\sigma }{\left(s_{R}\right)}^{2}}\right). \end{array}$$
*K-Partitions:* Let $\left\{Y_{k}:k=1,\ldots ,K\right\}$ be a *K*-partition of Y and define the auxiliary model
(61)$$\begin{array}{c} q\left(y|x\right)=\frac{1}{\pi \sigma_{k}^{2}}e^{-|y-h_{k}{x|}^{2}/\sigma_{k}^{2}},\;y\in Y_{k}. \end{array}$$

Observe that q(y|x) is not necessarily a density. We choose X∼CN(0,P) so that (Equation 9) becomes (cf. (Equation 45))
(62)$$\begin{array}{c} q\left(y\right)=\frac{\pi \sigma_{k}^{2}/s}{{\left(\pi \sigma_{k}^{2}\right)}^{s}}\cdot \frac{\exp\left(\frac{-{\left|y\right|}^{2}}{\sigma_{k}^{2}/s+{\left|h_{k}\right|}^{2}P}\right)}{\pi (\sigma_{k}^{2}/s+|h_{k}{|}^{2}P)},\;y\in Y_{k}. \end{array}$$
Define the events $E_{k}=\left\{Y\in Y_{k}\right\}$ for k=1,…,K. We have
(63)$$\begin{array}{cc} I_{s}\left(X;Y\right)\; & =\sum_{k=1}^{K}\mathrm{Pr}\left[E_{k}\right]\cdot E\left[\left.\log\frac{q{\left(Y|X\right)}^{s}}{q(Y)}\right|E_{k}\right] \end{array}$$
and inserting (Equation 61) and (Equation 62) we have the following lemma.

**Lemma** **1.**
*The GMI (Equation 42) for the channel p(y|x), s=1, a CSCG input X with variance P, and the auxiliary model (Equation 61) is (see (Equation 49))*

(64)
$$\begin{array}{c} I_{1}\left(X;Y\right)=\sum_{k=1}^{K}\mathrm{Pr}\left[E_{k}\right]\left[\log\left(1+\frac{|h_{k}{|}^{2}P}{\sigma_{k}^{2}}\right)+\frac{E\left[{\left|Y\right|}^{2}|E_{k}\right]}{\sigma_{k}^{2}+{\left|h_{k}\right|}^{2}P}-\frac{E\left[|Y-h_{k}X{|}^{2}|E_{k}\right]}{\sigma_{k}^{2}}\right]. \end{array}$$



**Remark** **13.**
*K-partitioning formally includes (Equation 57) as a special case by including $S_{R}$ as part of the receiver’s “overall” channel output $\tilde{Y}=\left[Y,S_{R}\right]$. For example, one can partition $\tilde{Y}$ as $\left\{{\tilde{Y}}_{s_{R}}:s_{R}\in S_{R}\right\}$ where ${\tilde{Y}}_{s_{R}}=Y\times \left\{s_{R}\right\}$.*


**Remark** **14.**
*The models (Equation 16) and (Equation 61) suggest building receivers based on adaptive Gaussian statistics. However, we are motivated to introduce (Equation 61) to prove capacity scaling results. For this purpose, we will use K=2 with the partition*

(65)
$$\begin{array}{c} E_{1}={\left\{|Y\right|}^{2}<t_{R}\},\;E_{2}={\left\{|Y\right|}^{2}\geq t_{R}\} \end{array}$$

*and $h_{1}=0$, $\sigma_{1}^{2}=1$. The GMI (Equation 64) thus has only the k=2 term and it remains to choose $h_{2}$, $\sigma_{2}^{2}$, and $t_{R}$.*


**Remark** **15.**
*One can generalize Lemma 1 and partition X×Y rather than Y only. However, the q(y) in (Equation 62) might not have a CSCG form.*


**Remark** **16.**
*Define $P_{k}=E\left[{\left|X\right|}^{2}|E_{k}\right]$ and choose the LMMSE auxiliary models with*

(66)
$$\begin{array}{cc} h_{k} & =E\left[\left.YX^{*}\right|E_{k}\right]/P_{k} \end{array}$$


(67)
$$\begin{array}{cc} \sigma_{k}^{2} & =E\left[\left.|Y-h_{k}{\;X|}^{2}\right|E_{k}\right]=E\left[\left.{\left|Y\right|}^{2}\right|E_{k}\right]-{\left|h_{k}\right|}^{2}P_{k} \end{array}$$

*for k=1,…,K. The expression (Equation 64) is then*

(68)
$$\begin{array}{c} I_{1}\left(X;Y\right)=\sum_{k=1}^{K}\mathrm{Pr}\left[E_{k}\right]\left[\log\left(1+\frac{|h_{k}{|}^{2}P}{E\left[{\left|Y\right|}^{2}|E_{k}\right]-{\left|h_{k}\right|}^{2}P_{k}}\right)-\frac{|h_{k}{|}^{2}\left(P-P_{k}\right)}{E\left[{\left|Y\right|}^{2}|E_{k}\right]+{\left|h_{k}\right|}^{2}\left(P-P_{k}\right)}\right]. \end{array}$$



**Remark** **17.**
*The LMMSE-based GMI (Equation 68) reduces to the GMI of Proposition 1 by choosing the trivial partition with K=1 and $Y_{1}=Y$. However, the GMI (Equation 68) may not be optimal for K≥2. What can be said is that the phase of $h_{k}$ in (Equation 64) should be the same as the phase of $E\left[YX^{*}|E_{k}\right]$ for all k. We thus have K two-dimensional optimization problems, one for each pair $\left(|h_{k}|,\sigma_{k}^{2}\right)$, k=1,…,K.*


**Remark** **18.**
*Suppose we choose a different auxiliary model for each Y=y, i.e., consider K→∞. The reverse density GMI uses the auxiliary model (Equation 19) which gives the RHS of (Equation 15):*

(69)
$$\begin{array}{c} I_{1}\left(X;Y\right)=\int_{C}\;p\left(y\right)\log\frac{P}{\mathrm{Var}\left[X|Y=y\right]}\;dy. \end{array}$$

*Instead, the suboptimal (Equation 68) is the complicated expression*

(70)
$$\begin{array}{cc} I_{1}\left(X;Y\right)=\int_{C}\;p\left(y\right) & \left[\log\left(1+\frac{|E\left[X|Y=y\right]{|}^{2}\left(P/P_{y}\right)}{\mathrm{Var}\left[X|Y=y\right]}\right)\right.\\ \; & \;\left.-\;\frac{|E\left[X|Y=y\right]{|}^{2}\left(P/P_{y}-1\right)}{\mathrm{Var}\left[X|Y=y\right]+{\left|E\left[X|Y=y\right]\right|}^{2}\left(P/P_{y}\right)}\right]\;dy. \end{array}$$

*where $P_{y}=E\left[{\left|X\right|}^{2}|Y=y\right]$. We show how to compute these GMIs in Appendix C.*


### 3.3. Example: On-Off Fading

Consider the channel Y=HX+Z where H,X,Z are mutually independent, $P_{H}\left(0\right)=P_{H}\left(\sqrt{2}\right)=1/2$, and Z∼CN(0,1). The channel exhibits particularly simple fading, giving basic insight into more realistic fading models. We consider two basic scenarios: full CSIR and no CSIR.

*Full CSIR:* Suppose $S_{R}=H$ and
(71)$$\begin{array}{c} q\left(y|x,h\right)=p\left(y|x,h\right)=\frac{1}{\pi \sigma^{2}}e^{{-|y-hx|}^{2}/\sigma^{2}} \end{array}$$
which corresponds to having (Equation 58) and (Equation 59) as
(72)$$\begin{array}{c} \tilde{h}\left(0\right)=0,\;\tilde{h}\left(\sqrt{2}\right)=\sqrt{2},\;{\tilde{\sigma }}^{2}\left(0\right)=\sigma^{2}\left(\sqrt{2}\right)=1. \end{array}$$
The GMI (Equation 60) with X∼CN(0,P) thus gives the capacity
(73)$$\begin{array}{c} C\left(P\right)=\frac{1}{2}\log\left(1+2P\right). \end{array}$$
The wideband values (Equation 37) are
(74)$$\begin{array}{c} {\left.\frac{E_{b}}{N_{0}}\right|}_{\min}=\log2,\;S=1. \end{array}$$
Compared with (Equation 38), the minimal $E_{b}/N_{0}$ is the same as without fading, namely −1.59 dB. However, fading reduces the capacity slope *S*; see the dashed curve in Figure 1.

*No CSIR:* Suppose $S_{R}=0$ and X∼CN(0,P) and consider the densities
(75)$$\begin{array}{cc} \; & p\left(y|x\right)=\frac{e^{-{\left|y\right|}^{2}}}{2\pi }+\frac{e^{-|y-\sqrt{2}{x|}^{2}}}{2\pi } \end{array}$$
(76)$$\begin{array}{cc} \; & p\left(y\right)=\frac{e^{-{\left|y\right|}^{2}}}{2\pi }+\frac{e^{-{\left|y\right|}^{2}/\left(1+2P\right)}}{2\pi (1+2P)}. \end{array}$$
The mutual information can be computed by numerical integration or by Monte Carlo integration:(77)$$\begin{array}{c} I\left(X;Y\right)\approx \frac{1}{N}\sum_{i=1}^{N}\log\frac{p_{Y|X}\left(y_{i}|x_{i}\right)}{p_{Y}\left(y_{i}\right)} \end{array}$$
where the RHS of (Equation 77) converges to I(X;Y) for long strings $x^{N},y^{N}$ sampled from p(x,y). The results for X∼CN(0,P) are shown in Figure 1 as the curve labeled “I(X;Y) Gauss”.

Next, Proposition 1 gives $h=1/\sqrt{2}$, $\sigma^{2}=1+P/2$, and
(78)$$\begin{array}{c} I_{1}\left(X;Y\right)=\log\left(1+\frac{P}{2+P}\right). \end{array}$$
The wideband values (Equation 37) are
(79)$$\begin{array}{c} {\left.\frac{E_{b}}{N_{0}}\right|}_{\min}=\log4,\;S=2/3 \end{array}$$
so the minimal $E_{b}/N_{0}$ is 1.42 dB and the capacity slope *S* has decreased further. Moreover, the rate saturates at large SNR at 1 bit per channel use.

The “I(X;Y) Gauss” curve in Figure 1 suggests that the no-CSIR capacity approaches the full-CSIR capacity for large SNR. To prove this, consider the K=2 partition specified in Remark 14 with $h_{1}=0$, $h_{2}=\sqrt{2}$, and $\sigma_{2}^{2}=1$. Since we are not using LMMSE auxiliary models, we must compute the GMI using the general expression (Equation 64), which is
(80)$$\begin{array}{c} I_{1}\left(X;Y\right)=\mathrm{Pr}\left[E_{2}\right]\left[\log\left(1+2P\right)+\frac{E\left[{\left|Y\right|}^{2}|E_{2}\right]}{1+2P}-E\left[{\left|Y-\sqrt{2}X\right|}^{2}|E_{2}\right]\right]. \end{array}$$
In Section B.1, we show that choosing $t_{R}=P^{\lambda_{R}}+b$ where $0<\lambda_{R}<1$ and *b* is a real constant makes all terms behave as desired as *P* increases:(81)$$\begin{array}{c} \mathrm{Pr}\left[E_{2}\right]\to 1/2,\;\frac{E\left[{\left|Y\right|}^{2}|E_{2}\right]}{1+2P}\to 1,\;E\left[\left.{\left|Y-\sqrt{2}X\right|}^{2}\right|E_{2}\right]\to 1. \end{array}$$
The GMI (Equation 80) of Lemma 1 thus gives the maximal value (Equation 73) for large *P*:(82)$$\begin{array}{c} \lim_{P\to \infty }\left[\frac{1}{2}\log\left(1+2P\right)-I_{1}\left(X;Y\right)\right]=0. \end{array}$$
Figure 1 shows the behavior of $I_{1}\left(X;Y\right)$ for K=2, $\lambda_{R}=0.4$, and b=3. Effectively, at large SNR, the receiver can estimate *H* accurately, and one approaches the full-CSIR capacity.

**Remark** **19.**
*For on-off fading, one may compute I(X;Y) directly and use the densities (Equation 75) and (Equation 76) to decode. Nevertheless, the partitioning of Lemma 1 helps prove the capacity scaling (Equation 82).*


Consider next the reverse density GMI (Equation 69) and the forward model GMI (Equation 70). Section C.1 shows how to compute $E\left[X|Y=y\right]$, $E\left[{\left|X\right|}^{2}|Y=y\right]$, and $\mathrm{Var}\left[X|Y=y\right]$, and Figure 1 plots the GMIs as the curves labeled “rGMI” and “GMI, K = *∞*”, respectively. The rGMI curve gives the best possible rates for AWGN auxiliary models, as shown in Section 1.4. The results also show that the large-*K* GMI (Equation 70) is worse than the K=1 GMI at low SNR but better than the K=2 GMI of Remark 14.

Finally, the curve labeled “I(X;Y) Gauss” in Figure 1 suggests that the minimal $E_{b}/N_{0}$ is 1.42 dB even for the capacity-achieving distribution. However, we know from ([73], Theorem 1) that flash signaling (Equation 39) can approach the minimal $E_{b}/N_{0}$ of −1.59 dB. For example, the flash rates I(X;Y) with p=0.05 are plotted in Figure 1. Unfortunately, the wideband slope is S=0 ([73], Theorem 17), and one requires very large flash powers (very small *p*) to approach −1.59 dB.

**Remark** **20.**
*As stated in Remark 6, the paper [37] (see also [2,70]) derives two capacity lower bounds. These bounds are the same for our problem, and they are derived using the following steps (see ([37], Lemmas 3 and 4)):*

(83)
$$\begin{array}{cc} I(X;Y) & =I\left(X,S_{H};Y\right)-I\left(S_{H};Y|X\right)\\ \; & \geq I\left(X;Y|S_{H}\right)-I\left(S_{H};Y|X\right). \end{array}$$

*Now consider Y=HX+Z where H,X,Z are mutually independent, $S_{H}=H$, $\mathrm{Var}\left[Z\right]=1$, and X∼CN(0,P). We have*

(84)
$$\begin{array}{cc} I(X;Y|H) & \geq E\left[\log(1+|H{|}^{2}P)\right] \end{array}$$


(85)
$$\begin{array}{cc} I(H;Y|X) & =h(Y|X)-h(Z)\\ & \leq \log\left(\pi e(1+\mathrm{Var}\left[H\right]P)\right)-h\left(Z\right) \end{array}$$

*where (Equation 84) and (Equation 85) follow by (Equation 5), in the latter case with the roles of X and Y reversed. The bound (Equation 85) works well if $\mathrm{Var}\left[H\right]$ is small, as for massive MIMO with “channel hardening”. However, for our on-off fading model, the bound (Equation 83) is*

(86)
$$\begin{array}{cc} I(X;Y) & \geq E\left[\log\left(1+{\left|H\right|}^{2}P\right)\right]-\log\left(1+\mathrm{Var}\left[H\right]P\right)\\ \; & =\frac{1}{2}\log\left(1+2P\right)-\log\left(1+P/2\right) \end{array}$$

*which is worse than the K=1 and K=∞ GMIs and is not shown in Figure 1.*


## 4. Channels with CSIT

This section studies Shannon’s channel with side information, or state, known causally at the transmitter [5,6]. We begin by treating general channels and then focus mainly on complex-alphabet channels. The capacity expression has a random variable *A* that is either a list (for discrete-alphabet states) or a function (for continuous-alphabet states). We refer to *A* as an adaptive symbol of an adaptive codeword.

### 4.1. Model

The problem is specified by the functional dependence graph (FDG) in Figure 2. The model has a message *M*, a CSIT string $S_{T}^{n}$, and a noise string $Z^{n}$. The variables *M*, $S_{T}^{n}$, $Z^{n}$ are mutually statistically independent, and $S_{T}^{n}$ and $Z^{n}$ are strings of i.i.d. random variables with the same distributions as $S_{T}$ and *Z*, respectively. $S_{T}^{n}$ is available causally at the transmitter, i.e., the channel input $X_{i}$, i=1,…,n, is a function of *M* and the sub-string $S_{T}^{i}$. The receiver sees the channel outputs
(87)$$\begin{array}{c} Y_{i}=f\left(X_{i},S_{Ti},Z_{i}\right) \end{array}$$
for some function f(.) and i=1,2,…,n.

Each $A_{i}$ represents a list of possible choices of $X_{i}$ at time *i*. More precisely, suppose that $S_{T}$ has alphabet $S_{T}=\left\{0,1,\ldots ,\nu -1\right\}$ and define the adaptive symbol
$$\begin{array}{c} A=\left(X(0),\ldots ,X(\nu -1)\right) \end{array}$$
whose entries have alphabet X. Here $S_{T}=s_{T}$ means that $X(s_{T})$ is transmitted, i.e., we have $X=X(S_{T})$. If $S_{T}$ has a continuous alphabet, we make *A* a function rather than a list, and we may again write $X=X(S_{T})$. Some authors therefore write *A* as X(.). (Shannon in [6] denoted our *A* and *X* as the respective *X* and *x*.)

**Remark** **21.**
*The conventional choice for A if X=C is*

(88)
$$\begin{array}{c} A=\left(\sqrt{P(0)}\;e^{j\phi (0)},\ldots ,\sqrt{P(\nu -1)}\;e^{j\phi (\nu -1)}\right)\cdot U \end{array}$$

*where U has $E\left[{\left|U\right|}^{2}\right]=1$, $P\left(s_{T}\right)=E\left[|X\left(s_{T}\right){|}^{2}\right]$, and $\phi (s_{T})$ is a phase shift. The interpretation is that U represents the symbol of a conventional codebook without CSIT, and these symbols are scaled and rotated. In other words, one separates the message-carrying U from an adaptation due to $S_{T}$ via*

(89)
$$\begin{array}{c} X=\sqrt{P(S_{T})}\;e^{j\phi (S_{T})}\;U. \end{array}$$



**Remark** **22.**
*One may define the channel by the functional relation (Equation 87), by p(y|a), or by $p(y|x,s_{T})$; see Shannon’s emphasis in ([6], Theorem); see ([22], Remark 3). We generally prefer to use p(y|a) since we interpret A as a channel input.*


**Remark** **23.**
*One can add feedback and let $X_{i}$ be a function of $\left(M,S_{T}^{i},Y^{i-1}\right)$, but feedback does not increase the capacity if the state and noise processes are memoryless ([22], Section V).*


**Remark** **24.**
*The model (Equation 87) permits block fading and MIMO transmission by choosing $X_{i}$ and $Y_{i}$ as vectors [11,78].*


### 4.2. Capacity

The capacity of the model under study is (see [6])
(90)$$\begin{array}{c} C=\max_{A}\;I\left(A;Y\right) \end{array}$$
where $A-[S_{T},X]-Y$ forms a Markov chain. One may limit attention to *A* with cardinality |A| satisfying (see ([22], Equation (Equation 56)), [79], ([80], Theorem 1))
(91)$$\begin{array}{c} \left|A\right|\leq \min\left(\left|Y|,1+\right|S_{T}\left|(|X|-1\right)\right). \end{array}$$
As usual, for the cost function c(x,y) and the average block cost constraint
(92)$$\begin{array}{c} \frac{1}{n}\sum_{i=1}^{n}E\left[c(X_{i},Y_{i})\right]\leq P \end{array}$$
the unconstrained maximization in (Equation 90) becomes a constrained maximization over the *A* for which $E\left[c(X,Y)\right]\leq P$. Also, a simple upper bound on the capacity is
(93)$$\begin{array}{cc} C(P) & \leq \max_{A:\;E\left[c(X,Y)\right]\leq P}\;I\left(A;Y,S_{T}\right)\\ \; & \overset{\left(a\right)}{=}\max_{X\left(S_{T}\right):\;E\left[c(X\left(S_{T}\right),Y)\right]\leq P}\;I\left(X;Y|S_{T}\right) \end{array}$$
where step (a) follows by the independence of *A* and $S_{T}$. This bound is tight if the receiver knows $S_{T}$.

**Remark** **25.**
*The chain rule for mutual information gives*

(94)
$$\begin{array}{cc} I(A;Y) & =I\left(X(0)\ldots X(\nu -1);Y\right) \end{array}$$


(95)
$$\begin{array}{cc} \; & =\sum_{s_{T}=0}^{\nu -1}I\left(X\left(s_{T}\right);Y|X\left(0\right),\ldots ,X\left(s_{T}-1\right)\right). \end{array}$$

*The RHS of (Equation 94) suggests treating the channel as a multi-input, single-output (MISO) channel, and the expression (Equation 95) suggests using multi-level coding with multi-stage decoding [81]. For example, one may use polar coded modulation [82,83,84] with Honda-Yamamoto shaping [85,86].*


**Remark** **26.**
*For X=C and the conventional adaptive symbol (Equation 88), we compute I(A;Y)=I(U;Y) and*

(96)
$$\begin{array}{c} C\left(P\right)=\max_{P\left(S_{T}\right),\phi \left(S_{T}\right):\;E\left[c(X\left(S_{T}\right),Y)\right]\leq P}\;I\left(U;Y\right). \end{array}$$



### 4.3. Structure of the Optimal Input Distribution

Let A be the alphabet of *A* and let X=C, i.e., we have $A=C^{\nu }$ for discrete $S_{T}$. Consider the expansions
(97)$$\begin{array}{cc} p(y|a) & =\sum_{s_{T}}P_{S_{T}}\left(s_{T}\right)\;p\left(y|x\left(s_{T}\right),s_{T}\right)\\ p(y) & =\int_{A}p\left(a\right)\;p\left(y|a\right)\;da \end{array}$$
(98)$$\begin{array}{cc} \; & =\sum_{s_{T}}P_{S_{T}}\left(s_{T}\right)\int_{C}p\left(x\left(s_{T}\right)\right)\;p\left(y|x\left(s_{T}\right),s_{T}\right)\;dx\left(s_{T}\right). \end{array}$$
Observe that p(y), and hence h(Y), depends only on the marginals $p(x\left(s_{T}\right))$ of *A*; see ([80], Section III). So define the set of densities having the same marginals as *A*:$$\begin{array}{c} P\left(A\right)=\left\{p\left(\tilde{a}\right):p\left(\tilde{x}\left(s_{T}\right)\right)=p\left(x\left(s_{T}\right)\right)\;\mathrm{for}\;\mathrm{all}\;s_{T}\in S_{T}\right\}. \end{array}$$This set is convex, since for any $p^{\left(1\right)}\left(a\right),p^{\left(2\right)}\left(a\right)\in P\left(A\right)$ and 0≤λ≤1 we have
(99)$$\begin{array}{c} \lambda p^{\left(1\right)}\left(a\right)+\left(1-\lambda \right)p^{\left(2\right)}\left(a\right)\in P\left(A\right). \end{array}$$Moreover, for fixed p(y), the expression I(A;Y) is a convex function of p(a|y), and p(a|y)=p(a)p(y|a)/p(y) is a linear function of p(a). Maximizing I(A;Y) over P(A) is thus the same as minimizing the concave function h(Y|A) over the convex set P(A). An optimal p(a) is thus an extreme of P(A). Some properties of such extremes are developed in [87,88].

For example, consider $|S_{T}|=2$ and $X=S_{T}=\left\{0,1\right\}$, for which (Equation 91) states that at most |A|=3 adaptive symbols need have positive probability (and at most |A|=2 adaptive symbols if |Y|=2). Suppose the marginals have $P_{X(0)}\left(0\right)=1/2$, $P_{X(1)}\left(0\right)=3/4$ and consider the matrix notation
$$\begin{array}{c} P_{A}=\left[\begin{array}{cc} P_{A}\left(0,0\right) & P_{A}\left(0,1\right)\\ P_{A}\left(1,0\right) & P_{A}\left(1,1\right) \end{array}\right] \end{array}$$
where we write $P_{A}\left(x_{1},x_{2}\right)$ for $P_{A}\left(\left[x_{1},x_{2}\right]\right)$. The optimal $P_{A}$ must then be one of the two extremes
(100)$$\begin{array}{c} \begin{array}{c} P_{A}=\left[\begin{array}{cc} 1/2 & 0\\ 1/4 & 1/4 \end{array}\right],\;P_{A}=\left[\begin{array}{cc} 1/4 & 1/4\\ 1/2 & 0 \end{array}\right]. \end{array} \end{array}$$For the first $P_{A}$, the codebook has the property that if X(0)=0 then X(1)=0 while if X(0)=1 then X(1) is uniformly distributed over X={0,1}.

Next, consider $|S_{T}|=2$ and marginals $P_{X(0)}$, $P_{X(1)}$ that are uniform over X={0,1,…,|X|−1}. This case was treated in detail in ([80], Section VI.A), see also [89], and we provide a different perspective. A classic theorem of Birkhoff [90] ensures that the extremes of P(A) are the |X|! distributions $P_{A}$ for which the |X|×|X| matrix
$$\begin{array}{c} P_{A}=\left[\begin{array}{ccc} P_{A}\left(0,0\right) & \ldots & P_{A}\left(0,\left|X\right|-1\right)\\ \vdots & \ddots & \vdots \\ P_{A}\left(\left|X\right|-1,0\right) & \ldots & P_{A}\left(\left|X\right|-1,\left|X\right|-1\right) \end{array}\right]. \end{array}$$
is a permutation matrix multiplied by 1/|X|. For example, for |X|=2 we have the two extremes
(101)$$\begin{array}{c} \begin{array}{c} P_{A}=\frac{1}{2}\left[\begin{array}{cc} 1 & 0\\ 0 & 1 \end{array}\right],\;P_{A}=\frac{1}{2}\left[\begin{array}{cc} 0 & 1\\ 1 & 0 \end{array}\right]. \end{array} \end{array}$$
The permutation property means that $X(s_{T})$ is a function of X(0), i.e., the encoding simplifies to a conventional codebook as in Remark 21 with uniformly-distributed *U* and a permutation $\pi_{s_{T}}\left(.\right)$ indexed by $s_{T}$ such that $X\left(S_{T}\right)=\pi_{S_{T}}\left(U\right)$. For example, for the first $P_{A}$ in (Equation 101) we may choose $X(S_{T})=U$, which is independent of $S_{T}$. On the other hand, for the second $P_{A}$ in (Equation 101) we may choose $X\left(S_{T}\right)=U⊕S_{T}$ where ⊕ denotes addition modulo-2.

For $|S_{T}|>2$, the geometry of P(A) is more complicated; see ([80], Section VI.B). For example, consider X={0,1} and suppose the marginals $P_{X(s_{T})}$, $s_{T}\in S_{T}$, are all uniform. Then the extremes include $P_{A}$ related to linear codes and their cosets, e.g., two extremes for $|S_{T}|=3$ are related to the repetition code and single parity check code:$$\begin{array}{c} \begin{array}{c} P_{A}\left(a\right)=1/2,\;a\in \left\{\left[0,0,0\right],\left[1,1,1\right]\right\}\\ P_{A}\left(a\right)=1/4,\;a\in \left\{\left[0,0,0\right],\left[0,1,1\right],\left[1,0,1\right],\left[1,1,0\right]\right\}. \end{array} \end{array}$$
This observation motivates concatenated coding, where the message is first encoded by an outer encoder followed by an inner code that is the coset of a linear code. The transmitter then sends the entries at position $S_{T}$ of the inner codewords, which are vectors of dimension $|S_{T}|$. We do not know if there are channels for which such codes are helpful.

### 4.4. Generalized Mutual Information

Consider the vector channel $p(\underline{y}|\underline{x})$ with input set $X=C^{M}$ and output set $Y=C^{N}$. The GMI for adaptive symbols is $\max_{s\geq 0}I_{s}\left(A;\underline{Y}\right)$ where
(102)$$\begin{array}{c} I_{s}\left(A;\underline{Y}\right)=E\left[\log\frac{q{\left(\underline{Y}|A\right)}^{s}}{q(\underline{Y})}\right] \end{array}$$
and the expectation is with respect to $p(a,\underline{y})$. Suppose the auxiliary model is $q(\underline{y}|a)$ and define
(103)$$\begin{array}{c} q\left(\underline{y}\right)=\int_{A}p\left(a\right)q{\left(\underline{y}|a\right)}^{s}\;da. \end{array}$$
The GMI again provides a lower bound on the mutual information since (cf. (Equation 43))
(104)$$\begin{array}{c} I_{s}\left(A;\underline{Y}\right)=I\left(A;\underline{Y}\right)-D\left(p_{A,\underline{Y}}∥p_{\underline{Y}}\;q_{A|\underline{Y}}\right) \end{array}$$
where $q\left(a|\underline{y}\right)=p\left(a\right)q{\left(\underline{y}|a\right)}^{s}/q\left(\underline{y}\right)$ is a reverse channel density.

We next study reverse and forward models as in Section 1.3 and Section 1.4. Suppose the entries $\underline{X}\left(s_{T}\right)$ of *A* are jointly CSCG.

*Reverse Model:* We write $\underline{A}$ when we consider *A* to be a column vector that stacks the $\underline{X}\left(s_{T}\right)$. Consider the following reverse density motivated by (Equation 13):(105)$$\begin{array}{c} q\left(\underline{a}|\underline{y}\right)=\frac{\exp\left(-{\left(\underline{a}-E\left[\underline{A}|\underline{Y}=\underline{y}\right]\right)}^{†}Q_{\underline{A}|\underline{Y}=\underline{y}}^{-1}\left(\underline{a}-E\left[\underline{A}|\underline{Y}=\underline{y}\right]\right)\right)}{\pi^{\nu M}\det Q_{\underline{A}|\underline{Y}=\underline{y}}}. \end{array}$$
A corresponding forward model is $q\left(\underline{y}|a\right)=q\left(a|\underline{y}\right)/p\left(a\right)$ and the GMI with s=1 becomes (cf. (Equation 35))
(106)$$\begin{array}{c} I_{1}\left(A;\underline{Y}\right)=E\left[\log\frac{\det Q_{\underline{A}}}{\det Q_{\underline{A}|\underline{Y}}}\right]. \end{array}$$
To simplify, one may focus on adaptive symbols as in (Equation 89):(107)$$\begin{array}{c} \underline{X}=Q_{\underline{X}\left(S_{T}\right)}^{1/2}\cdot \underline{U} \end{array}$$
where $\underline{U}∼\mathrm{CN}\left(\underline{0},I\right)$ and the $Q_{\underline{X}\left(s_{T}\right)}$ are covariance matrices. We thus have $I\left(A;\underline{Y}\right)=I\left(\underline{U};\underline{Y}\right)$ (cf. (Equation 96)) and using (Equation 105) but with $\underline{A}$ replaced with $\underline{U}$ we obtain
(108)$$\begin{array}{c} I_{1}\left(A;\underline{Y}\right)=E\left[-\log\det Q_{\underline{U}|\underline{Y}}\right]. \end{array}$$

*Forward Model:* Perhaps the simplest forward model is $q\left(\underline{y}|a\right)=p\left(\underline{y}|\underline{x}\left(s_{T}\right)\right)$ for some fixed value $s_{T}\in S_{T}$. One may interpret this model as having the receiver assume that $S_{T}=s_{T}$. A natural generalization of this idea is as follows: define the auxiliary vector
(109)$$\begin{array}{c} \underline{\bar{X}}=\sum_{s_{T}}W\left(s_{T}\right)\;\underline{X}\left(s_{T}\right) \end{array}$$
where the $W(s_{T})$ are M×M complex matrices, i.e., $\underline{\bar{X}}$ is a linear function of the entries of $A=[\underline{X}\left(s_{T}\right):s_{T}\in S_{T}]$. For example, the matrices might be chosen based on $P_{S_{T}}\left(.\right)$. However, observe that $\underline{\bar{X}}$ is independent of $S_{T}$. Now define the auxiliary model
$$\begin{array}{c} q\left(\underline{y}|a\right)=q\left(\underline{y}|\underline{\bar{x}}\right) \end{array}$$
where we abuse notation by using the same q(.). The expression (Equation 103) becomes
(110)$$\begin{array}{cc} q(\underline{y}) & =\int_{A}p\left(a\right)\;q{\left(\underline{y}|a\right)}^{s}\;da=\int_{C}p\left(\underline{\bar{x}}\right)\;q{\left(\underline{y}|\underline{\bar{x}}\right)}^{s}\;d\underline{\bar{x}}. \end{array}$$

**Remark** **27.**
*We often consider $S_{T}$ to be a discrete set, but for CSCG channels we also consider $S_{T}=C$ so that the sum over $S_{T}$ in (Equation 109) is replaced by an integral over C.*


We now specialize further by choosing the auxiliary channel ${\underline{Y}}_{a}=H\;\underline{\bar{X}}+\underline{Z}$ where H is an N×M complex matrix, $\underline{Z}$ is an *N*-dimensional CSCG vector that is independent of $\underline{\bar{X}}$ and has invertible covariance matrix $Q_{\underline{Z}}$, and H and $Q_{\underline{Z}}$ are to be optimized. Further choose $A=[\underline{X}\left(s_{T}\right):s_{T}\in S_{T}]$ whose entries are jointly CSCG with correlation matrices
$$\begin{array}{c} R\left(s_{T1},s_{T2}\right)=E\left[\underline{X}\left(s_{T1}\right)\underline{X}{\left(s_{T2}\right)}^{†}\right]. \end{array}$$
Since $\underline{\bar{X}}$ in (Equation 109) is independent of $S_{T}$, we have
(111)$$\begin{array}{c} q\left(\underline{y}|a\right)=\frac{\exp\left(-{\left(\underline{y}-H\;\underline{\bar{x}}\right)}^{†}Q_{\underline{Z}}^{-1}\left(\underline{y}-H\;\underline{\bar{x}}\right)\right)}{\pi^{N}\det Q_{\underline{Z}}}. \end{array}$$
Moreover, $\underline{\bar{X}}$ is CSCG so (Equation 110) is
$$\begin{array}{c} q\left(\underline{y}\right)=\frac{\pi^{N}\det\left(Q_{\underline{Z}}/s\right)}{{\left(\pi^{N}\det Q_{\underline{Z}}\right)}^{s}}\cdot \frac{\exp\left(-{\underline{y}}^{†}{\left(Q_{\underline{Z}}/s+HQ_{\underline{\bar{X}}}H^{†}\right)}^{-1}\underline{y}\right)}{\pi^{N}\det\left(Q_{\underline{Z}}/s+HQ_{\underline{\bar{X}}}H^{†}\right)} \end{array}$$
where
$$\begin{array}{cc} Q_{\underline{\bar{X}}}\; & =\sum_{s_{T1},s_{T_{2}}}W\left(s_{T1}\right)R\left(s_{T1},s_{T2}\right)W{\left(s_{T2}\right)}^{†}. \end{array}$$
We have the following generalization of Proposition 1.

**Lemma** **2.**
*The maximum GMI (Equation 102) for the channel $p(\underline{y}|a)$, an adaptive vector $A=[\underline{X}\left(s_{T}\right):s_{T}\in S_{T}]$ that has jointly CSCG entries, an $\underline{\bar{X}}$ as in (Equation 109) with $Q_{\underline{\bar{X}}}≻0$, and the auxiliary model (Equation 111) with $Q_{\underline{Z}}≻0$ is*

(112)
$$\begin{array}{cc} \; & I_{1}\left(A;\underline{Y}\right)=\log\det\left(I+Q_{\underline{\tilde{Z}}}^{-1}\;\tilde{H}Q_{\underline{\bar{X}}}{\tilde{H}}^{†}\right) \end{array}$$

*where (cf. (Equation 31))*

(113)
$$\begin{array}{cc} \tilde{H} & =E\left[\underline{Y}\;{\underline{\bar{X}}}^{†}\right]Q_{\underline{\bar{X}}}^{-1} \end{array}$$


(114)
$$\begin{array}{cc} Q_{\underline{\tilde{Z}}} & =Q_{\underline{Y}}-\tilde{H}Q_{\underline{\bar{X}}}{\tilde{H}}^{†}. \end{array}$$

*The expectation is with respect to the actual channel with joint distribution/density $p(a,\underline{y})$.*


**Proof.** See Appendix D. □

**Remark** **28.**
*Since $\underline{\bar{X}}$ is a function of A, the rate (Equation 112) can alternatively be derived by using $I\left(A;\underline{Y}\right)\geq I\left(\underline{\bar{X}};\underline{Y}\right)$ and applying the bound (Equation 30) with $\underline{X}$ replaced with $\underline{\bar{X}}$.*


**Remark** **29.**
*The estimate $\tilde{H}$ is the MMSE estimate of H:*

(115)
$$\begin{array}{c} \tilde{H}=\arg\min_{H}E\left[∥\underline{Y}-H\underline{\bar{X}}{∥}^{2}\right] \end{array}$$

*and $Q_{\tilde{\underline{Z}}}$ is the resulting covariance matrix of the error. To see this, expand (cf. (Equation 54))*

(116)
$$\begin{array}{cc} E\left[∥\underline{Y}-H\underline{\bar{X}}{∥}^{2}\right] & =E\left[∥\left(\underline{Y}-\tilde{H}\underline{\bar{X}}\right)+\left(\tilde{H}-H\right)\underline{\bar{X}}{∥}^{2}\right]\\ \; & =E\left[∥\underline{Y}-\tilde{H}\underline{\bar{X}}{∥}^{2}\right]+\mathrm{tr}\left(\left(\tilde{H}-H\right)Q_{\underline{\bar{X}}}{\left(\tilde{H}-H\right)}^{†}\right) \end{array}$$

*where the final step follows by the definition of $\tilde{H}$ in (Equation 113).*


**Remark** **30.**
*Suppose that H is an estimate other than (Equation 115). Generalizing (Equation 55), if $Q_{\underline{Y}}≻Q_{\underline{\bar{Z}}}$ we may choose*

(117)
$$\begin{array}{c} Q_{\underline{Z}}/s={\left(HQ_{\underline{\bar{X}}}H^{†}\right)}^{1/2}{\left(Q_{\underline{Y}}-Q_{\underline{\bar{Z}}}\right)}^{-1/2}Q_{\underline{\bar{Z}}}{\left(Q_{\underline{Y}}-Q_{\underline{\bar{Z}}}\right)}^{-1/2}{\left(HQ_{\underline{\bar{X}}}H^{†}\right)}^{1/2} \end{array}$$

*where*

(118)
$$\begin{array}{cc} Q_{\underline{\bar{Z}}} & =E\left[\left(\underline{Y}-H\underline{\bar{X}}\right){\left(\underline{Y}-H\underline{\bar{X}}\right)}^{†}\right]. \end{array}$$

*Appendix D shows that (Equation 102) then simplifies to (cf. (Equation 56))*

(119)
$$\begin{array}{c} I_{s}\left(A;\underline{Y}\right)=\log\det\left(Q_{\underline{\bar{Z}}}^{-1}Q_{\underline{Y}}\right). \end{array}$$



**Remark** **31.**
*The GMI (Equation 112) does not depend on the scaling of $\underline{\bar{X}}$ since this is absorbed in $\tilde{H}$. For example, one can choose the weighting matrices in (Equation 109) so that $E\left[∥\underline{\bar{X}}{∥}^{2}\right]=P$.*


### 4.5. Optimal Codebooks for CSCG Forward Models

The following Lemma maximizes the GMI for scalar channels and *A* with CSCG entries without requiring *A* to have the form (Equation 89). Nevertheless, this form is optimal, and we refer to ([10], page 2013) and Section 6.4 for similar results. In the following, let $U\left(s_{T}\right)∼\mathrm{CN}\left(0,1\right)$ for all $s_{T}$.

**Lemma** **3.**
*The maximum GMI (Equation 102) for the channel p(y|a), an adaptive symbol A with jointly CSCG entries, the forward model (Equation 111), and with fixed $P\left(s_{T}\right)=E\left[|X\left(s_{T}\right){|}^{2}\right]$ is*

(120)
$$\begin{array}{c} I_{1}\left(A;Y\right)=\log\left(1+\frac{\tilde{P}}{E\left[{\left|Y\right|}^{2}\right]-\tilde{P}}\right) \end{array}$$

*where, writing $X\left(s_{T}\right)=\sqrt{P(s_{T})}\;U\left(s_{T}\right)$ for all $s_{T}$, we have*

(121)
$$\begin{array}{cc} \tilde{P} & =E{\left[\;\left|E\left[\left.YU{\left(S_{T}\right)}^{*}\right|S_{T}\right]\right|\;\right]}^{2}. \end{array}$$

*This GMI is achieved by choosing fully-correlated symbols:*

(122)
$$\begin{array}{c} X\left(s_{T}\right)=\sqrt{P(s_{T})}\;e^{j\phi (s_{T})}\;U \end{array}$$

*and $\bar{X}=c\;U$ for some non-zero constant c and a common U∼CN(0,1), and where*

(123)
$$\begin{array}{c} \phi \left(s_{T}\right)=-\arg\left(E\left[\left.Y\;U{\left(s_{T}\right)}^{*}\right|S_{T}=s_{T}\right]\right). \end{array}$$



**Proof.** See Appendix E. □

**Remark** **32.**
*The expression (Equation 121) is based on (Equation 425) in Appendix E and can alternatively be written as $\tilde{P}=|\tilde{h}{|}^{2}\bar{P}$ where*

$$\begin{array}{cc} \tilde{h} & =E\left[Y{\bar{X}}^{*}\right]/\bar{P}. \end{array}$$



**Remark** **33.**
*The power levels $P(s_{T})$ may be optimized, usually under a constraint such as $E\left[P(S_{T})\right]\leq P$.*


**Remark** **34.**
*By the Cauchy-Schwarz inequality, we have*

$$\begin{array}{c} E{\left[\;\left|E\left[\left.YU{\left(S_{T}\right)}^{*}\right|S_{T}\right]\right|\;\right]}^{2}\leq E\left[{\left|Y\right|}^{2}\right]. \end{array}$$

*Furthermore, equality holds if and only if $\left|YU{\left(s_{T}\right)}^{*}\right|$ is a constant for each $s_{T}$, but this case is not interesting.*


### 4.6. Forward Model GMI for MIMO Channels

The following lemma generalizes Lemma 3 to MIMO channels without claiming a closed-form expression for the optimal GMI. Let $\underline{U}\left(s_{T}\right)∼\mathrm{CN}\left(\underline{0},I\right)$ for all $s_{T}$.

**Lemma** **4.**
*A GMI (Equation 102) for the channel $p(\underline{y}|a)$, an adaptive vector A with jointly CSCG entries, the auxiliary model (Equation 111), and with fixed $Q_{\underline{X}\left(s_{T}\right)}$ is given by (Equation 112) that we write as*

(124)
$$\begin{array}{c} I_{1}\left(A;\underline{Y}\right)=\log\left(\frac{\det Q_{\underline{Y}}}{\det\left(Q_{\underline{Y}}-\tilde{D}\;{\tilde{D}}^{†}\right)}\right). \end{array}$$

*where for M×M unitary $V_{R}\left(s_{T}\right)$ we have*

(125)
$$\begin{array}{c} \tilde{D}=E\left[U_{T}\left(S_{T}\right)\;\Sigma \left(S_{T}\right)\;V_{R}{\left(S_{T}\right)}^{†}\right] \end{array}$$

*and $U_{T}\left(s_{T}\right)$ and $\Sigma (s_{T})$ are N×N unitary and N×M rectangular diagonal matrices, respectively, of the SVD*

(126)
$$\begin{array}{c} E\left[\left.\underline{Y}\;{\underline{U}\left(s_{T}\right)}^{†}\right|S_{T}=s_{T}\right]=U_{T}\left(s_{T}\right)\;\Sigma \left(s_{T}\right)\;V_{T}{\left(s_{T}\right)}^{†} \end{array}$$

*for all $s_{T}$, and the $V_{T}\left(s_{T}\right)$ are M×M unitary matrices. The GMI (Equation 124) is achieved by choosing the symbols (cf. (Equation 122) and (Equation 454) below):*

(127)
$$\begin{array}{c} \underline{X}\left(s_{T}\right)=Q_{\underline{X}\left(s_{T}\right)}^{1/2}\;V_{T}\left(s_{T}\right)\;\underline{U} \end{array}$$

*and $\underline{\bar{X}}=C\;\underline{U}$ for some invertible M×M matrix C and a common M-dimensional vector $\underline{U}∼\mathrm{CN}\left(\underline{0},I\right)$. One may maximize (Equation 124) over the unitary $V_{R}\left(s_{T}\right)$.*


**Proof.** See Appendix G. □

Using Lemma 4, the theory for MISO channels with N=1 is similar to the scalar case of Lemma 3; see Remark 35 below. However, optimizing the GMI is more difficult for N>1 because one must optimize over the unitary matrices $V_{R}\left(s_{T}\right)$ in (Equation 125); see Remark 36 below.

**Remark** **35.**
*Consider N=1 in which case one may set $U_{T}\left(s_{T}\right)=1$ and (Equation 126) is a 1×M vector where $\Sigma (s_{T})$ has as the only non-zero singular value*

(128)
$$\begin{array}{c} \sigma \left(s_{T}\right)=∥E\left[\left.Y\;\underline{U}{\left(s_{T}\right)}^{†}\right|S_{T}=s_{T}\right]∥={\left(\sum_{m=1}^{M}{\left|E\left[\left.Y\;U_{m}{\left(s_{T}\right)}^{*}\right|S_{T}=s_{T}\right]\right|}^{2}\right)}^{1/2}. \end{array}$$

*The absolute value of the scalar (Equation 125) is maximized by choosing $V_{R}\left(s_{T}\right)=I$ for all $s_{T}$ to obtain (cf. (Equation 121))*

(129)
$$\begin{array}{c} \tilde{D}\;{\tilde{D}}^{†}=E{\left[\sigma (S_{T})\right]}^{2}. \end{array}$$



**Remark** **36.**
*Consider M=1 in which case one may set $V_{T}\left(s_{T}\right)=1$ and (Equation 126) is a N×1 vector where $\Sigma (s_{T})$ has as the only non-zero singular value*

(130)
$$\begin{array}{c} \sigma \left(s_{T}\right)=∥E\left[\left.\underline{Y}\;U{\left(s_{T}\right)}^{†}\right|S_{T}=s_{T}\right]∥={\left(\sum_{n=1}^{N}{\left|E\left[\left.Y_{n}\;U{\left(s_{T}\right)}^{*}\right|S_{T}=s_{T}\right]\right|}^{2}\right)}^{1/2}. \end{array}$$

*We should now find the $V_{R}\left(s_{T}\right)=e^{j\phi_{R}\left(s_{T}\right)}$ that minimize the determinant in the denominator of (Equation 124) where (see (Equation 125))*

(131)
$$\begin{array}{c} \tilde{D}=E\left[{\underline{u}}_{T}\left(S_{T}\right)\;\sigma \left(S_{T}\right)\;e^{-j\phi_{R}\left(S_{T}\right)}\right] \end{array}$$

*and where each ${\underline{u}}_{T}\left(s_{T}\right)$ is one of the columns of the N×N unitary matrix $U_{T}\left(s_{T}\right)$.*


**Remark** **37.**
*Consider M=N and the product channel*

(132)
$$\begin{array}{c} p\left(\underline{y}|a\right)=\prod_{m=1}^{M}p\left(y_{m}\;|\;\left[x_{m}\left(s_{T}\right):s_{T}\in S_{T}\right]\right) \end{array}$$

*where $x_{m}\left(s_{T}\right)$ is the m’th entry of $\underline{x}\left(s_{T}\right)$. We choose $Q_{\underline{X}\left(s_{T}\right)}$ as diagonal with diagonal entries $\sqrt{P_{m}\left(s_{T}\right)}$, m=1,…,M. Also choosing $V_{R}\left(s_{T}\right)=I$ makes the matrix $\tilde{D}\;{\tilde{D}}^{†}$ diagonal with the diagonal entries (cf. (Equation 121) where M=N=1)*

(133)
$$\begin{array}{c} {\left(\sum_{s_{T}}P_{S_{T}}\left(s_{T}\right)\left|E\left[\left.Y_{m}U_{m}{\left(s_{T}\right)}^{*}\right|S_{T}=s_{T}\right]\right|\right)}^{2} \end{array}$$

*for m=1,…,M. The GMI (Equation 124) is thus (cf. (Equation 120))*

(134)
$$\begin{array}{c} I_{1}\left(A;\underline{Y}\right)=\sum_{m=1}^{M}\log\left(\frac{E\left[|Y_{m}{|}^{2}\right]}{E\left[|Y_{m}{|}^{2}\right]-E{\left[\left|E\left[\left.Y_{m}U_{m}{\left(S_{T}\right)}^{*}\right|S_{T}\right]\right|\right]}^{2}}\right). \end{array}$$



**Remark** **38.**
*For general $p(\underline{y}|a)$, one might wish to choose diagonal $Q_{\underline{X}\left(s_{T}\right)}$ and a product model*

$$\begin{array}{c} q\left(\underline{y}|a\right)=\prod_{m=1}^{M}q_{m}\left(y_{m}|{\bar{x}}_{m}\right) \end{array}$$

*where the $q_{m}\left(.\right)$ are scalar AWGN channels*

$$\begin{array}{c} q_{m}\left(y|x\right)=\frac{1}{\pi \sigma_{m}^{2}}\exp\left(-|y-h_{m}{\;x|}^{2}/\sigma_{m}^{2}\right) \end{array}$$

*with possibly different $h_{m}$ and $\sigma_{m}^{2}$ for each m. Consider also*

$$\begin{array}{c} {\bar{X}}_{m}=\sum_{s_{T}}w_{m}\left(s_{T}\right)X_{m}\left(s_{T}\right) \end{array}$$

*for some complex weights $w_{m}\left(s_{T}\right)$, i.e., ${\bar{X}}_{m}$ is a weighted sum of entries from the list $\left[X_{m}\left(s_{T}\right):s_{T}\in S_{T}\right]$. The maximum GMI is now the same as (Equation 134) but without requiring the actual channel to have the form (Equation 132).*


**Remark** **39.**
*If the actual channel is $\underline{Y}=H\;\underline{X}+\underline{Z}$ then*

(135)
$$\begin{array}{c} E\left[\underline{Y}\;\underline{U}{\left(s_{T}\right)}^{†}|S_{T}=s_{T}\right]=E\left[H\;\underline{X}\left(s_{T}\right)\;\underline{U}{\left(s_{T}\right)}^{†}|S_{T}=s_{T}\right]=E\left[H|S_{T}=s_{T}\right]\;Q_{\underline{X}\left(s_{T}\right)}^{1/2} \end{array}$$

*where the final step follows because $\underline{U}\left(S_{T}\right)-S_{T}-H$ forms a Markov chain. The expression (Equation 135) is useful because it separates the effects of the channel and the transmitter.*


**Remark** **40.**
*Combining Remarks 37 and 39, suppose the actual channel is $\underline{Y}=H\;\underline{X}+\underline{Z}$ with M=N and where H is diagonal with diagonal entries $H_{m}$, m=1,…,M. The GMI (Equation 124) is then (cf. (Equation 134))*

(136)
$$\begin{array}{c} I_{1}\left(A;\underline{Y}\right)=\sum_{m=1}^{M}\log\left(\frac{E\left[|Y_{m}{|}^{2}\right]}{E\left[|Y_{m}{|}^{2}\right]-E{\left[\left|E\left[\left.H_{m}\sqrt{P_{m}\left(S_{T}\right)}\right|S_{T}\right]\right|\right]}^{2}}\right) \end{array}$$

*where $E\left[|Y_{m}{|}^{2}\right]=1+E\left[|H_{m}{|}^{2}P_{m}\left(S_{T}\right)\right]$.*


## 5. Channels with CSIR and CSIT

Shannon’s model includes CSIR [11]. The FDG is shown in Figure 3 where there is a hidden state $S_{H}$, the CSIR $S_{R}$ and CSIT $S_{T}$ are functions of $S_{H}$, and the receiver sees the channel outputs
(137)$$\begin{array}{c} \left[Y_{i},S_{Ri}\right]=\left[f\left(X_{i},S_{Hi},Z_{i}\right),S_{Ri}\right] \end{array}$$
for some function f(.) and i=1,2,…,n. (By defining $S_{H}=\left[S_{H1},Z_{H}\right]$ and calling $S_{H1}$ the hidden channel state we can include the case where $S_{R}$ and $S_{T}$ are noisy functions of $S_{H1}$.) As before, *M*, $S_{H}^{n}$, $Z^{n}$ are mutually statistically independent, and $S_{H}^{n}$ and $Z^{n}$ are i.i.d. strings of random variables with the same distributions as $S_{T}$ and *Z*, respectively. Observe that we have changed the notation by writing *Y* for only part of the channel output. The new *Y* (without the $S_{R}$) is usually called the “channel output”.

### 5.1. Capacity and GMI

We begin with scalar channels for which (Equation 90) is
(138)$$\begin{array}{c} C=\max_{A}\;I\left(A;Y,S_{R}\right)=\max_{A}\;I\left(A;Y|S_{R}\right) \end{array}$$
where *A* and $S_{R}$ are independent.

*Reverse Model:* The expression (Equation 108) with the adaptive symbol (Equation 88) is
(139)$$\begin{array}{c} I_{1}\left(A;Y,S_{R}\right)=E\left[-\log\mathrm{Var}\left[U|Y,S_{R}\right]\right]. \end{array}$$

*Forward Model:* Consider the expansion
(140)$$\begin{array}{c} I_{1}\left(A;Y|S_{R}\right)=\int_{S_{R}}p\left(s_{R}\right)\;I_{1}\left(A;Y|S_{R}=s_{R}\right)\;ds_{R} \end{array}$$
where $I_{1}\left(A;Y|S_{R}=s_{R}\right)$ is the GMI (Equation 102) with all densities conditioned on $S_{R}=s_{R}$. We choose the forward model
(141)$$\begin{array}{c} q\left(y|a,s_{R}\right)=\frac{1}{\pi \sigma {\left(s_{R}\right)}^{2}}\exp\left(-\frac{|y-h\left(s_{R}\right)\;\bar{x}\left(s_{R}\right){|}^{2}}{\sigma {\left(s_{R}\right)}^{2}}\right). \end{array}$$
where similar to (Equation 109) we define
(142)$$\begin{array}{c} \bar{X}\left(s_{R}\right)=\sum_{s_{T}}w\left(s_{T},s_{R}\right)\;X\left(s_{T}\right) \end{array}$$
for complex weights $w(s_{T},s_{R})$, i.e., $\bar{X}\left(s_{R}\right)$ is a weighted sum of entries from the list $A=[X\left(s_{T}\right):s_{T}\in S_{T}]$. We have the following straightforward generalization of Lemma 3.

**Theorem** **1.**
*The maximum GMI (Equation 140) for the channel $p(y|a,s_{R})$, an adaptive symbol A with jointly CSCG entries, the model (Equation 141), and with fixed $P\left(s_{T}\right)=E\left[|X\left(s_{T}\right){|}^{2}\right]$ is*

(143)
$$\begin{array}{c} I_{1}\left(A;Y|S_{R}\right)=E\left[\log\left(1+\frac{\tilde{P}\left(S_{R}\right)}{E\left[{\left|Y\right|}^{2}|S_{R}\right]-\tilde{P}\left(S_{R}\right)}\right)\right] \end{array}$$

*where for all $s_{R}\in S_{R}$ we have*

(144)
$$\begin{array}{cc} \tilde{P}\left(s_{R}\right) & =E{\left[\;\left|E\left[\left.YU{\left(S_{T}\right)}^{*}\right|S_{T},S_{R}=s_{R}\right]\right|\;\right]}^{2}. \end{array}$$



**Remark** **41.**
*To establish Theorem 1, the receiver may choose $\bar{X}=\sqrt{P}\;U$ to be independent of $s_{R}$. Alternatively, the receiver may choose $\bar{X}\left(s_{R}\right)=\sqrt{E\left[{\left|X\right|}^{2}|S_{R}=s_{R}\right]}\;U$. Both choices give the same GMI since the expectation in (Equation 144) does not depend on the scaling of $\bar{X}$; see Remark 31.*


**Remark** **42.**
*The partition idea of Lemmas 1 and 5 carries over to Theorem 1. We may generalize (Equation 143) as*

(145)
$$\begin{array}{cc} I_{1}\left(A;Y|S_{R}\right)=\int_{S_{R}}p\left(s_{R}\right) & \sum_{k=1}^{K}\mathrm{Pr}\left[E_{k}|S_{R}=s_{R}\right]\\ \; & \left[\log\left(1+\frac{|h_{k}\left(s_{R}\right){|}^{2}P}{\sigma_{k}^{2}\left(s_{R}\right)}\right)+\frac{E\left[\left.{\left|Y\right|}^{2}\right|E_{k},S_{R}=s_{R}\right]}{\sigma_{k}^{2}\left(s_{R}\right)+{\left|h_{k}\left(s_{R}\right)\right|}^{2}P}\right.\\ \; & \left.\;-\frac{E\left[\left.|Y-h_{k}\left(s_{R}\right)\sqrt{P}{\;U|}^{2}\right|E_{k},S_{R}=s_{R}\right]}{\sigma_{k}^{2}\left(s_{R}\right)}\right]\;ds_{R} \end{array}$$

*where the $X(s_{T})$, $s_{T}\in S_{T}$, are given by (Equation 122) and the $h_{k}\left(s_{R}\right)$ and $\sigma_{k}^{2}\left(s_{R}\right)$, k=1,…,K, $s_{R}\in S_{R}$, can be optimized.*


**Remark** **43.**
*One is usually interested in the optimal power control policy $P(s_{T})$ under the constraint $E\left[P(S_{T})\right]\leq P$. Taking the derivative of (Equation 143) with respect to $\sqrt{P(s_{T})}$ and setting to zero we obtain*

(146)
$$\begin{array}{c} E\left[\frac{E\left[{\left|Y\right|}^{2}|S_{R}\right]\tilde{P}{\left(S_{R}\right)}^{\prime }-\tilde{P}\left(S_{R}\right)\;E{\left[{\left|Y\right|}^{2}|S_{R}\right]}^{\prime }}{E\left[{\left|Y\right|}^{2}|S_{R}\right]\left[E\left[{\left|Y\right|}^{2}|S_{R}\right]-\tilde{P}\left(S_{R}\right)\right]}\right]=2\lambda \sqrt{P(s_{T})}P_{S_{T}}\left(s_{T}\right) \end{array}$$

*where $\tilde{P}{\left(S_{R}\right)}^{\prime }$ and $E{\left[{\left|Y\right|}^{2}|S_{R}\right]}^{\prime }$ are derivatives with respect to $\sqrt{P(s_{T})}$. We use (Equation 146) below to derive power control policies.*


**Remark** **44.**
*A related model is a compound channel where $p(y|a,s_{R})$ is indexed by the parameter $s_{R}$ ([91], Chapter 4). The problem is to find the maximum worst-case reliable rate if the transmitter does not know $s_{R}$. Alternatively, the transmitter must send its message to all $|S_{R}|$ receivers indexed by $s_{R}\in S_{R}$. A compound channel may thus be interpreted as a broadcast channel with a common message.*


### 5.2. CSIT@ R

An interesting specialization of Shannon’s model is when the receiver knows $S_{T}$ and can determine $X(S_{T})$. We refer to this scenario as CSIT@R. The model was considered in ([10], Section II) when $S_{T}$ is a function of $S_{R}$. More generally, suppose $S_{T}$ is a function of $\left[Y,S_{R}\right]$. The capacity (Equation 138) then simplifies to (see ([10], Proposition 1))
(147)$$\begin{array}{cc} C & \overset{\left(a\right)}{=}\max_{A}\;I\left(A;Y,S_{T}|S_{R}\right)\\ \; & \overset{\left(b\right)}{=}\max_{A}\;I\left(X;Y|S_{R},S_{T}\right)\\ \; & \overset{\left(c\right)}{=}\sum_{s_{T}}P_{S_{T}}\left(s_{T}\right)\left[\max_{X(s_{T})}\;I\left(X\left(s_{T}\right);Y|S_{R},S_{T}=s_{T}\right)\right] \end{array}$$
where step (a) follows because $S_{T}$ is a function of $\left[Y,S_{R}\right]$; step (b) follows because *A* and $\left(S_{R},S_{T}\right)$ are independent, *X* is a function of $\left[A,S_{T}\right]$, and $A-[S_{T},X]-Y$ forms a Markov chain; and step (c) follows because one may optimize $X(s_{T})$ separately for each $s_{T}\in S_{T}$.

As discussed in [10], a practical motivation for this model is when the CSIT is based on error-free feedback from the receiver to the transmitter. In this case, where $S_{T}$ is a function of $S_{R}$, the expression (Equation 144) becomes
(148)$$\begin{array}{cc} \tilde{P}\left(s_{R}\right) & ={\left|E\left[\left.YU{\left(s_{T}\right)}^{*}\;\right|S_{R}=s_{R}\;\right]\right|}^{2}. \end{array}$$

**Remark** **45.**
*The insight that one can replace adaptive symbols A with channel inputs X when X is a function of A and past Y appeared for two-way channels in ([9], Section 4.2.3) and networks in ([22], Section V.A), ([72], Section IV.F).*


### 5.3. MIMO Channels and *K*-Partitions

We consider generalizations to MIMO channels and *K*-partitions as in Section 3.2.

*MIMO Channels:* Consider the average GMI
(149)$$\begin{array}{c} I_{1}\left(A;\underline{Y}|S_{R}\right)=\int_{S_{R}}p\left(s_{R}\right)I_{1}\left(A;\underline{Y}|S_{R}=s_{R}\right)\;ds_{R} \end{array}$$
and choose the parameters (Equation 113) and (Equation 114) for the event $S_{R}=s_{R}$. We have
(150)$$\begin{array}{cc} \tilde{H}\left(s_{R}\right) & =E\left[\left.\underline{Y}\;{\underline{\bar{X}}}^{†}\right|S_{R}=s_{R}\right]E{\left[\left.\underline{\bar{X}}\;{\underline{\bar{X}}}^{†}\right|S_{R}=s_{R}\right]}^{-1} \end{array}$$
(151)$$\begin{array}{cc} Q_{\underline{\tilde{Z}}}\left(s_{R}\right) & =E\left[\left.\underline{Y}\;{\underline{Y}}^{†}\right|S_{R}=s_{R}\right]-\tilde{H}\left(s_{R}\right)E\left[\left.\underline{\bar{X}}\;{\underline{\bar{X}}}^{†}\right|S_{R}=s_{R}\right]\tilde{H}{\left(s_{R}\right)}^{†} \end{array}$$
and the GMI (Equation 149) is (cf. (Equation 60) and (Equation 112))
(152)$$\begin{array}{c} I_{1}\left(A;\underline{Y}|S_{R}\right)=E\left[\log\det\left(I+Q_{\underline{\tilde{Z}}}{\left(S_{R}\right)}^{-1}\;\tilde{H}\left(S_{R}\right)Q_{\underline{\bar{X}}}\tilde{H}{\left(S_{R}\right)}^{†}\right)\right]. \end{array}$$
*K-Partitions:* Let $\left\{{\underline{Y}}_{k}:k=1,\ldots ,K\right\}$ be a *K*-partition of $\underline{Y}$ and define the events $E_{k}=\left\{\underline{Y}\in {\underline{Y}}_{k}\right\}$ for k=1,…,K. As in Remark 13, *K*-partitioning formally includes (Equation 149) as a special case by including $S_{R}$ as part of the receiver’s “overall” channel output $\underline{\tilde{Y}}=\left[\underline{Y},S_{R}\right]$. The following lemma generalizes Lemma 1.

**Lemma** **5.**
*A GMI with s=1 for the channel $p(\underline{y}|a)$ is*

(153)
$$\begin{array}{cc} I_{1}\left(A;\underline{Y}\right)=\sum_{k=1}^{K}\mathrm{Pr}\left[E_{k}\right] & \left\{\log\det\left(I+Q_{{\underline{Z}}_{k}}^{-1}\;H_{k}Q_{\underline{\bar{X}}}H_{k}^{†}\right)\right.\\ \; & +E\left[\left.{\underline{Y}}^{†}{\left(Q_{{\underline{Z}}_{k}}+H_{k}Q_{\underline{\bar{X}}}H_{k}^{†}\right)}^{-1}\underline{Y}\;\right|E_{k}\right]\\ \; & \left.-E\left[\left.{\left(\underline{Y}-H_{k}\;\underline{\bar{X}}\right)}^{†}Q_{{\underline{Z}}_{k}}^{-1}\left(\underline{Y}-H_{k}\;\underline{\bar{X}}\right)\right|E_{k}\right]\right\} \end{array}$$

*where the $H_{k}$ and $Q_{{\underline{Z}}_{k}}$, k=1,…,K, can be optimized.*


**Remark** **46.**
*For scalars the GMI (Equation 153) is*

(154)
$$\begin{array}{cc} I_{1}\left(A;Y\right)=\sum_{k=1}^{K}\mathrm{Pr}\left[E_{k}\right] & \left[\log\left(1+\frac{|h_{k}{|}^{2}\bar{P}}{\sigma_{k}^{2}}\right)+\frac{E\left[{\left|Y\right|}^{2}|E_{k}\right]}{\sigma_{k}^{2}+{\left|h_{k}\right|}^{2}\bar{P}}-\frac{E\left[|Y-h_{k}\bar{X}{|}^{2}|E_{k}\right]}{\sigma_{k}^{2}}\right] \end{array}$$

*which is the same as (Equation 64) except that $\bar{X}$, $\bar{P}$ replace X,P. If we follow (Equation 66) and (Equation 67) then (Equation 154) becomes (Equation 68) but with*

$$\begin{array}{c} h_{k}=E\left[\left.Y{\bar{X}}^{*}\right|E_{k}\right]/P_{k},\;P_{k}=E\left[\left.{\left|\bar{X}\right|}^{2}\right|E_{k}\right]. \end{array}$$



**Remark** **47.**
*Consider Remark 14 and choose K=2, $h_{1}=0$, $\sigma_{1}^{2}=1$. The GMI (Equation 154) then has only the k=2 term, and it again remains to select $h_{2}$, $\sigma_{2}^{2}$, and $t_{R}$.*


**Remark** **48.**
*If we define*

(155)
$$\begin{array}{c} Q_{\underline{\bar{X}}}^{\left(k\right)}=E\left[\left.\underline{\bar{X}}\;{\underline{\bar{X}}}^{†}\right|E_{k}\right],\;Q_{\underline{Y}}^{\left(k\right)}=E\left[\left.\underline{Y}\;{\underline{Y}}^{†}\right|E_{k}\right] \end{array}$$

*and choose the LMMSE auxiliary models with*

(156)
$$\begin{array}{cc} H_{k} & =E\left[\left.\underline{Y}\;{\underline{\bar{X}}}^{†}\right|E_{k}\right]{\left(Q_{\underline{\bar{X}}}^{\left(k\right)}\right)}^{-1} \end{array}$$


(157)
$$\begin{array}{cc} Q_{{\underline{Z}}_{k}} & =Q_{\underline{Y}}^{\left(k\right)}-H_{k}Q_{\underline{\bar{X}}}^{\left(k\right)}H_{k}^{†} \end{array}$$

*for k=1,…,K then the expression (Equation 153) is (cf. (Equation 68))*

$$\begin{array}{cc} I_{1}\left(A;\underline{Y}\right)=\sum_{k=1}^{K}\mathrm{Pr}\left[E_{k}\right] & \left[\log\det\left(I+Q_{{\underline{Z}}_{k}}^{-1}\;H_{k}Q_{\underline{\bar{X}}}H_{k}^{†}\right)\right. \end{array}$$


(158)
$$\begin{array}{c} \left.-\mathrm{tr}\left({\left(Q_{\underline{Y}}^{\left(k\right)}+H_{k}D_{\underline{\bar{X}}}^{\left(k\right)}H_{k}^{†}\right)}^{-1}H_{k}D_{\underline{\bar{X}}}^{\left(k\right)}H_{k}^{†}\right)\right] \end{array}$$

*where $D_{\underline{\bar{X}}}^{\left(k\right)}=Q_{\underline{\bar{X}}}-Q_{\underline{\bar{X}}}^{\left(k\right)}$.*


**Remark** **49.**
*We may proceed as in Remark 18 and consider large K. These steps are given in Appendix F.*


## 6. Fading Channels with AWGN

This section treats scalar, complex-alphabet, AWGN channels with CSIR for which the channel output is
(159)$$\begin{array}{c} \left[Y,S_{R}\right]=\left[HX+Z,S_{R}\right] \end{array}$$
where H,A,Z are mutually independent, $E\left[{\left|H\right|}^{2}\right]=1$, and Z∼CN(0,1). The capacity under the power constraint $E\left[{\left|X\right|}^{2}\right]\leq P$ is (cf. (Equation 138))
(160)$$\begin{array}{c} C\left(P\right)=\max_{A:\;E\left[{\left|X\right|}^{2}\right]\leq P}I\left(A;Y|S_{R}\right). \end{array}$$
However, the optimization in (Equation 160) is often intractable, and we desire expressions with log(1+SNR) terms to gain insight. We develop three such expressions: an upper bound and two lower bounds. It will be convenient to write $G={\left|H\right|}^{2}$.

*Capacity Upper Bound:* We state this bound as a lemma since we use it to prove Proposition 2 below.

**Lemma** **6.**
*The capacity (Equation 160) is upper bounded as*

(161)
$$\begin{array}{c} C\left(P\right)\leq \max\;E\left[\log\left(1+GP(S_{T})\right)\right] \end{array}$$

*where the maximization is over $P(S_{T})$ with $E\left[P(S_{T})\right]=P$.*


**Proof.** Consider the steps
(162)$$\begin{array}{cc} I(A;Y|S_{R}) & \leq I(A;Y,S_{T},H|S_{R})\\ \; & \overset{\left(a\right)}{=}I\left(A;Y|S_{R},S_{T},H\right)\\ \; & =h\left(Y|S_{R},S_{T},H\right)-h\left(Z\right)\\ \; & \overset{\left(b\right)}{\leq }E\left[\log\mathrm{Var}\left[Y|S_{R},S_{T},H\right]\right] \end{array}$$
where step (a) is because *A* and $\left[S_{R},S_{T},H\right]$ are independent, and step (b) follows by the entropy bound
(163)$$\begin{array}{c} h\left(Y|B=b\right)\leq \log\left(\pi e\;\mathrm{Var}\left[Y|B=b\right]\right) \end{array}$$
which we applied with $B=[S_{R},S_{T},H]$. Finally, we compute $\mathrm{Var}\left[Y|S_{R},S_{T},H\right]=1+GP\left(S_{T}\right)$. □

*Reverse Model GMI:* Consider the adaptive symbol (Equation 88) and the GMI (Equation 139). We expand the variances in (Equation 139) as
$$\begin{array}{c} \mathrm{Var}\left[U|Y=y,S_{R}=s_{R}\right]=E\left[{\left|U\right|}^{2}|Y=y,S_{R}=s_{R}\right]-|E\left[U|Y=y,S_{R}=s_{R}\right]{|}^{2}. \end{array}$$

Appendix C shows that one may write
(164)$$\begin{array}{c} E\left[U|Y=y,S_{R}=s_{R}\right]=\int_{C\times S_{T}}p\left(h,s_{T}|y,s_{R}\right)\;\frac{h\sqrt{P(s_{T})}e^{j\phi (s_{T})}y}{1+{\left|h\right|}^{2}P\left(s_{T}\right)}\;ds_{T}\;dh \end{array}$$
and
(165)$$\begin{array}{cc} E\left[{\left|U\right|}^{2}|Y=y,S_{R}=s_{R}\right]\; & =\int_{C\times S_{T}}p\left(h,s_{T}|y,s_{R}\right)\\ \; & \;\left(\frac{1}{1+{\left|h\right|}^{2}P\left(s_{T}\right)}+\frac{{\left|h\right|}^{2}P\left(s_{T}\right){\left|y\right|}^{2}}{{\left(1+{\left|h\right|}^{2}P\left(s_{T}\right)\right)}^{2}}\right)\;ds_{T}\;dh. \end{array}$$
We use the expressions (Equation 164) and (Equation 165) to compute achievable rates by numerical integration. For example, suppose that $S_{T}=0$ and $S_{R}=H$, i.e., we have full CSIR and no CSIT. The averaging density is then
$$\begin{array}{c} p\left(h,s_{T}|y,s_{R}\right)=\delta \left(h-s_{R}\right)\;\delta (s_{T}) \end{array}$$
and the variance simplifies to the capacity-achieving form
$$\begin{array}{c} \mathrm{Var}\left[U|Y=y,S_{R}=h\right]=\frac{1}{1+{\left|h\right|}^{2}P}. \end{array}$$
*Forward Model GMI:* A forward model GMI is given by Theorem 1 where
(166)$$\begin{array}{cc} \tilde{P}\left(s_{R}\right) & =E{\left[\;\left|E\left[\left.H\sqrt{P(S_{T})}\right|S_{T},S_{R}=s_{R}\right]\right|\;\right]}^{2} \end{array}$$
(167)$$\begin{array}{cc} E\left[{\left|Y\right|}^{2}|S_{R}=s_{R}\right] & =1+E\left[GP\left(S_{T}\right)|S_{R}=s_{R}\right] \end{array}$$
so that (Equation 143) becomes
(168)$$\begin{array}{c} I_{1}\left(A;Y|S_{R}\right)=E\left[\log\left(1+\frac{\tilde{P}\left(S_{R}\right)}{1+E\left[GP\left(S_{T}\right)|S_{R}\right]-\tilde{P}\left(S_{R}\right)}\right)\right]. \end{array}$$

**Remark** **50.**
*Jensen’s inequality implies that the denominator in (Equation 168) is greater than or equal to*

(169)
$$\begin{array}{c} 1+\mathrm{Var}\left[\left.\sqrt{GP(S_{T})}\;\right|S_{R}\right]. \end{array}$$

*Equality requires that for all $S_{R}=s_{R}$ we have*

(170)
$$\begin{array}{c} \tilde{P}\left(s_{R}\right)=E{\left[\left.\sqrt{GP(S_{T})}\right|S_{R}=s_{R}\right]}^{2} \end{array}$$

*which is valid if H is a function of $\left[S_{R},S_{T}\right]$, for example. However, if there is channel uncertainty after conditioning on $\left[S_{R},S_{T}\right]$ then $\tilde{P}\left(s_{R}\right)$ is usually smaller than the RHS of (Equation 170).*


**Remark** **51.**
*Consider $S_{R}=H$ or $S_{R}=H\sqrt{P(S_{T})}$. For both cases, H is a function of $\left[S_{R},S_{T}\right]$ and the denominator in (Equation 168) is the variance (Equation 169). In fact, for $S_{R}=H\sqrt{P(S_{T})}$, the expression (Equation 169) takes on the minimal value 1. This CSIR is thus the best possible; see Proposition 2.*


**Remark** **52.**
*For MIMO channels we replace (Equation 159) with*

(171)
$$\begin{array}{c} \left[\underline{Y},S_{R}\right]=\left[H\underline{X}+\underline{Z},S_{R}\right] \end{array}$$

*where $H,A,\underline{Z}$ are mutually independent and $\underline{Z}∼\mathrm{CN}\left(\underline{0},I\right)$. One usually considers the constraint $E\left[∥\underline{X}{∥}^{2}\right]\leq P$.*


**Remark** **53.**
*The model (Equation 171) includes block fading. For example, choosing M=N and H=HI gives scalar block fading. Moreover, the capacity per symbol without in-block feedback is the same as for the M=N=1 case except that P is replaced with P/M; see [11] and Section 9.*


### 6.1. CSIR and CSIT Models

We study two classes of CSIR, as shown in Table 1. The first class has full (or “perfect”) CSIR, by which we mean either $S_{R}=H$ or $S_{R}=H\sqrt{P(S_{T})}$. The motivation for studying the latter case is that it models block fading channels with long blocks where the receiver estimates $H\sqrt{P(S_{T})}$ using pilot symbols, and the number of pilot symbols is much smaller than the block length [10]. Moreover, one achieves the upper bound (Equation 161), see Proposition 2 below.

We coarsely categorize the CSIT as follows:Full CSIT: $S_{T}=H$;CSIT@R: $S_{T}=q_{u}\left(G\right)$ where $q_{u}\left(.\right)$ is the quantizer of Section 2.9 with B=0,1,∞;Partial CSIT: $S_{T}$ is not known exactly at the receiver.
The capacity of the CSIT@R models is given by log(1+SNR) expressions [10,92]; see also [93]. The partial CSIT model is interesting because achieving capacity generally requires adaptive codewords and closed-form capacity expressions are unavailable. The GMI lower bound of Theorem 1 and Remark 42 and the capacity upper bound of Lemma 6 serve as benchmarks.

The partial CSIR models have $S_{R}$ being a lossy function of *H*. For example, a common model is based on LMMSE channel estimation with
(172)$$\begin{array}{c} H=\sqrt{\bar{\epsilon }}\;S_{R}+\sqrt{\epsilon }\;Z_{R} \end{array}$$
where 0≤ϵ≤1 and $S_{R},Z_{R}$ are uncorrelated. The CSIT is categorized as above, except that we consider $S_{T}=f_{T}\left(S_{R}\right)$ for some function $f_{T}\left(.\right)$ rather than $S_{T}=q_{u}\left(G\right)$.

To illustrate the theory, we study two types of fading: one with discrete *H* and one with continuous *H*, namely
Section 7: on-off fading with $P_{H}\left(0\right)=P_{H}\left(\sqrt{2}\right)=1/2$;Section 8: Rayleigh fading with H∼CN(0,1).
For on-off fading we have $p\left(g\right)=\frac{1}{2}\delta \left(g\right)+\frac{1}{2}\delta \left(g-2\right)$ and for Rayleigh fading we have $p\left(g\right)=e^{-g}\cdot 1\left(g\geq 0\right)$.

**Remark** **54.**
*For channels with partial CSIR, we will study the GMI for partitions with K=1 and K=2. The full CSIT model has received relatively little attention in the literature, perhaps because CSIR is usually more accurate than CSIT ([5], Section 4.2.3).*


### 6.2. No CSIR, No CSIT

Without CSIR or CSIT, the channel is a classic memoryless channel [94] for which the capacity (Equation 160) becomes the usual expression with $S_{R}=0$ and A=X. For CSCG *X* and $U=X/E\left[{\left|X\right|}^{2}\right]$, the reverse and forward model GMIs (Equation 139) and (Equation 168) are the respective
(173)$$\begin{array}{cc} I_{1}\left(X;Y\right) & =E\left[-\log\mathrm{Var}\left[U|Y\right]\right] \end{array}$$
(174)$$\begin{array}{cc} I_{1}\left(X;Y\right) & =\log\left(1+\frac{P\;|E\left[H\right]{|}^{2}}{1+P\;\mathrm{Var}\left[H\right]}\right). \end{array}$$

For example, the forward model GMI is zero if $E\left[H\right]=0$.

### 6.3. Full CSIR, CSIT@ R

Consider the full CSIR models with $S_{R}=H$ and CSIT@R. The capacity is given by log(1+SNR) expressions that we review.

First, the capacity with B=0 (no CSIT) is
(175)$$\begin{array}{c} C\left(P\right)=E\left[\log\left(1+G\;P\right)\right]=\int_{0}^{\infty }p\left(g\right)\log\left(1+gP\right)\;dg. \end{array}$$
The wideband derivatives are (see (Equation 37))
(176)$$\begin{array}{c} C^{\prime }\left(0\right)=E\left[G\right]=1,\;C^{\prime \prime }\left(0\right)=-E\left[G^{2}\right] \end{array}$$
so that the wideband values (Equation 37) are (see ([73], Theorem 13))
(177)$$\begin{array}{c} {\left.\frac{E_{b}}{N_{0}}\right|}_{\min}=\log2,\;S=\frac{2}{E\left[G^{2}\right]}. \end{array}$$
The minimal $E_{b}/N_{0}$ is the same as without fading, namely −1.59 dB. However, Jensen’s inequality gives $E\left[G^{2}\right]\geq E{\left[G\right]}^{2}=1$ with equality if and only if G=1. Thus, fading reduces the capacity slope *S*.

More generally, the capacity with full CSIR and $S_{T}=q_{u}\left(G\right)$ is (see [10])
(178)$$\begin{array}{cc} C(P) & =\max_{P\left(S_{T}\right):\;E\left[P(S_{T})\right]\leq P}E\left[\log\left(1+G\;P(S_{T})\right)\right]\\ \; & =\max_{P\left(S_{T}\right):\;E\left[P(S_{T})\right]\leq P}\int_{0}^{\infty }p\left(g,s_{T}\right)\log\left(1+gP(s_{T})\right)\;dg\;ds_{T}. \end{array}$$
To optimize the power levels $P(s_{T})$, consider the Lagrangian
(179)$$\begin{array}{c} E\left[\log\left(1+GP(S_{T})\right)\right]+\lambda \left(P-E\left[P(S_{T})\right]\right) \end{array}$$
where λ≥0 is a Lagrange multiplier. Taking the derivative with respect to $P(s_{T})$, we have
(180)$$\begin{array}{c} \lambda =E\left[\left.\frac{G}{1+GP(s_{T})}\right|S_{T}=s_{T}\right]=\int_{0}^{\infty }p\left(g|s_{T}\right)\frac{g}{1+gP(s_{T})}dg \end{array}$$
as long as $P(s_{T})\geq 0$. If this equation cannot be satisfied, choose $P(s_{T})=0$. Finally, set λ so that $E\left[P(S_{T})\right]=P$.

For example, consider B=∞ and $S_{T}=G$. We then have $p\left(g|s_{T}\right)=\delta \left(g-s_{T}\right)$ and therefore
(181)$$\begin{array}{c} P\left(g\right)={\left(\frac{1}{\lambda }-\frac{1}{g}\right)}^{+} \end{array}$$
where λ is chosen so that $E\left[P(G)\right]=P$. The capacity (Equation 178) is then (see ([95], Equation (Equation 7)))
(182)$$\begin{array}{cc} C(P) & =\int_{\lambda }^{\infty }p\left(g\right)\log\left(g/\lambda \right)\;dg. \end{array}$$

Consider now the quantizer $q_{u}\left(.\right)$ of Section 2.9 with B=1. We have two equations for λ, namely
(183)$$\begin{array}{cc} \lambda & =\int_{0}^{\Delta }\frac{p(g)}{P_{S_{T}}\left(\Delta /2\right)}\cdot \frac{g}{1+gP(\Delta /2)}dg \end{array}$$
(184)$$\begin{array}{cc} \lambda & =\int_{\Delta }^{\infty }\frac{p(g)}{P_{S_{T}}\left(3\Delta /2\right)}\cdot \frac{g}{1+gP(3\Delta /2)}dg. \end{array}$$
Observe the following for (Equation 183) and (Equation 184):both P(Δ/2) and P(3Δ/2) decrease as λ increases;the maximal λ permitted by (Equation 183) is $E\left[G|G\leq \Delta \right]$ which is obtained with P(Δ/2)=0;the maximal λ permitted by (184) is $E\left[G|G\geq \Delta \right]$ which is obtained with P(3Δ/2)=0.
Thus, if $E\left[G|G\geq \Delta \right]>E\left[G|G\leq \Delta \right]$, then at *P* below some threshold, we have P(Δ/2)=0 and $P\left(3\Delta /2\right)=P/P_{S_{T}}\left(3\Delta /2\right)$. The capacity in nats per symbol at low power and for fixed Δ is thus
(185)$$\begin{array}{cc} C(P) & =\int_{\Delta }^{\infty }p\left(g\right)\log\left(1+gP(3\Delta /2)\right)dg\\ \; & \approx P\;E\left[G|G\geq \Delta \right]-\frac{P^{2}}{2P_{S_{T}}\left(3\Delta /2\right)}E\left[G^{2}|G\geq \Delta \right] \end{array}$$
where we used
$$\begin{array}{c} \log\left(1+x\right)\approx x-\frac{x^{2}}{2} \end{array}$$
for small *x*. The wideband values (Equation 37) are
(186)$$\begin{array}{cc} \; & {\left.\frac{E_{b}}{N_{0}}\right|}_{\min}=\frac{\log2}{E\left[G|G\geq \Delta \right]} \end{array}$$
(187)$$\begin{array}{cc} \; & S=\frac{2P_{S_{T}}\left(3\Delta /2\right)\;E{\left[G|G\geq \Delta \right]}^{2}}{E\left[G^{2}|G\geq \Delta \right]}. \end{array}$$
One can thus make the minimum $E_{b}/N_{0}$ approach −∞ if one can make $E\left[G|G\geq \Delta \right]$ as large as desired by increasing Δ.

**Remark** **55.**
*Consider the MIMO model (Equation 171) with $S_{R}=H$. Suppose the CSIT is $S_{T}=f_{T}\left(S_{R}\right)$ for some function $f_{T}\left(\cdot \right)$. The capacity (Equation 178) generalizes to*

(188)
$$\begin{array}{cc} C(P)\; & =\max_{\underline{X}\left(S_{T}\right):\;E\left[∥\underline{X}\left(S_{T}\right){∥}^{2}\right]\leq P}I\left(\underline{X};H\underline{X}+\underline{Z}|H,S_{T}\right)\\ \; & =\max_{Q\left(S_{T}\right):\;E\left[\mathrm{tr}\left(Q(S_{T})\right)\right]\leq P}E\left[\log\det\left(I+HQ\left(S_{T}\right)H^{†}\right)\right]. \end{array}$$



### 6.4. Full CSIR, Partial CSIT

Consider first the full CSIR $S_{R}=H\sqrt{P(S_{T})}$ and then the less informative $S_{R}=H$.

*$S_{R}=H\sqrt{P(S_{T})}$:* We have the following capacity result that implies this CSIR is the best possible since one can achieve the same rate as if the receiver sees both *H* and $S_{T}$; see the first step of (Equation 162). We could thus have classified this model as CSIT@R.

**Proposition** **2**(see ([10], Proposition 3))**.** *The capacity of the channel (Equation 159) with*
$S_{R}=H\sqrt{P(S_{T})}$ and general $S_{T}$
*is*
(189)$$\begin{array}{cc} C(P)\; & =\max_{P\left(S_{T}\right):\;E\left[P(S_{T})\right]\leq P}\int_{C}p\left(s_{R}\right)\log\left(1+|s_{R}{|}^{2}\right)\;ds_{R}\\ \; & =\max_{P\left(S_{T}\right):\;E\left[P(S_{T})\right]\leq P}E\left[\log\left(1+GP(S_{T})\right)\right]. \end{array}$$

**Proof.** Achievability follows by Theorem 1 with Remark 51. The converse is given by Lemma 6. □

**Remark** **56.**
*Proposition 2 gives an upper bound and (thus) a target rate when the receiver has partial CSIR. For example, we will use the K-partition idea of Lemma 1 (see also Remark 46) to approach the upper bound for large SNR.*


**Remark** **57.**
*Proposition 2 partially generalizes to block-fading channels; see Proposition 3 in Section 9.5.*


*$S_{R}=H$:* The capacity is (Equation 138) with
(190)$$\begin{array}{c} I\left(A;Y|H\right)=E\left[\log\frac{p(Y|A,H)}{p(Y|H)}\right] \end{array}$$
where $E\left[{\left|X\right|}^{2}\right]\leq P$ and where
(191)$$\begin{array}{cc} p(y|a,h)\; & =\int_{C}p\left(s_{T}|h\right)\;\frac{e^{-|y-h\;x\left(s_{T}\right){|}^{2}}}{\pi }\;ds_{T} \end{array}$$
and
(192)$$\begin{array}{cc} p(y|h) & =\int_{C}p\left(s_{T}|h\right)\left(\int_{A}p\left(a\right)p\left(y|a,h,s_{T}\right)\;da\right)ds_{T}\\ \; & =\int_{C}p\left(s_{T}|h\right)\left(\int_{C}p\left(x\left(s_{T}\right)\right)\frac{e^{-|y-h\;x\left(s_{T}\right){|}^{2}}}{\pi }\;dx\left(s_{T}\right)\right)ds_{T}. \end{array}$$
For example, if each entry $X(s_{T})$ of *A* is CSCG with variance $P(s_{T})$ then
(193)$$\begin{array}{c} p\left(y|h\right)=\int_{C}\;p\left(s_{T}|h\right)\frac{\exp\left(-\frac{{\left|y\right|}^{2}}{1+gP(s_{T})}\right)}{\pi (1+gP\left(s_{T}\right))}\;ds_{T}. \end{array}$$
In general, one can compute I(A;Y|H) numerically by using (Equation 190)–(Equation 192), but the calculations are hampered if the integrals in (Equation 191) and (Equation 192) do not simplify.

For the reverse model GMI (Equation 139), the averaging density in (Equation 164) and (Equation 165) is here
(194)$$\begin{array}{c} p\left(h,s_{T}|y,s_{R}\right)=\delta \left(h-s_{R}\right)\;\frac{p\left(s_{T}|h\right)\;p\left(y|h,s_{T}\right)}{p(y|h)}. \end{array}$$
We use numerical integration to compute the GMI.

To obtain more insight, we state the forward model rates of Theorem 1 and Remark 51 as a Corollary.

**Corollary** **1.**
*An achievable rate for the fading channels (Equation 159) with $S_{R}=H$ and partial CSIT is the forward model GMI*

(195)
$$\begin{array}{cc} I_{1}\left(A;Y|H\right) & =E\left[\log\left(1+\mathrm{SNR}(H)\right)\right] \end{array}$$

*where*

(196)
$$\begin{array}{c} \mathrm{SNR}\left(h\right)=\frac{{\left|h\right|}^{2}{\tilde{P}}_{T}\left(h\right)}{1+{\left|h\right|}^{2}\mathrm{Var}\left[\left.\sqrt{P(S_{T})}\right|H=h\right]} \end{array}$$

*and*

(197)
$$\begin{array}{c} {\tilde{P}}_{T}\left(h\right)=E{\left[\left.\sqrt{P(S_{T})}\right|H=h\right]}^{2}. \end{array}$$



**Remark** **58.**
*Jensen’s inequality gives*

(198)
$$\begin{array}{c} {\tilde{P}}_{T}\left(h\right)\leq E\left[P(S_{T})|H=h\right] \end{array}$$

*by the concavity of the square root. Equality holds if and only if $P(S_{T})$ is a constant given H=h.*


**Remark** **59.**
*Choosing $P(s_{T})=P$ for all $s_{T}$ in Corollary 1 gives ${\tilde{P}}_{T}\left(h\right)=P$ for all h and the rate (Equation 195) is the capacity (Equation 175) without CSIT.*


**Remark** **60.**
*For large P, the SNR(h) in (Equation 196) saturates unless $P(s_{T})/P\to 1$ for all $s_{T}$, i.e., the high-SNR capacity is the same as the capacity without CSIT. The CSIT thus must become more accurate as P increases to improve the rate.*


**Remark** **61.**
*To optimize the power levels, consider (Equation 146) and*

(199)
$$\begin{array}{cc} \tilde{P}{\left(h\right)}^{\prime } & ={2|h|}^{2}\sqrt{{\tilde{P}}_{T}\left(h\right)}\;p\left(s_{T}|h\right) \end{array}$$


(200)
$$\begin{array}{cc} E{\left[{\left|Y\right|}^{2}|H=h\right]}^{\prime } & ={2|h|}^{2}\sqrt{P(s_{T})}\;p\left(s_{T}|h\right). \end{array}$$

*However, the resulting equations give little insight due to the expectation over H in (Equation 146). An exception is the on-off fading case where the expectation has only one term; see (Equation 254) and (Equation 255).*


### 6.5. Partial CSIR, Full CSIT

Suppose $S_{R}$ is a (perhaps noisy) function of *H*; see (Equation 172). The capacity is given by (Equation 160) for which we need to compute $p(y|a,s_{R})$ and $p(y|s_{R})$. The GMI with a *K*-partition of the output space $Y\times S_{R}$ can be helpful for these problems. We assume that the CSIR is either $S_{R}=0$ or $S_{R}=1\left(G\geq t\right)$ for some transmitter threshold *t*; see [95].

Suppose that $S_{T}=H$. We then have
$$\begin{array}{cc} p(y|a,s_{R}) & =\int_{C}p\left(h|s_{R}\right)\frac{\exp\left(-{\left|y-h\;x(h)\right|}^{2}\right)}{\pi }\;dh\\ p(y|s_{R}) & =\int_{C^{2}}p\left(h|s_{R}\right)\;p\left(x\left(h\right)\right)\frac{\exp\left(-{\left|y-h\;x(h)\right|}^{2}\right)}{\pi }\;dx\left(h\right)\;dh. \end{array}$$
Now select the X(h) to be jointly CSCG with variances $E\left[{\left|X\left(h\right)\right|}^{2}\right]=P\left(h\right)$ and correlation coefficients
$$\begin{array}{c} \rho \left(h,h^{\prime }\right)=\frac{E\left[X\left(h\right)X{\left(h^{\prime }\right)}^{*}\right]}{\sqrt{P\left(h\right)P\left(h^{\prime }\right)}} \end{array}$$
and where $E\left[P(H)\right]\leq P$. We then have
$$\begin{array}{c} p\left(y|s_{R}\right)=\int_{C}p\left(h\right)\frac{e^{-{\left|y\right|}^{2}/{\left(|h\right|}^{2}P\left(h\right)+1)}}{2\pi (|h{|}^{2}P\left(h\right)+1)}\;dh. \end{array}$$
As in (Equation 97), $p(y|s_{R})$ and therefore $h(Y|S_{R})$ depend only on the marginals p(x(h)) of *A* and not on the $\rho (h,h^{\prime })$. We thus have the problem of finding the $\rho (h,h^{\prime })$ that minimize
$$\begin{array}{c} h\left(Y|S_{R},A\right)=\int_{A}p\left(a\right)\;h\left(Y|S_{R},A=a\right)\;da. \end{array}$$
However, we study the conventional *A* in (Equation 88) for simplicity.

For the reverse model GMI (Equation 139), the averaging density in (Equation 164) and (Equation 165) is (cf. (Equation 194))
(201)$$\begin{array}{c} p\left(h,s_{T}|y,s_{R}\right)=\delta \left(s_{T}-h\right)\;\frac{p\left(h|s_{R}\right)\;p\left(y|h,s_{R}\right)}{p(y|s_{R})}. \end{array}$$
We again use numerical integration to compute the GMI.

For the forward model GMI, consider the same model and CSCG *X* as in Theorem 1. Since *H* is a function of $S_{T}$, we use (Equation 169) in Remark 50 to write
(202)$$\begin{array}{c} I_{1}\left(A;Y|S_{R}\right)=E\left[\log\left(1+\frac{\tilde{P}\left(S_{R}\right)}{1+\mathrm{Var}\left[\left.\sqrt{GP(H)}\right|S_{R}\right]}\right)\right] \end{array}$$
where (see (Equation 170))
(203)$$\begin{array}{cc} \tilde{P}\left(s_{R}\right) & =E{\left[\left.\sqrt{GP(H)}\;\right|S_{R}=s_{R}\right]}^{2} \end{array}$$
(204)$$\begin{array}{cc} E\left[{\left|Y\right|}^{2}|S_{R}=s_{R}\right] & =1+E\left[GP\left(H\right)|S_{R}=s_{R}\right]. \end{array}$$
The transmitter compensates for the phase of *H*, and it remains to adjust the transmit power levels P(h). We study five power control policies and two types of CSIR; see Table 2.

*Heuristic Policies:* The first three policies are reasonable heuristics and have the form
(205)$$\begin{array}{c} P\left(h\right)=\left\{\begin{array}{cc} \hat{P}\;g^{a}, & g\geq t\\ 0, & \mathrm{else} \end{array}\right. \end{array}$$
for some choice of real *a* and where
(206)$$\begin{array}{c} \hat{P}=\frac{P}{\int_{t}^{\infty }p\left(g\right)\;g^{a}\;dg}. \end{array}$$
In particular, choosing a=0,+1,−1, we obtain policies that we call truncated constant power (TCP), truncated matched filtering (TMF), and truncated channel inversion (TCI), respectively; see ([5], page 487), [95]. For such policies, we compute
(207)$$\begin{array}{cc} \tilde{P}\left(s_{R}\right) & =\hat{P}{\left(\int_{t}^{\infty }p\left(g|s_{R}\right)\;\sqrt{g^{1+a}}\;dg\right)}^{2} \end{array}$$
(208)$$\begin{array}{cc} E\left[GP\left(H\right)|S_{R}=s_{R}\right] & =\hat{P}\;\int_{t}^{\infty }p\left(g|s_{R}\right)\;g^{1+a}\;dg. \end{array}$$
These policies all have the form P(h)=P·f(h) for some function f(.) that is independent of *P*. The minimum SNR in (Equation 37) with C(P) replaced with the GMI is thus
(209)$$\begin{array}{cc} {\left.\frac{E_{b}}{N_{0}}\right|}_{\min}\; & =\frac{\left(\int_{t}^{\infty }p\left(g\right)\;g^{a}\;dg\right)\log2}{E\left[{\left(\int_{t}^{\infty }p\left(g|S_{R}\right)\;\sqrt{g^{1+a}}\;dg\right)}^{2}\right]}. \end{array}$$

For instance, consider the threshold t=0 (no truncation). The TCP (a=0) and TMF (a=1) policies have $\hat{P}=P$ while TCI (a=−1) has $P=\hat{P}/E\left[G^{-1}\right]$. For TCP, TMF, and TCI, we compute the respective
(210)$$\begin{array}{cc} \; & {\left.\frac{E_{b}}{N_{0}}\right|}_{\min}=\frac{\log2}{E\left[E{\left[\left.\sqrt{G}\;\right|S_{R}\right]}^{2}\right]} \end{array}$$
(211)$$\begin{array}{cc} \; & {\left.\frac{E_{b}}{N_{0}}\right|}_{\min}=\frac{\log2}{E\left[E{\left[G|S_{R}\right]}^{2}\right]} \end{array}$$
(212)$$\begin{array}{cc} \; & {\left.\frac{E_{b}}{N_{0}}\right|}_{\min}=E\left[G^{-1}\right]\log2. \end{array}$$
Applying Jensen’s inequality to the square root, square, and inverse functions in (Equation 210)–(Equation 212), we find that for t=0:the minimum $E_{b}/N_{0}$ of TCP and TCI is larger (worse) than −1.59 dB unless there is no fading;the minimum $E_{b}/N_{0}$ of TMF is smaller (better) than −1.59 dB unless $E\left[G|S_{R}\right]=E\left[G\right]=1$.
However, we emphasize that these claims apply to the GMI and not necessarily the mutual information; see Section 8.4.

*GMI-Optimal Policy:* The fourth policy is optimal for the GMI (Equation 202) and has the form of an MMSE precoder. This policy motivates a truncated MMSE (TMMSE) policy that generalizes and improves TMF and TCI.

Taking the derivative of the Lagrangian
(213)$$\begin{array}{c} I_{1}\left(A;Y|S_{R}\right)+\lambda \left(P-E\left[P(H)\right]\right) \end{array}$$
with respect to P(h) we have the following result.

**Theorem** **2.**
*The optimal power control policy for the GMI $I_{1}\left(A;Y|S_{R}\right)$ for the fading channels (Equation 159) with $S_{T}=H$ is*

(214)
$$\begin{array}{cc} \sqrt{P(h)} & =\frac{\alpha (h)|h|}{\lambda +{\beta \left(h\right)|h|}^{2}} \end{array}$$

*where λ>0 is chosen so that $E\left[P(H)\right]=P$ and*

(215)
$$\begin{array}{cc} \; & \alpha \left(h\right)=\int_{C}p\left(s_{R}|h\right)\frac{\sqrt{\tilde{P}\left(s_{R}\right)}}{E\left[{\left|Y\right|}^{2}|S_{R}=s_{R}\right]-\tilde{P}\left(s_{R}\right)}\;ds_{R} \end{array}$$


(216)
$$\begin{array}{cc} \; & \beta \left(h\right)=\int_{C}p\left(s_{R}|h\right)\frac{\tilde{P}\left(s_{R}\right)}{\left[E\left[{\left|Y\right|}^{2}|S_{R}=s_{R}\right]-\tilde{P}\left(s_{R}\right)\right]E\left[{\left|Y\right|}^{2}|S_{R}=s_{R}\right]}\;ds_{R}. \end{array}$$



**Proof.** Apply (Equation 146) with (Equation 203) and (Equation 204) to obtain
(217)$$\begin{array}{cc} \; & \tilde{P}{\left(s_{R}\right)}^{\prime }=2\left|h\right|\sqrt{\tilde{P}\left(s_{R}\right)}\;p\left(h|s_{R}\right) \end{array}$$
(218)$$\begin{array}{cc} \; & E{\left[{\left|Y\right|}^{2}|S_{R}=s_{R}\right]}^{\prime }=2{\left|h\right|}^{2}\sqrt{P(h)}\;p\left(h|s_{R}\right). \end{array}$$
Inserting into (Equation 146) and rearranging terms we obtain (Equation 214) with (Equation 215) and (Equation 216). □

**Remark** **62.**
*The expressions (Equation 215) and (Equation 216) are self-referencing, as $\tilde{P}\left(s_{R}\right)$ itself depends on α(h) and β(h). However, one simplification occurs if $S_{R}$ is a function of H: α(h) and β(h) are functions of $s_{R}$ only since the $p(s_{R}|h)$ in (Equation 215) and (Equation 216) is a Dirac generalized function.*


**Remark** **63.**
*Consider the expression (Equation 214). We effectively have a matched filter for small |h|; for large |h|, we effectively have a channel inversion. Recall that LMMSE filtering has similar behavior for low and high SNR, respectively.*


**Remark** **64.**
*A heuristic based on the optimal policy is a TMMSE policy where the transmitter sets P(h)=0 if G<t, and otherwise uses (Equation 214) but where α(h), β(h) are independent of h. There are thus four parameters to optimize: λ, α, β, and t. This TMMSE policy will outperform TMF and TCI in general, as these are special cases where β=0 and λ=0, respectively.*


*$S_{R}=0$:* For this CSIR, the GMI (Equation 202) simplifies to $I_{1}\left(A;Y\right)$ and the heuristic policy (TCP, TMF, TCI) rates are
(219)$$\begin{array}{c} I_{1}\left(A;Y\right)=\log\left(1+\frac{\hat{P}\;E{\left[\sqrt{G^{1+a}}\cdot 1\left(G\geq t\right)\right]}^{2}}{1+\hat{P}\;\mathrm{Var}\left[\sqrt{G^{1+a}}\cdot 1\left(G\geq t\right)\right]}\right). \end{array}$$
Moreover, the expression (Equation 209) gives
(220)$$\begin{array}{cc} {\left.\frac{E_{b}}{N_{0}}\right|}_{\min}\; & =\frac{E\left[G^{a}\cdot 1\left(G\geq t\right)\right]}{E{\left[\sqrt{G^{1+a}}\cdot 1\left(G\geq t\right)\right]}^{2}}\log2. \end{array}$$

For TCP, TMF, and TCI, we compute the respective
(221)$$\begin{array}{cc} \; & {\left.\frac{E_{b}}{N_{0}}\right|}_{\min}=\frac{\log2}{\mathrm{Pr}\left[G\geq t\right]E{\left[\left.\sqrt{G}\;\right|G\geq t\right]}^{2}} \end{array}$$
(222)$$\begin{array}{cc} \; & {\left.\frac{E_{b}}{N_{0}}\right|}_{\min}=\frac{\log2}{\int_{t}^{\infty }p\left(g\right)\;g\;dg} \end{array}$$
(223)$$\begin{array}{cc} \; & {\left.\frac{E_{b}}{N_{0}}\right|}_{\min}=\frac{E\left[\left.G^{-1}\right|G\geq t\right]}{\mathrm{Pr}\left[G\geq t\right]}\log2. \end{array}$$
Again applying Jensen’s inequality to the various functions in (Equation 221)–(Equation 223), we find that:the minimum $E_{b}/N_{0}$ of TMF is smaller (better) than that of TCP and TCI unless there is no fading, or if the minimal $E_{b}/N_{0}$ is −∞;the best threshold for TMF is t=0 and the minimal $E_{b}/N_{0}$ is −1.59 dB.

For the optimal policy, the parameters α(h) and β(h) in (Equation 215) and (Equation 216) are constants independent of *h*, see Remark 62, and the TMMSE policy with t=0 is the GMI-optimal policy.

**Remark** **65.**
*The TCI channel densities are*

$$\begin{array}{cc} p(y|a) & =\mathrm{Pr}\left[G<t\right]\frac{e^{-{\left|y\right|}^{2}}}{\pi }+\mathrm{Pr}\left[G\geq t\right]\frac{e^{-{\left|y-\sqrt{\hat{P}}\;u\right|}^{2}}}{\pi }\\ p(y) & =\mathrm{Pr}\left[G<t\right]\frac{e^{-{\left|y\right|}^{2}}}{\pi }+\mathrm{Pr}\left[G\geq t\right]\frac{e^{-{\left|y\right|}^{2}/\left(1+\hat{P}\right)}}{\pi (1+\hat{P})}. \end{array}$$



**Remark** **66.**
*At high SNR, one might expect that the receiver can estimate $P(S_{T})$ precisely even if $S_{R}=0$. We show that this is indeed the case for on-off fading by using the K=2 partition (Equation 154) of Remark 46. Moreover, the results prove that at high SNR one can approach I(A;Y); see Section 7.3.*


**Remark** **67.**
*For Rayleigh fading, the GMI with K=2 in (Equation 154) is helpful for both high and low SNR. For instance, for $S_{R}=0$ and TCI, the K=2 GMI approaches the mutual information for $S_{R}=1\left(G\geq t\right)$ as the SNR increases; see Remark 74 in Section 8.4. We further show that for $S_{R}=0$, the TCI policy can achieve a minimal $E_{b}/N_{0}$ of −∞ dB, see (Equation 301) in Section 8.4.*


*$S_{R}=1\left(G\geq t\right)$:* The heuristic policy rates are now (cf. (Equation 219) and note the $\mathrm{Pr}\left[G\geq t\right]$ term and conditioning)
(224)$$\begin{array}{c} I_{1}\left(A;Y|S_{R}\right)=\mathrm{Pr}\left[G\geq t\right]\log\left(1+\frac{\hat{P}\;E{\left[\left.\sqrt{G^{1+a}}\right|G\geq t\right]}^{2}}{1+\hat{P}\;\mathrm{Var}\left[\left.\sqrt{G^{1+a}}\right|G\geq t\right]}\right). \end{array}$$

Moreover, the expression (Equation 209) is
(225)$$\begin{array}{cc} \; & {\left.\frac{E_{b}}{N_{0}}\right|}_{\min}=\frac{E\left[\left.G^{a}\right|G\geq t\right]}{E{\left[\left.\sqrt{G^{1+a}}\right|G\geq t\right]}^{2}}\log2. \end{array}$$

For TCP, TMF, and TCI we compute the respective
(226)$$\begin{array}{cc} \; & {\left.\frac{E_{b}}{N_{0}}\right|}_{\min}=\frac{\log2}{E{\left[\left.\sqrt{G}\right|G\geq t\right]}^{2}} \end{array}$$
(227)$$\begin{array}{cc} \; & {\left.\frac{E_{b}}{N_{0}}\right|}_{\min}=\frac{\log2}{E\left[G|G\geq t\right]} \end{array}$$
(228)$$\begin{array}{cc} \; & {\left.\frac{E_{b}}{N_{0}}\right|}_{\min}=E\left[\left.G^{-1}\right|G\geq t\right]\log2. \end{array}$$

Again applying Jensen’s inequality to the various functions in (Equation 226)–(Equation 228), we find that:the minimum $E_{b}/N_{0}$ of all policies can be better than −1.59 dB by choosing t>0;the minimum $E_{b}/N_{0}$ of TMF is smaller (better) than that of TCP and TCI unless there is no fading or the minimal $E_{b}/N_{0}$ is −∞.

For the optimal policy, Remark 62 points out that α(h) and β(h) depend on $s_{R}$ only. We compute
(229)$$\begin{array}{c} \sqrt{P(h)}=\left\{\begin{array}{cc} \frac{\alpha_{0}\;\left|h\right|}{\lambda +\beta_{0}{\left|h\right|}^{2}}, & g<t\;\\ \frac{\alpha_{1}\;\left|h\right|}{\lambda +\beta_{1}{\left|h\right|}^{2}}, & g\geq t \end{array}\right. \end{array}$$
where for $s_{R}\in \left\{0,1\right\}$ we have
$$\begin{array}{cc} \; & \alpha_{s_{R}}=\frac{\sqrt{\tilde{P}\left(s_{R}\right)}}{E\left[{\left|Y\right|}^{2}|S_{R}=s_{R}\right]-\tilde{P}\left(s_{R}\right)}\\ \; & \beta_{s_{R}}=\frac{\sqrt{\tilde{P}\left(s_{R}\right)}}{\left[E\left[{\left|Y\right|}^{2}|S_{R}=s_{R}\right]-\tilde{P}\left(s_{R}\right)\right]E\left[{\left|Y\right|}^{2}|S_{R}=s_{R}\right]}. \end{array}$$

**Remark** **68.**
*The GMI (Equation 224) for TCI (a=−1) is the mutual information $I(A;Y|S_{R})$. To see this, observe that the model $q(y|a,s_{R})$ has*

$$\begin{array}{c} q\left(y|a,0\right)=\frac{e^{-{\left|y\right|}^{2}}}{\pi },\;q\left(y|a,1\right)=\frac{e^{-{\left|y-\sqrt{\hat{P}}\;u\right|}^{2}}}{\pi } \end{array}$$

*and thus we have $q\left(y|a,s_{R}\right)=p\left(y|a,s_{R}\right)$ for all $y,a,s_{R}$.*


### 6.6. Partial CSIR, CSIT@ R

Suppose next that $S_{R}$ is a noisy function of *H* (see for instance (Equation 172)) and $S_{T}=f_{T}\left(S_{R}\right)$. The capacity is given by (Equation 147) and we compute
(230)$$\begin{array}{c} I\left(X;Y|S_{R}\right)=E\left[\log\frac{p(Y|X,S_{R})}{p(Y|S_{R})}\right] \end{array}$$
where writing $s_{T}=f_{T}\left(s_{R}\right)$ we have
(231)$$\begin{array}{cc} p(y|s_{R},x) & =\int_{C}p\left(h|s_{R}\right)\frac{e^{-|y-h\;x\left(s_{T}\right){|}^{2}}}{\pi }\;dh \end{array}$$
(232)$$\begin{array}{cc} p(y|s_{R}) & =\int_{C^{2}}p\left(h|s_{R}\right)\;p\left(x\left(s_{T}\right)\right)\;\frac{e^{-|y-h\;x\left(s_{T}\right){|}^{2}}}{\pi }\;dx\left(s_{T}\right)\;dh. \end{array}$$
For example, if $X(s_{T})$ is CSCG with variance $P(s_{T})$ then
(233)$$\begin{array}{c} p\left(y|s_{R}\right)=\int_{C}p\left(h|s_{R}\right)\frac{\exp\left(-\frac{{\left|y\right|}^{2}}{1+{\left|h\right|}^{2}P\left(s_{T}\right)}\right)}{\pi (1+|h{|}^{2}P\left(s_{T}\right))}\;dh. \end{array}$$
One can compute $I(X;Y|S_{R})$ numerically using (Equation 231) and (Equation 232). However, optimizing over $X(s_{T})$ is usually difficult.

For the reverse model GMI (Equation 139), the averaging density in (Equation 164) and (Equation 165) is now (cf. (Equation 194) and (Equation 201))
(234)$$\begin{array}{c} p\left(h,s_{T}|y,s_{R}\right)=\delta \left(s_{T}-f_{T}\left(s_{R}\right)\right)\;\frac{p\left(h|s_{R}\right)\;p\left(y|h,s_{R}\right)}{p(y|s_{R})}. \end{array}$$
We use numerical integration to compute the rates.

The forward model GMI again gives more insight. Define the channel gain and variance as the respective
(235)$$\begin{array}{cc} \tilde{g}\left(s_{R}\right) & ={\left|E\left[H|S_{R}=s_{R}\right]\right|}^{2} \end{array}$$
(236)$$\begin{array}{cc} {\tilde{\sigma }}^{2}\left(s_{R}\right) & =\mathrm{Var}\left[H|S_{R}=s_{R}\right]. \end{array}$$

**Theorem** **3.**
*An achievable rate for AWGN fading channels (Equation 159) with power constraint $E\left[{\left|X\right|}^{2}\right]\leq P$ and with partial CSIR $S_{R}$ and $S_{T}=f_{T}\left(S_{R}\right)$ is*

(237)
$$\begin{array}{c} I_{1}\left(X;Y|S_{R}\right)=E\left[\log\left(1+\frac{\tilde{g}\left(S_{R}\right)P\left(S_{T}\right)}{1+{\tilde{\sigma }}^{2}\left(S_{R}\right)P\left(S_{T}\right)}\right)\right] \end{array}$$

*where $E\left[P(S_{T})\right]=P$. The optimal power levels $P(s_{T})$ are obtained by solving*

(238)
$$\begin{array}{cc} \; & \lambda =\int_{R}p\left(s_{R}|s_{T}\right)\frac{\tilde{g}\left(s_{R}\right)}{\left[1+\left(\tilde{g}\left(s_{R}\right)+{\tilde{\sigma }}^{2}\left(s_{R}\right)\right)P\left(s_{T}\right)\right]\left[1+{\tilde{\sigma }}^{2}\left(s_{R}\right)P\left(s_{T}\right)\right]}\;ds_{R}. \end{array}$$

*In particular, if $S_{T}$ determines $S_{R}$ (CSIR@T) then we have the quadratic waterfilling expression*

(239)
$$\begin{array}{c} f\left(P\left(s_{T}\right),\;\tilde{g}\left(s_{R}\right),\;{\tilde{\sigma }}^{2}\left(s_{R}\right)\right)={\left(\frac{1}{\lambda }-\frac{1}{\tilde{g}\left(s_{R}\right)}\right)}^{+} \end{array}$$

*where*

(240)
$$\begin{array}{c} f\left(Q,g,\sigma^{2}\right)=\left(1+2\frac{\sigma^{2}}{g}\right)Q+\left(1+\frac{\sigma^{2}}{g}\right)\sigma^{2}Q^{2} \end{array}$$

*and where λ is chosen so that $E\left[P(H_{R})\right]=P$.*


**Proof.** Apply Theorem 1 with
(241)$$\begin{array}{cc} \tilde{P}\left(s_{R}\right) & =\tilde{g}\left(s_{R}\right)P\left(s_{T}\right) \end{array}$$
(242)$$\begin{array}{cc} E\left[{\left|Y\right|}^{2}|S_{R}=s_{R}\right] & =1+\left(\tilde{g}\left(s_{R}\right)+{\tilde{\sigma }}^{2}\left(s_{R}\right)\right)P\left(s_{T}\right) \end{array}$$
to obtain (Equation 237). To optimize the power levels $P(s_{T})$ with (Equation 146), consider the derivatives
(243)$$\begin{array}{cc} \tilde{P}{\left(s_{R}\right)}^{\prime } & =2\tilde{g}\left(s_{R}\right)\sqrt{P(s_{T})}1\left(s_{T}=f_{T}\left(s_{R}\right)\right) \end{array}$$
(244)$$\begin{array}{cc} E{\left[{\left|Y\right|}^{2}|S_{R}=s_{R}\right]}^{\prime } & =2\left(\tilde{g}\left(s_{R}\right)+{\tilde{\sigma }}^{2}\left(s_{R}\right)\right)\sqrt{P(s_{T})}1\left(s_{T}=f_{T}\left(s_{R}\right)\right). \end{array}$$The expression (Equation 146) thus becomes (Equation 238). If $S_{T}$ determines $S_{R}$ then the expression simplifies to
$$\begin{array}{c} \lambda =\frac{\tilde{g}\left(s_{R}\right)}{\left[1+\left(\tilde{g}\left(s_{R}\right)+{\tilde{\sigma }}^{2}\left(s_{R}\right)\right)P\left(s_{T}\right)\right]\left[1+{\tilde{\sigma }}^{2}\left(s_{R}\right)P\left(s_{T}\right)\right]} \end{array}$$
from which we obtain (Equation 239). □

**Remark** **69.**
*The optimal power control policy with CSIT@R and CSIR@T can be written explicitly by solving the quadratic in (Equation 239). The result is*

(245)
$$\begin{array}{c} P\left(s_{T}\right)=\frac{\tilde{g}+2{\tilde{\sigma }}^{2}}{2{\tilde{\sigma }}^{2}\left(\tilde{g}+{\tilde{\sigma }}^{2}\right)}\left[\sqrt{1+4{\tilde{\sigma }}^{2}{\left(\frac{1}{\lambda }-\frac{1}{\tilde{g}}\right)}^{+}\frac{\tilde{g}\;\left(\tilde{g}+{\tilde{\sigma }}^{2}\right)}{{\left(\tilde{g}+2{\tilde{\sigma }}^{2}\right)}^{2}}}-1\right] \end{array}$$

*where we have discarded the dependence on $s_{R}$ for convenience. The alternative form (Equation 239) relates to the usual waterfilling where the left-hand side of (Equation 239) is $P(s_{T})$. Observe that ${\tilde{\sigma }}^{2}=0$ gives conventional waterfilling.*


**Remark** **70.**
*As in Section 3.3, we show that at high SNR the K=2 GMI of Remark 42 approaches the upper bound of Proposition 2 in some cases; see Section 7.4. The channel parameters depend on $s_{R}$, and we choose $h_{1}\left(s_{R}\right)=0$ and $\sigma_{1}^{2}\left(s_{R}\right)=\sigma_{2}^{2}\left(s_{R}\right)=1$ for all $s_{R}$.*


## 7. On-Off Fading

Consider again on-off fading with $P_{G}\left(0\right)=P_{G}\left(2\right)=1/2$. We study the scenarios listed in Table 1. The case of no CIR and no CSIT was studied in Section 3.3.

### 7.1. Full CSIR, CSIT@ R

Consider $S_{R}=H$. The capacity with B=0 (no CSIT) is given by (Equation 175) (cf. (Equation 73)):(246)$$\begin{array}{c} C\left(P\right)=\frac{1}{2}\log\left(1+2P\right) \end{array}$$
and the wideband values are given by (Equation 177) (cf. (Equation 74)); the minimal $E_{b}/N_{0}$ is log2 and the slope is S=1.

The capacity with B=∞ (or $S_{T}=G$) increases to
(247)$$\begin{array}{cc} C(P) & =\frac{1}{2}\log\left(1+4P\right) \end{array}$$
where P(0)=0 and P(2)=2P. This capacity is also achieved with B=1 since there are only two values for *G*. We compute $C^{\prime }\left(0\right)=2$ and $C^{\prime \prime }\left(0\right)=-8$, and therefore
(248)$$\begin{array}{c} {\left.\frac{E_{b}}{N_{0}}\right|}_{\min}=\frac{\log2}{2},\;S=1. \end{array}$$
The power gain due to CSIT compared to no fading is thus 3.01 dB, but the capacity slope is the same. The rate curves are compared in Figure 4.

### 7.2. Full CSIR, Partial CSIT

Consider next noisy CSIT with $0\leq \epsilon \leq \frac{1}{2}$ and
$$\begin{array}{c} \mathrm{Pr}\left[S_{T}=G\right]=\bar{\epsilon },\;\mathrm{Pr}\left[S_{T}\neq G\right]=\epsilon . \end{array}$$
*$S_{R}=H\sqrt{P(S_{T})}$:* The capacity of Proposition 2 is
(249)$$\begin{array}{c} C\left(P\right)=\max_{P(0)+P(2)=2P}\frac{\epsilon }{2}\log\left(1+2P(0)\right)+\frac{\bar{\epsilon }}{2}\log\left(1+2P(2)\right). \end{array}$$
Optimizing the power levels, we have
(250)$$\begin{array}{c} P\left(0\right)={\left(2\epsilon \;P-\frac{\bar{\epsilon }-\epsilon }{2}\right)}^{+},\;P\left(2\right)=2P-P\left(0\right). \end{array}$$
Figure 4 shows C(P) for ϵ=0.1 as the curve labeled “Best CSIR”. For $P\geq \left(\bar{\epsilon }-\epsilon \right)/\left(4\epsilon \right)$, we compute
(251)$$\begin{array}{c} C\left(P\right)=\frac{1}{2}\log\left(1+2P\right)+\frac{1}{2}\left[1-H_{2}\left(\epsilon \right)\right]\log2 \end{array}$$
where $H_{2}\left(\epsilon \right)=-\epsilon \log_{2}\epsilon -\bar{\epsilon }\log_{2}\bar{\epsilon }$ is the binary entropy function. For example, if ϵ=0.1 then for P≥2 one gains $\Delta C=[1-H_{2}\left(0.1\right)]/2\approx 0.27$ bits over the capacity without CSIT. This translates to an SNR gain of $2\Delta C\cdot 10\log_{10}\left(2\right)\approx 1.60$ dB. On the other hand, for $P\leq \left(\bar{\epsilon }-\epsilon \right)/\left(4\epsilon \right)$ we have P(0)=0, P(2)=2P, and the capacity is
(252)$$\begin{array}{c} C\left(P\right)=\frac{\bar{\epsilon }}{2}\log\left(1+4P\right). \end{array}$$
We have $C^{\prime }\left(0\right)=2\;\bar{\epsilon }$ and lose a fraction of $\bar{\epsilon }$ of the power as compared to having full CSIT (ϵ=0). For example, if ϵ=0.1, the minimal $E_{b}/N_{0}$ is approximately −4.14 dB.

*$S_{R}=H$:* To compute I(A;Y|H) in (Equation 190), we write (Equation 191) and (Equation 193) for CSCG $X(s_{T})$ as
$$\begin{array}{cc} p_{Y|A,H}\left(y|a,0\right) & =p_{Y|H}\left(y|0\right)=\frac{e^{-{\left|y\right|}^{2}}}{\pi }\\ p_{Y|A,H}\left(y|a,\sqrt{2}\;\right) & =\epsilon \;\frac{e^{-{\left|y-\sqrt{2}x\left(0\right)\right|}^{2}}}{\pi }+\bar{\epsilon }\;\frac{e^{-{\left|y-\sqrt{2}x\left(\sqrt{2}\;\right)\right|}^{2}}}{\pi }\\ p_{Y|H}\left(y|\sqrt{2}\;\right) & =\epsilon \frac{\exp\left(-\frac{{\left|y\right|}^{2}}{1+2P(0)}\right)}{\pi (1+2P(0))}+\bar{\epsilon }\;\frac{\exp\left(-\frac{{\left|y\right|}^{2}}{1+2P(2)}\right)}{\pi (1+2P(2))}. \end{array}$$
Figure 4 shows the rates as the curve labeled “I(A;Y|H)”. This curve was computed by Monte Carlo integration with P(0)=0.1·P and P(2)=1.9·P, which is near-optimal for the range of SNRs depicted.

The reverse model GMI (Equation 139) requires $\mathrm{Var}\left[U|Y,H\right]$. We show how to compute this variance in Section C.2 by applying (Equation 164) and (Equation 165). Figure 4 shows the GMIs as the curve labeled “rGMI”, where we used the same power levels as for the I(A;Y|H) curve. The two curves are indistinguishable for small *P*, but the “rGMI” rates are poor at large *P*. This example shows that the forward model GMI with optimized powers can be substantially better than the reverse model GMI with a reasonable but suboptimal power policy.

The forward model GMI (Equation 195) is
(253)$$\begin{array}{cc} \; & I_{1}\left(A;Y|H\right)=\frac{1}{2}\log\left(1+\mathrm{SNR}\left(\sqrt{2}\right)\right) \end{array}$$
where $\mathrm{SNR}\left(\sqrt{2}\right)$ is given by (Equation 196) with
$$\begin{array}{cc} {\tilde{P}}_{T}\left(\sqrt{2}\right) & ={\left(\epsilon \sqrt{P(0)}+\bar{\epsilon }\sqrt{P(2)}\right)}^{2}\\ \mathrm{Var}\left[\left.\sqrt{P(S_{T})}\right|H=h\right] & =1+2\;\epsilon \;\bar{\epsilon }{\left(\sqrt{P(2)}-\sqrt{P(0)}\right)}^{2}. \end{array}$$
Applying Remark 61, the optimal power control policy is
(254)$$\begin{array}{c} \sqrt{P(s_{T})}=\frac{p_{H|S_{T}}\left(\sqrt{2}\;|s_{T}\right)}{\gamma +\beta \;p_{H|S_{T}}\left(\sqrt{2}\;|s_{T}\right)}=\left\{\begin{array}{cc} \frac{\epsilon }{\gamma +\beta \;\epsilon }, & s_{T}=0\;\\ \frac{\bar{\epsilon }}{\gamma +\beta \;\bar{\epsilon }}, & s_{T}=2 \end{array}\right. \end{array}$$
where
(255)$$\begin{array}{cc} \beta & =\frac{2\sqrt{{\tilde{P}}_{T}\left(\sqrt{2}\;\right)}}{E\left[{\left|Y\right|}^{2}|H=\sqrt{2}\;\right]} \end{array}$$
and γ≥0 is chosen so that P(0)+P(2)=2P. Figure 4 shows the resulting GMI as the curve labeled “GMI, *K* = 1”. At low SNR, we achieve the rate ${\tilde{P}}_{T}\left(\sqrt{2}\;\right)$ and the optimal power control has β→0 so that
(256)$$\begin{array}{cc} \; & P\left(0\right)=\frac{2P\epsilon^{2}}{\epsilon^{2}+{\bar{\epsilon }}^{2}},\;P\left(2\right)=\frac{2P{\bar{\epsilon }}^{2}}{\epsilon^{2}+{\bar{\epsilon }}^{2}} \end{array}$$
and therefore
(257)$$\begin{array}{cc} \; & {\tilde{P}}_{T}\left(\sqrt{2}\right)=2\left(\epsilon^{2}+{\bar{\epsilon }}^{2}\right)P. \end{array}$$
We have $C^{\prime }\left(0\right)=2\left(\epsilon^{2}+{\bar{\epsilon }}^{2}\right)$ and lose a fraction of $\left(\epsilon^{2}+{\bar{\epsilon }}^{2}\right)$ of the power as compared to having full CSIT (ϵ=0). For example, if ϵ=0.1, the minimal $E_{b}/N_{0}$ is approximately −3.74 dB.

We remark that the I(A;Y|H) and reverse model GMI curves lie above the forward model curve if we choose the same power policy as for the forward channel.

### 7.3. Partial CSIR, Full CSIT

This section studies $S_{T}=H$. The capacity with partial CSIR is given by (Equation 138) for which we need to compute $p(y|a,s_{R})$ and $p(y|s_{R})$. We consider two cases.

*$S_{R}=1\left(G\geq t\right)$:* Here we recover the case with full CSIR by choosing *t* to satisfy 0<t≤2.

*$S_{R}=0$:* The best power policy clearly has P(0)=0 and $P(\sqrt{2}\;)=2P$. The mutual information is thus $I\left(A;Y\right)=I\left(X\left(\sqrt{2}\right);Y\right)$ and the channel densities are (cf. (Equation 75) and (Equation 76))
$$\begin{array}{cc} p(y|a) & =\frac{e^{-{\left|y\right|}^{2}}}{2\pi }+\frac{e^{-{\left|y-2\sqrt{P}u\left(\sqrt{2}\;\right)\right|}^{2}}}{2\pi }\\ p(y) & =\frac{e^{-{\left|y\right|}^{2}}}{2\pi }+\frac{e^{-{\left|y\right|}^{2}/\left(1+4P\right)}}{2\pi (1+4P)}. \end{array}$$

The rates I(A;Y) are shown in Figure 5. Observe that the low-SNR rates are larger than without fading; this is a consequence of the slightly bursty nature of transmission.

The reverse model GMI (Equation 139) requires $\mathrm{Var}\left[U|Y\right]$. We compute this variance in Section C.3 by using (Equation 164) and (Equation 165) with (Equation 201) and $\phi (s_{T})=0$. Figure 5 shows the GMIs as the curve labeled “rGMI”.

Next, the TCP, TMF, TCI, and TMMSE policies are the same for 0<t≤2, since they use P(0)=0 and $P\left(\sqrt{2}\;\right)=2P$. The resulting rate is given by (Equation 202)–(Equation 204) with $\tilde{P}\left(0\right)=0$, $\tilde{P}\left(1\right)=P$, and $\mathrm{Var}\left[\left.\sqrt{GP(S_{T})}\right|S_{R}=1\right]=P$ and
(258)$$\begin{array}{cc} \; & I_{1}\left(A;Y\right)=\log\left(1+\frac{P}{1+P}\right). \end{array}$$
The rates are plotted in Figure 5 as the curve labeled “GMI, *K* = 1”. This example again shows that choosing K=1 is a poor choice at high SNR.

To improve the auxiliary model at high SNR, consider the GMI (Equation 154) with K=2 and the subsets (Equation 65). We further choose the parameters $h_{1}=0$, $\sigma_{1}^{2}=0$, $h_{2}=2$, $\sigma_{2}^{2}=1$, and adaptive coding with X(0)=0, $X\left(\sqrt{2}\;\right)=\sqrt{2P}\;U$, $\bar{X}=\sqrt{P}\;U$, where U∼CN(0,1). The GMI (Equation 154) is
(259)$$\begin{array}{cc} I_{1}\left(A;Y\right)=\mathrm{Pr}\left[E_{2}\right] & \left[\log\left(1+4P\right)+\frac{E\left[{\left|Y\right|}^{2}|E_{2}\right]}{1+4P}-E\left[\left.{\left|Y-\sqrt{4P}\;U\right|}^{2}\right|E_{2}\right]\right]. \end{array}$$
In Section B.2, we show that choosing $t_{R}=P^{\lambda_{R}}+b$ where $0<\lambda_{R}<1$ and *b* is a real constant makes all terms behave as desired as *P* increases:(260)$$\begin{array}{c} \mathrm{Pr}\left[E_{2}\right]\to 1/2,\;\frac{E\left[{\left|Y\right|}^{2}|E_{2}\right]}{1+4P}\to 1,\;E\left[\left.{\left|Y-\sqrt{4P}U\right|}^{2}\right|E_{2}\right]\to 1. \end{array}$$
We thus have
(261)$$\begin{array}{c} \lim_{P\to \infty }\left[\frac{1}{2}\log\left(1+4P\right)-I_{1}\left(X;Y\right)\right]=0. \end{array}$$
Figure 5 shows the behavior of $I_{1}\left(A;Y\right)$ for $\lambda_{R}=1/2$ and b=3 as the curve labeled “GMI, *K* = 2”. As for the case without CSIT, the receiver can estimate *H* accurately at large SNR, and one approaches the capacity with full CSIR.

Finally, the large-*K* forward model rates are computed using (Equation 70) but where $\bar{X}$ replaces *X*. One may again use the results of Section C.3 and the relations
$$\begin{array}{cc} E\left[\left.\bar{X}\right|Y=y\right] & =\sqrt{P}\;E\left[U|Y=y\right]\\ E\left[\left.|\bar{X}{|}^{2}\right|Y=y\right] & =P\;E\left[{\left|U\right|}^{2}|Y=y\right]\\ \mathrm{Var}\left[\bar{X}|Y=y\right] & =P\;\mathrm{Var}\left[U|Y=y\right]. \end{array}$$
The rates are shown as the curve labeled “GMI, K = *∞*” in Figure 5. So again, the large-*K* forward model is good at high SNR but worse than the best K=1 model at low SNR.

### 7.4. Partial CSIR, CSIT@ R

Consider partial CSIR with $S_{T}=S_{R}$ and
(262)$$\begin{array}{c} \mathrm{Pr}\left[S_{R}=H\right]=\bar{\epsilon },\;\mathrm{Pr}\left[S_{R}\neq H\right]=\epsilon \end{array}$$
where $0\leq \epsilon \leq \frac{1}{2}$. We thus have both CSIT@R and CSIR@T. To compute $I(X;Y|S_{R})$ in (Equation 230), we write (Equation 231) and (Equation 232) as
$$\begin{array}{cc} p_{Y|S_{R},X}\left(y|0,x\right) & =\bar{\epsilon }\;\frac{e^{-{\left|y\right|}^{2}}}{\pi }+\epsilon \;\frac{e^{-{\left|y-\sqrt{2}\;x\left(0\right)\right|}^{2}}}{\pi }\\ p_{Y|S_{R},X}\left(y|\sqrt{2},x\right) & =\bar{\epsilon }\;\frac{e^{-{\left|y-\sqrt{2}\;x\left(\sqrt{2}\;\right)\right|}^{2}}}{\pi }+\epsilon \;\frac{e^{-{\left|y\right|}^{2}}}{\pi }\\ p_{Y|S_{R}}\left(y|0\right) & =\bar{\epsilon }\;\frac{e^{-{\left|y\right|}^{2}}}{\pi }+\epsilon \;\frac{e^{-{\left|y\right|}^{2}/\left[1+2P\left(0\right)\right]}}{\pi [1+2P(0)]}\\ p_{Y|S_{R}}\left(y|\sqrt{2}\right) & =\bar{\epsilon }\;\frac{e^{-{\left|y\right|}^{2}/\left[1+2P\left(\sqrt{2}\;\right)\right]}}{\pi \left[1+2P\left(\sqrt{2}\;\right)\right]}+\epsilon \;\frac{e^{-{\left|y\right|}^{2}}}{\pi } \end{array}$$
where $X(s_{T})$ is CSCG. We choose the transmit powers P(0) and $P\left(\sqrt{2}\;\right)$ as in (Equation 250) to compare with the best CSIR. Figure 6 shows the resulting rates for ϵ=0.1 as the curve labeled “Partial CSIR, $I(X;Y|S_{R})$”. Observe that at high SNR, the curve seems to approach the best CSIR curve from Figure 4 with $S_{R}=H\sqrt{P(S_{T})}$. We prove this by studying a forward model GMI with K=2.

The reverse model GMI requires $\mathrm{Var}\left[U|Y,S_{R}\right]$, which can be computed by simulation; see Section C.4. However, optimizing the powers seems difficult. We instead focus on the forward model GMI of Theorem 3 for which we compute
$$\begin{array}{cc} \; & \tilde{g}\left(0\right)=2\;\epsilon^{2},\;\tilde{g}\left(\sqrt{2}\right)=2\;{\bar{\epsilon }}^{2},\;{\tilde{\sigma }}^{2}\left(0\right)={\tilde{\sigma }}^{2}\left(\sqrt{2}\right)=2\;\epsilon \;\bar{\epsilon } \end{array}$$
and therefore (Equation 237) is
(263)$$\begin{array}{c} I_{1}\left(X;Y|S_{R}\right)=\frac{1}{2}\log\left(1+\frac{2\;\epsilon^{2}P\left(0\right)}{1+2\;\epsilon \;\bar{\epsilon }\;P\left(0\right)}\right)+\frac{1}{2}\log\left(1+\frac{2\;{\bar{\epsilon }}^{2}P\left(\sqrt{2}\;\right)}{1+2\;\epsilon \;\bar{\epsilon }\;P\left(\sqrt{2}\;\right)}\right). \end{array}$$
For CSIR@T, the optimal power control policy is given by the quadratic waterfilling specified by (Equation 239) or (Equation 245):$$\begin{array}{cc} P(0) & =\frac{1+\bar{\epsilon }}{4\;\epsilon \;\bar{\epsilon }}\left[\sqrt{1+8\;\epsilon \;\bar{\epsilon }{\left(\frac{1}{\lambda }-\frac{1}{2\;\epsilon^{2}}\right)}^{+}\frac{\epsilon }{{\left(1+\bar{\epsilon }\right)}^{2}}}-1\right]\\ P\left(\sqrt{2}\right) & =\frac{1+\epsilon }{4\;\epsilon \;\bar{\epsilon }}\left[\sqrt{1+8\;\epsilon \;\bar{\epsilon }{\left(\frac{1}{\lambda }-\frac{1}{2\;{\bar{\epsilon }}^{2}}\right)}^{+}\frac{\bar{\epsilon }}{{\left(1+\epsilon \right)}^{2}}}-1\right]. \end{array}$$
The rates are shown in Figure 6 as the curve labeled “Partial CSIR, GMI, *K* = 1”. Observe that at high SNR the GMI (Equation 263) saturates at
(264)$$\begin{array}{c} \frac{1}{2}\log\left(1+\frac{\epsilon }{\bar{\epsilon }}\right)+\frac{1}{2}\log\left(1+\frac{\bar{\epsilon }}{\epsilon }\right). \end{array}$$
For example, for ϵ=0.1, we approach 1.74 bits at high SNR. On the other hand, at low SNR, the rate is maximized with P(0)=0 and $P\left(\sqrt{2}\right)=2P$ so that $I_{1}\left(X;Y|S_{R}\right)\approx 2\;{\bar{\epsilon }}^{2}P$. We thus achieve a fraction of ${\bar{\epsilon }}^{2}$ of the power compared to full CSIT. For example, if ϵ=0.1, the minimal $E_{b}/N_{0}$ is approximately −3.69 dB.

Figure 6 also shows the conventional waterfilling rates as the curve labeled “Partial CSIR, GMI, c-waterfill”. These rates are almost the same as the quadratic waterfilling rates except for the range of $E_{b}/N_{0}$ between 9 to 13 dB shown in the inset.

To improve the auxiliary model at high SNR, we use a K=2 GMI with (see Remark 70)
$$\begin{array}{c} h_{1}\left(s_{R}\right)=0,\;h_{2}\left(s_{R}\right)=\sqrt{2},\;\sigma_{1}^{2}\left(s_{R}\right)=\sigma_{2}^{2}\left(s_{R}\right)=1 \end{array}$$
for $s_{R}=0,\sqrt{2}$. The receiver chooses $\bar{X}\left(s_{R}\right)=\sqrt{P(s_{R})}\;U$ (see Remark 41) and we have (see Remark 42)
(265)$$\begin{array}{cc} I_{1}\left(X;Y|S_{R}\right) & =\frac{1}{2}\mathrm{Pr}\left[E_{2}|S_{R}=0\right]\left\{\log\left(1+2P(0)\right)+\frac{E\left[\left.{\left|Y\right|}^{2}\right|E_{2},S_{R}=0\right]}{1+2P(0)}\right.\\ \; & \left.\;-E\left[\left.|Y-\sqrt{2}\;X\left(0\right){|}^{2}\right|E_{2},S_{R}=0\right]\right\}\\ \; & \;+\frac{1}{2}\mathrm{Pr}\left[E_{2}|S_{R}=\sqrt{2}\;\right]\left\{\log\left(1+2P\left(\sqrt{2}\;\right)\right)+\frac{E\left[\left.{\left|Y\right|}^{2}\right|E_{2},S_{R}=\sqrt{2}\;\right]}{1+2P\left(\sqrt{2}\;\right)}\right.\\ \; & \left.\;-E\left[\left.|Y-\sqrt{2}\;X\left(\sqrt{2}\;\right){|}^{2}\right|E_{2},S_{R}=\sqrt{2}\right]\right\} \end{array}$$
where the $X(s_{T})$, $s_{T}\in S_{T}$, are given by (Equation 122). We consider P(0) and $P\left(\sqrt{2}\;\right)$ that scale in proportion to *P*. In this case, Section B.3 shows that choosing $t_{R}=P^{\lambda_{R}}$ where $0<\lambda_{R}<1$ gives the (best) full-CSIR capacity for large *P*, which is the rate specified in (Equation 249):(266)$$\begin{array}{cc} \lim_{P\to \infty } & \left[\frac{\epsilon }{2}\log\left(1+2P(0)\right)+\frac{\bar{\epsilon }}{2}\log\left(1+2P\left(\sqrt{2}\;\right)\right)-I_{1}\left(X;Y|S_{R}\right)\right]=0. \end{array}$$
In other words, by optimizing P(0) and $P\left(\sqrt{2}\;\right)$, at high SNR the K=2 GMI can approach the capacity of Proposition 2. This is expected since the receiver can estimate $H\sqrt{P(S_{T})}$ reliably at high SNR.

Figure 6 shows the behavior of this GMI and $t_{R}=P^{0.4}$, and where we have chosen P(0) and $P\left(\sqrt{2}\;\right)$ according to (Equation 250). The abrupt change in slope at approximately 2.5 dB is because P(0) becomes positive beyond this $E_{b}/N_{0}$. Keeping P(0)=0 for $E_{b}/N_{0}$ up to about 12 dB gives better rates, but for high SNR one should choose the powers according to (Equation 250).

## 8. Rayleigh Fading

Rayleigh fading has H∼CN(0,1). The random variable $G={\left|H\right|}^{2}$ thus has the density $p\left(g\right)=e^{-g}\cdot 1\left(g\geq 0\right)$. Section 8.1 and Section 8.2 review known results.

### 8.1. No CSIR, No CSIT

Suppose $S_{R}=S_{T}=0$ and X∼CN(0,P). The densities to compute I(X;Y) for CSCG *X* are
(267)$$\begin{array}{cc} p(y|x) & =\frac{e^{-{\left|y\right|}^{2}/{\left(|x\right|}^{2}+1)}}{\pi (|x{|}^{2}+1)} \end{array}$$
(268)$$\begin{array}{cc} p(y) & =\int_{0}^{\infty }\frac{e^{-g/P}}{P}\;\frac{e^{-{\left|y\right|}^{2}/\left(g+1\right)}}{\pi (g+1)}\;dg. \end{array}$$
The minimum $E_{b}/N_{0}$ is approximately 9.2 dB, and the forward model GMI (Equation 174) is zero. The capacity is achieved by discrete and finite *X* [96], and at large SNR, the capacity behaves as loglogP [97]. Further results are derived in [98,99,100,101,102].

### 8.2. Full CSIR, CSIT@ R

The capacity (Equation 175) for B=0 (no CSIT) is
(269)$$\begin{array}{c} C\left(P\right)=\int_{0}^{\infty }e^{-g}\log\left(1+g\;P\right)dg=e^{1/P}E_{1}\left(1/P\right)\log\left(e\right) \end{array}$$
where the exponential integral $E_{1}\left(.\right)$ is given by (Equation 371) below. The wideband values are given by (Equation 177):$$\begin{array}{c} {\left.\frac{E_{b}}{N_{0}}\right|}_{\min}=\log2,\;S=1. \end{array}$$
The minimal $E_{b}/N_{0}$ is −1.59 dB, but the fading reduces the capacity slope. At high SNR, we have
$$\begin{array}{c} C(P)\approx \log(P)-\gamma \end{array}$$
where γ≈0.57721 is Euler’s constant. The capacity thus behaves as for the case without fading but with an SNR loss of approximately 2.5 dB.

The capacity (Equation 182) with B=∞ (or $S_{T}=G$) is (see ([95], Equation (Equation 7)))
(270)$$\begin{array}{cc} C(P)\; & =\int_{\lambda }^{\infty }e^{-g}\log\left(g/\lambda \right)dg=E_{1}\left(\lambda \right). \end{array}$$
where P(g) is given by (Equation 181) and λ is chosen so that
$$\begin{array}{cc} P & =\int_{\lambda }^{\infty }e^{-g}P\left(g\right)\;dg=\frac{e^{-\lambda }}{\lambda }-E_{1}\left(\lambda \right). \end{array}$$
At low SNR we have large λ and using the approximation (Equation 374) below we compute
(271)$$\begin{array}{c} C\left(P\right)\approx e^{-\lambda }/\lambda \;\mathrm{and}\;P\approx e^{-\lambda }/\lambda^{2}. \end{array}$$
We thus have $E_{b}/N_{0}\approx \log\left(2\right)/\lambda$ and the minimal $E_{b}/N_{0}$ is −∞.

Consider now B=1 for which $P_{S_{T}}\left(3\Delta /2\right)=e^{-\Delta }$ and
(272)$$\begin{array}{cc} \; & E\left[G|G\geq \Delta \right]=1+\Delta \end{array}$$
(273)$$\begin{array}{cc} \; & E\left[G^{2}|G\geq \Delta \right]=2+2\Delta +\Delta^{2}. \end{array}$$
We thus have the wideband quantities in (Equation 186) and (Equation 187): (274)$$\begin{array}{cc} \; & {\left.\frac{E_{b}}{N_{0}}\right|}_{\min}=\frac{\log2}{1+\Delta } \end{array}$$(275)$$\begin{array}{cc} \; & S=\frac{2e^{-\Delta }{\left(1+\Delta \right)}^{2}}{2+2\Delta +\Delta^{2}}. \end{array}$$

Figure 7 shows the capacities for B=1 and Δ=1,2,1/2. The minimum $E_{b}/N_{0}$ value is
(276)$$\begin{array}{c} -1.59\;\mathrm{dB}-10\log_{10}\left(1+\Delta \right) \end{array}$$
and for Δ=1,2,1/2 we gain 3 dB, 4.8 dB, 1.8 dB, respectively, over no CSIT at low power. Note that one bit of feedback allows one to approach the full CSIT rates closely.

**Remark** **71.**
*For the scalar channel (Equation 159), knowing H at both the transmitter and receiver provides significant gains at low SNR [73] but small gains at high SNR ([95], Figure 4) as compared to knowing H at the receiver only. Furthermore, the reliability can be improved ([78], Figures 5–7). Significant gains are also possible for MIMO channels.*


**Remark** **72.**
*An alternative way to derive (Equation 272)–(Equation 275) is as follows. Define $\hat{P}=Pe^{\Delta }$ so for small P the capacity is*

$$\begin{array}{cc} C(P) & =\int_{\Delta }^{\infty }e^{-g}\log\left(1+g\;\hat{P}\right)dg\\ \; & =e^{1/\hat{P}}E_{1}\left(\frac{1}{\hat{P}}+\Delta \right)+e^{-\Delta }\log\left(1+\hat{P}\Delta \right)\\ \; & \approx P\;\left(1+\Delta \right)-\frac{1}{2}P^{2}e^{\Delta }\left(2+2\Delta +\Delta^{2}\right). \end{array}$$



### 8.3. Full CSIR, Partial CSIT

Consider noisy CSIT with
$$\begin{array}{c} \mathrm{Pr}\left[S_{T}=1\left(G\geq \Delta \right)\right]=\bar{\epsilon },\;\mathrm{Pr}\left[S_{T}\neq 1\left(G\geq \Delta \right)\right]=\epsilon . \end{array}$$
We begin with the most informative CSIR.

*$S_{R}=\sqrt{P(S_{T})}H$:* Proposition 2 gives the capacity
(277)$$\begin{array}{cc} C(P) & =\int_{0}^{\infty }e^{-g}\sum_{s_{T}}P\left(s_{T}|g\right)\log\left(1+g\;P(s_{T})\right)\;dg\\ \; & =\int_{0}^{\Delta }e^{-g}\left[\bar{\epsilon }\;\log\left(1+g\;P(0)\right)+\epsilon \;\log\left(1+g\;P(1)\right)\right]\;dg\\ \; & \;+\int_{\Delta }^{\infty }e^{-g}\left[\bar{\epsilon }\;\log\left(1+g\;P(1)\right)+\epsilon \;\log\left(1+g\;P(0)\right)\right]\;dg. \end{array}$$
It remains to optimize P(0), P(1) and Δ. The two equations for the Lagrange multiplier λ are
(278)$$\begin{array}{cc} \lambda \cdot P_{S_{T}}\left(0\right) & =\int_{0}^{\Delta }e^{-g}\cdot \frac{\bar{\epsilon }\;g}{1+gP(0)}dg+\int_{\Delta }^{\infty }e^{-g}\cdot \frac{\epsilon \;g}{1+gP(0)}dg \end{array}$$
(279)$$\begin{array}{cc} \lambda \cdot P_{S_{T}}\left(1\right) & =\int_{0}^{\Delta }e^{-g}\cdot \frac{\epsilon \;g}{1+gP(1)}dg+\int_{\Delta }^{\infty }e^{-g}\cdot \frac{\bar{\epsilon }\;g}{1+gP(1)}dg \end{array}$$
where $P_{S_{T}}\left(0\right)=\bar{\epsilon }-\left(\bar{\epsilon }-\epsilon \right)e^{-\Delta }$ and $P_{S_{T}}\left(1\right)=\epsilon +\left(\bar{\epsilon }-\epsilon \right)e^{-\Delta }$. The rates are shown in Figure 8.

For fixed Δ and large *P*, we have 1/λ≈P(0)≈P(1)≈P and approach the capacity (Equation 269) without CSIT. In contrast, for small *P* we may use similar steps as for (Equation 183) and (Equation 184). Observe the following for (Equation 278) and (Equation 279):both P(0) and P(1) decrease as λ increases;the maximal λ in (Equation 278) is obtained with P(0)=0; this value is
(280)$$\begin{array}{c} E\left[G|S_{T}=0\right]=\frac{\bar{\epsilon }-\left(\bar{\epsilon }-\epsilon \right)\left(1+\Delta \right)\;e^{-\Delta }}{P_{S_{T}}\left(0\right)} \end{array}$$the maximal λ in (Equation 279) is obtained with P(1)=0; this value is
(281)$$\begin{array}{c} E\left[G|S_{T}=1\right]=\frac{\epsilon +\left(\bar{\epsilon }-\epsilon \right)\left(1+\Delta \right)\;e^{-\Delta }}{P_{S_{T}}\left(1\right)}. \end{array}$$
Thus, if $E\left[G|S_{T}=0\right]<E\left[G|S_{T}=1\right]$ and 0≤ϵ<1/2, then for *P* below some threshold we have P(0)=0, $P\left(1\right)=P/P_{S_{T}}\left(1\right)$ and the capacity is
(282)$$\begin{array}{c} C\left(P\right)=\int_{0}^{\Delta }e^{-g}\;\epsilon \;\log\left(1+\frac{g\;P}{P_{S_{T}}\left(1\right)}\right)\;dg+\int_{\Delta }^{\infty }e^{-g}\;\bar{\epsilon }\;\log\left(1+\frac{g\;P}{P_{S_{T}}\left(1\right)}\right)\;dg. \end{array}$$
We compute $C^{\prime }\left(0\right)=E\left[G|S_{T}=1\right]$ which is given by (Equation 281) so that $1\leq C^{\prime }\left(0\right)\leq 1+\Delta$, as expected from (Equation 274). For example, for ϵ=0.1 and Δ=1 we have $C^{\prime }\left(0\right)\approx 1.75$ and therefore the minimal $E_{b}/N_{0}$ is approximately −4.01 dB.

The best Δ is the unique solution $\hat{\Delta }$ of the equation
(283)$$\begin{array}{c} e^{-\Delta }=\frac{\epsilon }{\bar{\epsilon }-\epsilon }\left(\Delta -1\right) \end{array}$$
and the result is $C^{\prime }\left(0\right)=\hat{\Delta }\geq 1$. We have the simple bounds
(284)$$\begin{array}{c} 1+\frac{1}{2}\log\left(\frac{1}{\epsilon }-2\right)\leq C^{\prime }\left(0\right)\leq 1+\frac{1}{e}\left(\frac{1}{\epsilon }-2\right) \end{array}$$
where the left inequality follows by taking logarithms and using log(Δ−1)≤Δ−2, and the right inequality follows by using $e^{-\Delta }\leq e^{-1}$ in (Equation 283). For example, for ϵ→0 we have $C^{\prime }\left(0\right)\to \infty$, and for ϵ→1/2 we have $C^{\prime }\left(0\right)\to 1$.

*$S_{R}=H$:* For the less informative CSIR, one may use (Equation 191) and (Equation 193) to compute I(A;Y|H). The reverse model GMI requires $\mathrm{Var}\left[U|Y,S_{R}\right]$, which can be computed by simulation; see Section C.2. Again, however, optimizing the powers seems difficult. We instead focus on the forward model GMI of Corollary 1, which is
(285)$$\begin{array}{c} I_{1}\left(A;Y|H\right)=\int_{0}^{\infty }e^{-g}\log\left(1+\mathrm{SNR}(g)\right)\;dg \end{array}$$
where
(286)$$\begin{array}{c} \mathrm{SNR}\left(g\right)=\frac{g{\tilde{P}}_{T}\left(g\right)}{1+g\;\epsilon \;\bar{\epsilon }{\left(\sqrt{P(0)}-\sqrt{P(1)}\right)}^{2}} \end{array}$$
and
(287)$$\begin{array}{cc} {\tilde{P}}_{T}\left(g\right) & =\left\{\begin{array}{cc} {\left(\bar{\epsilon }\sqrt{P(0)}+\epsilon \sqrt{P(1)}\right)}^{2}, & g<\Delta \\ {\left(\epsilon \sqrt{P(0)}+\bar{\epsilon }\sqrt{P(1)}\right)}^{2}, & g\geq \Delta . \end{array}\right. \end{array}$$
It remains to optimize P(0), P(1) and Δ. Computing the derivatives seems complicated, so we use numerical optimization for fixed Δ=1 as in Figure 8. The results are shown in Figure 9. For fixed Δ and large *P*, it is best to choose P(0)≈P(1) so that SNR(g)≈gP and we approach the rate of no CSIT. For small *P*, however, the best P(0) is no longer zero and $C^{\prime }\left(0\right)$ is smaller than (Equation 281).

### 8.4. Partial CSIR, Full CSIT

Consider $S_{T}=H$ and suppose we choose the X(h) to be jointly CSCG with variances $E\left[{\left|X\left(h\right)\right|}^{2}\right]=P\left(h\right)$ and correlation coefficients
$$\begin{array}{c} \rho \left(h,h^{\prime }\right)=\frac{E\left[X\left(h\right)X{\left(h^{\prime }\right)}^{*}\right]}{\sqrt{P\left(h\right)P\left(h^{\prime }\right)}} \end{array}$$
and where $E\left[P(H)\right]\leq P$. We then have
$$\begin{array}{c} p\left(y|s_{R}\right)=\int_{C}p\left(h|s_{R}\right)\frac{e^{-{\left|y\right|}^{2}/{\left(|h\right|}^{2}P\left(h\right)+1)}}{\pi (|h{|}^{2}P\left(h\right)+1)}\;dh. \end{array}$$
As in (Equation 97), $p(y|s_{R})$ and $h(Y|S_{R})$ depend only on the marginals of *A* and not on the $\rho (h,h^{\prime })$. We thus have the problem of finding the $\rho (h,h^{\prime })$ that minimize
$$\begin{array}{c} h\left(Y|A,S_{R}\right)=\int_{A}p\left(a\right)\;h\left(Y|S_{R},A=a\right)\;da. \end{array}$$
We will use fully-correlated X(h) as discussed in Section 6.5. We again consider $S_{R}=0$ and $S_{R}=1\left(G\geq t\right)$.

*$S_{R}=0$:* For the heuristic policies, the power (Equation 206) is
(288)$$\begin{array}{c} \hat{P}=\frac{P}{\Gamma \left(1+a,t\right)} \end{array}$$
and the rate (Equation 219) is
(289)$$\begin{array}{c} I_{1}\left(A;Y\right)=\log\left(1+\frac{P\;\Gamma {\left(\frac{3+a}{2},t\right)}^{2}}{\Gamma \left(1+a,t\right)+P\left[\Gamma \left(2+a,t\right)-\Gamma {\left(\frac{3+a}{2},t\right)}^{2}\right]}\right) \end{array}$$
where Γ(s,x) is the upper incomplete gamma function; see Section A.3. Moreover, the expression (Equation 220) is
(290)$$\begin{array}{cc} {\left.\frac{E_{b}}{N_{0}}\right|}_{\min}\; & =\frac{\Gamma \left(1+a,t\right)}{\Gamma {\left(\frac{3+a}{2},t\right)}^{2}}\cdot \log2. \end{array}$$
We remark that Γ(s,0)=Γ(s) where Γ(x) is the gamma function. We further have
$$\begin{array}{c} \begin{array}{cc} \Gamma \left(0,t\right)=E_{1}\left(t\right), & \Gamma \left(1,t\right)=e^{-t},\\ \Gamma \left(2,t\right)=e^{-t}\left(t+1\right), & \Gamma \left(3,t\right)=e^{-t}\left(t^{2}+2t+2\right). \end{array} \end{array}$$

For example, the TCP policy (a=0) has $\hat{P}=P\;e^{t}$. At low SNR, it turns out that the best choice is t=0.283 for which we have $\Gamma \left(1,t\right)/\Gamma {\left(3/2,t\right)}^{2}\approx 1.174$. The minimum $E_{b}/N_{0}$ in (222) is thus −0.90 dB. At high SNR, the best choice is t=0 so that (Equation 289) with $\Gamma \left(3/2,0\right)=\Gamma \left(3/2\right)=\sqrt{\pi }/2$ gives
(291)$$\begin{array}{c} I_{1}\left(A;Y\right)=\log\left(1+\frac{P\;\pi /4}{1+P(1-\pi /4)}\right). \end{array}$$
The TCP rate thus saturates at 2.22 bits per channel use; see the curve labeled “TCP, GMI, *K* = 1” in Figure 10.

The TMF policy (a=1) has $\hat{P}=P\;e^{t}/\left(t+1\right)$. The best choice is t=0 for which we have Γ(2)=1 and Γ(3)=2 and therefore (Equation 289) is
(292)$$\begin{array}{c} I_{1}\left(A;Y\right)=\log\left(1+\frac{P}{1+P}\right). \end{array}$$
The minimum $E_{b}/N_{0}$ in (222) is −1.59 dB, and at high SNR, the TMF rate saturates at 1 bit per channel use. The rates are shown as the curve labeled “TMF, GMI, *K* = 1” in Figure 10.

The TCI policy (a=−1) has $\hat{P}=P/E_{1}\left(t\right)$ and using $\Gamma \left(0,t\right)=E_{1}\left(t\right)$ and $\Gamma \left(1,t\right)=e^{-t}$ gives
(293)$$\begin{array}{c} I_{1}\left(A;Y\right)=\log\left(1+\frac{P}{e^{2t}\;E_{1}\left(t\right)+P\left(e^{t}-1\right)}\right). \end{array}$$
The minimum $E_{b}/N_{0}$ in (Equation 290) is
(294)$$\begin{array}{c} {\left.\frac{E_{b}}{N_{0}}\right|}_{\min}=E_{1}\left(t\right)\;e^{2t}\cdot \log2. \end{array}$$
Optimizing over *t* by taking derivatives (see (Equation 372) below), the best *t* satisfies the equation $2te^{t}E_{1}\left(t\right)=1$ which gives t≈0.61 and the minimal $E_{b}/N_{0}$ is approximately 0.194 dB. On the other hand, for large SNR, we may choose t=1/P and using $E_{1}\left(t\right)\approx \log\left(1/t\right)$ for small *t* gives
$$\begin{array}{c} I_{1}\left(A;Y\right)\approx \log\left(1+\frac{P}{1+\log P}\right). \end{array}$$
Since the pre-log is at most 1, the capacity grows with pre-log 1 for large *P*. We see that TMF is best at small *P* while TCI is best at large *P*. The rates are shown as the curve labeled “TCI, GMI, *K* = 1” in Figure 10.

The simple channel output of TCI permits further analysis. Using Remark 65, we compute the mutual information I(A;Y) by numerical integration; see the curve labeled “TCI, I(A;Y)” in Figure 10. We see that at high SNR, the TCI mutual information is larger than the GMI for TCP, TMF, and (of course) TCI. Moreover, as we show, the TCI mutual information can work well at low SNR.

Motivated by Section 7.3 and Figure 5, we again use the GMI (Equation 154) with K=2 and (Equation 65). We further choose $h_{1}=0$, $\sigma_{1}^{2}=\sigma_{2}^{2}=1$, and
$$\begin{array}{c} \bar{X}=\frac{\sqrt{\hat{P}}}{h_{2}}\;U,\;U∼\mathrm{CN}\left(0,1\right). \end{array}$$
The expression (Equation 154) simplifies to
(295)$$\begin{array}{c} I_{1}\left(A;Y\right)=\mathrm{Pr}\left[E_{2}\right]\left[\log\left(1+\hat{P}\right)+\frac{E\left[{\left|Y\right|}^{2}|E_{2}\right]}{1+\hat{P}}-E\left[\left.{\left|Y-\sqrt{\hat{P}}\;U\right|}^{2}\right|E_{2}\right]\right]. \end{array}$$
The GMI (Equation 295) exhibits interesting high and low SNR scaling by choosing the following thresholds $t,t_{R}$.

For high SNR, we choose
(296)$$\begin{array}{c} t=P^{-\lambda }\;\mathrm{and}\;t_{R}={\hat{P}}^{\lambda_{R}} \end{array}$$
where 0<λ<1 and $0<\lambda_{R}<1$. As *P* increases, *t* decreases and Section B.4 shows that
(297)$$\begin{array}{c} \mathrm{Pr}\left[E_{2}\right]\to 1,\;\frac{E\left[{\left|Y\right|}^{2}|E_{2}\right]}{1+\hat{P}}\to 1,\;E\left[{\left|Y-\sqrt{\hat{P}}\;U\right|}^{2}|E_{2}\right]\to 1. \end{array}$$Inserting $\hat{P}=P/E_{1}\left(t\right)$, we thus have
(298)$$\begin{array}{c} \lim_{P\to \infty }\left[I_{1}\left(A;Y\right)-\log\left(1+\frac{P}{E_{1}\left(t\right)}\right)\right]=0. \end{array}$$We further have $E_{1}\left(t\right)\approx \lambda \log P$ by using (Equation 373) in Section A.2, and the high-SNR slope of the GMI matches the slope of logP but the additive gap to logP increases. The high SNR rates are shown as the curve labeled “TCI, GMI, *K* = 2” in Figure 10 for $\lambda =\lambda_{R}=0.4$.For low SNR, we choose
(299)$$\begin{array}{c} t=-\log\left(P/c\right)\;\mathrm{and}\;t_{R}=\hat{P} \end{array}$$
for a constant c>0. As *P* decreases, both *t* and $\hat{P}=P/E_{1}\left(t\right)$ increase and Section B.4 shows that
(300)$$\begin{array}{c} \mathrm{Pr}\left[E_{2}\right]\approx e^{-t-1},\;\frac{E\left[{\left|Y\right|}^{2}|E_{2}\right]}{1+2\hat{P}}\to 1,\;E\left[{\left|Y-\sqrt{\hat{P}}\;U\right|}^{2}|E_{2}\right]\to 1. \end{array}$$Using (Equation 374), we have $I_{1}\left(A;Y\right)\approx e^{-t-1}\log t$ which vanishes as *t* grows. But we also have
(301)$$\begin{array}{cc} \frac{E_{b}}{N_{0}}=\frac{P}{R}\log2 & \approx \frac{c\;e^{-t}\log2}{e^{-t-1}\log t}\approx \frac{c\;e\log2}{\log(-\log P)} \end{array}$$
which decreases (very slowly) as *P* decreases. The minimal $E_{b}/N_{0}$ is therefore −∞. The low SNR rates are shown as the curve labeled “TCI, GMI, *K* = 2” in Figure 11 for c=1.4.

**Figure 11 entropy-25-00728-f011:**
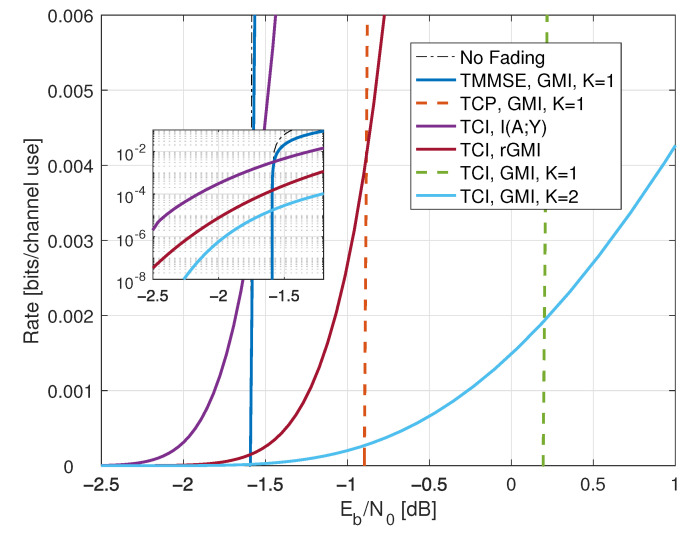
Low-SNR rates for Rayleigh fading with $S_{T}=H$ and $S_{R}=0$. The threshold *t* was optimized for the K=1 curves, while t=−log(P/1.4) for the I(A;Y), rGMI, and K=2 curves. The K=2 GMI uses $t_{R}=\hat{P}$. The TMF and TMMSE GMIs are indistinguishable for this range of rates.

Figure 11 shows that the TCI mutual information achieves a minimal $E_{b}/N_{0}$ below −1.59 dB. At $E_{b}/N_{0}=-2$ dB, we computed $I_{1}\left(A;Y\right)\approx 6\times 10^{-7}$ and $I\left(A;Y\right)\approx 3\times 10^{-4}$. The K=2 partition is thus useful to prove that TCI can achieve $E_{b}/N_{0}$ arbitrarily close to zero. Figure 11 also shows the reverse model GMI as the curve labeled “TCI, rGMI” which has the rate $I_{1}\left(A;Y\right)\approx 8\times 10^{-6}$ at $E_{b}/N_{0}=-2$ dB.

We compare the full CSIR and full CSIT rates. At high SNR, the GMI for $S_{R}=0$ achieves the same capacity pre-log as $S_{R}=H$. At low SNR, recall from (Equation 271) that with full CSIR/CSIT we have $E_{b}/N_{0}\approx \log\left(2\right)/\lambda$. To compare the rates for similar $E_{b}/N_{0}$, we set λ=logt, where *t* is as in (Equation 299) and c≈1. The TCI K=2 GMI without CSIR is approximately $e^{-t}\log t$ while the full CSIR rate (Equation 271) is approximately $e^{-\lambda }/\lambda \approx 1/\left(t\log\left(t\right)\right)$. Thus, the K=2 GMI with no CSIR is a fraction $te^{-t}\log{\left(t\right)}^{2}$ of the full CSIR capacity.

*$S_{R}=1\left(G\geq t\right)$:* The power in (Equation 206) is again (Equation 288) and the rate (Equation 224) is
(302)$$\begin{array}{c} I_{1}\left(A;Y|S_{R}\right)=e^{-t}\cdot \log\left(1+\frac{P\;e^{2t}\;\Gamma {\left(\frac{3+a}{2},t\right)}^{2}}{\Gamma \left(1+a,t\right)+P\left[e^{t}\;\Gamma \left(2+a,t\right)-e^{2t}\;\Gamma {\left(\frac{3+a}{2},t\right)}^{2}\right]}\right). \end{array}$$

Moreover, the expression (Equation 225) is
(303)$$\begin{array}{cc} \; & {\left.\frac{E_{b}}{N_{0}}\right|}_{\min}=\frac{\Gamma \left(1+a,t\right)}{e^{t}\cdot \Gamma {\left(\frac{3+a}{2},t\right)}^{2}}\cdot \log2 \end{array}$$
which is the same as (Equation 290) except for the factor $e^{t}$ in the denominator. This implies that the minimal $E_{b}/N_{0}$ can be improved for t>0.

The TCP, TMF, and TCI rates (Equation 302) are the respective
(304)$$\begin{array}{cc} \; & I_{1}\left(A;Y|S_{R}\right)=e^{-t}\log\left(1+\frac{P\;e^{2t}\;\Gamma {\left(\frac{3}{2},t\right)}^{2}}{e^{-t}+P\left[t+1-e^{2t}\;\Gamma {\left(\frac{3}{2},t\right)}^{2}\right]}\right) \end{array}$$
(305)$$\begin{array}{cc} \; & I_{1}\left(A;Y|S_{R}\right)=e^{-t}\log\left(1+\frac{P\;{\left(t+1\right)}^{2}}{e^{-t}\left(t+1\right)+P}\right) \end{array}$$
(306)$$\begin{array}{cc} \; & I_{1}\left(A;Y|S_{R}\right)=e^{-t}\log\left(1+\frac{P}{E_{1}\left(t\right)}\right). \end{array}$$

**Remark** **73.**
*As pointed out in Remark 68, the TCI GMI (306) is $I(A;Y|S_{R})$. One can also understand this by observing that the receiver knows $\sqrt{GP(G)}$ for all G. The mutual information is thus related to the rate (Equation 189) of Proposition 2.*


The minimal $E_{b}/N_{0}$ in (Equation 303) are the respective
(307)$$\begin{array}{cc} \; & {\left.\frac{E_{b}}{N_{0}}\right|}_{\min}=\frac{1}{e^{2t}\cdot \Gamma {\left(\frac{3}{2},t\right)}^{2}}\cdot \log2 \end{array}$$
(308)$$\begin{array}{cc} \; & {\left.\frac{E_{b}}{N_{0}}\right|}_{\min}=\frac{1}{t+1}\cdot \log2 \end{array}$$
(309)$$\begin{array}{cc} \; & {\left.\frac{E_{b}}{N_{0}}\right|}_{\min}=e^{t}\;E_{1}\left(t\right)\cdot \log2. \end{array}$$
The above expressions mean that, for all three policies, we can make the minimal $E_{b}/N_{0}$ as small as desired by increasing *t*. For example, for TCI, we can bound (see (A9) below)
(310)$$\begin{array}{c} \frac{1}{t+1}<e^{t}E_{1}\left(t\right)<\frac{1}{t}. \end{array}$$
TCI thus has a slightly larger (slightly worse) minimal $E_{b}/N_{0}$ than TMF for the same *t*, as discussed after (212).

For large *P*, the TCP rate (Equation 304) is optimized by t≈0.163 and the rate saturates at ≈2.35 bits per channel use. The TMF rate (305) is optimized with t=0, and the rate saturates at 1 bit per channel use. For the TCI rate (306), we again choose t=1/P and use $E_{1}\left(t\right)\approx \log\left(1/t\right)$ for small *t* to show that the capacity grows with pre-log 1:$$\begin{array}{c} I_{1}\left(A;Y|S_{R}\right)\approx \log\left(1+\frac{P}{\log P}\right). \end{array}$$
Again, TMF is best at small *P* while TCI is best at large *P*.

**Remark** **74.**
*Comparing (Equation 298) and (Equation 306), the $S_{R}=0$, K=2, TCI GMI in (Equation 295) approaches the $S_{R}=1\left(G\geq t\right)$ mutual information $I(A;Y|S_{R})$ in (Equation 306) at high SNR.*


*Optimal Policy:* Consider now the optimal power control policy. Suppose first that $S_{R}=0$ for which Theorem 2 gives the TMMSE policy with t=0:(311)$$\begin{array}{cc} \sqrt{P(h)} & =\frac{\alpha |h|}{\beta +{\left|h\right|}^{2}}. \end{array}$$
For Rayleigh fading, we thus have (see (A13) below)
(312)$$\begin{array}{c} P=\int_{0}^{\infty }e^{-g}\frac{\alpha^{2}g}{{\left(\beta +g\right)}^{2}}\;dg=\alpha^{2}\left[\left(\beta +1\right)e^{\beta }E_{1}\left(\beta \right)-1\right] \end{array}$$
with the two expressions (see (Equation 379) and (A14) below)
(313)$$\begin{array}{cc} \tilde{P} & =\int_{0}^{\infty }e^{-g}\frac{\alpha^{2}g}{\beta +g}\;dg=\alpha^{2}{\left[1-\beta e^{\beta }E_{1}\left(\beta \right)\right]}^{2} \end{array}$$
(314)$$\begin{array}{cc} E\left[GP(H)\right] & =\int_{0}^{\infty }e^{-g}\frac{\alpha^{2}g^{2}}{{\left(\beta +g\right)}^{2}}\;dg=\alpha^{2}\left[1+\beta -\beta \left(\beta +2\right)e^{\beta }E_{1}\left(\beta \right)\right]. \end{array}$$
Given *P* and β, we may compute $\alpha^{2}$ from (Equation 312). We then search for the optimal β for fixed *P*. The rates are shown as the curve labeled “TMMSE, GMI, *K* = 1” in Figure 10 and Figure 11 and we see that the TMMSE strategy has the best K=1 rates.

Consider next $S_{R}=1\left(G\geq t\right)$ and the TMMSE policy. We compute (see (A13) below)
(315)$$\begin{array}{c} P=\int_{t}^{\infty }e^{-g}\frac{\alpha^{2}g}{{\left(\beta +g\right)}^{2}}\;dg=\alpha^{2}\left[\left(\beta +1\right)e^{\beta }E_{1}\left(t+\beta \right)-e^{-t}\frac{\beta }{t+\beta }\right] \end{array}$$
and (see (Equation 379) and (A14) below)
(316)$$\begin{array}{cc} \sqrt{\tilde{P}\left(1\right)}\; & =\int_{t}^{\infty }\frac{e^{-g}}{e^{-t}}\frac{\alpha g}{\beta +g}\;dg=\alpha \left[1-\beta e^{t+\beta }E_{1}\left(t+\beta \right)\right] \end{array}$$
(317)$$\begin{array}{cc} E\left[{\left|Y\right|}^{2}|S_{R}=1\right] & =\int_{t}^{\infty }\frac{e^{-g}}{e^{-t}}\left(1+\frac{\alpha^{2}g^{2}}{{\left(\beta +g\right)}^{2}}\right)\;dg\\ \; & =1+\alpha^{2}\left[1+\frac{\beta^{2}}{t+\beta }-\beta \left(\beta +2\right)e^{t+\beta }E_{1}\left(t+\beta \right)\right]. \end{array}$$
We optimize as for the $S_{R}=0$ case: given *P*, β, *t*, we compute $\alpha^{2}$ from (Equation 315). We then search for the optimal β for fixed *P* and *t*. The optimal *t* is approximately a factor of 1.1 smaller than for the TCI policy. The rates are shown in Figure 12 as the curve labeled “TMMSE, GMI”.

### 8.5. Partial CSIR, CSIT@ R

Suppose $S_{R}$ is defined by (see (Equation 172))
$$\begin{array}{c} H=\sqrt{\bar{\epsilon }}\;S_{R}+\sqrt{\epsilon }\;Z_{R} \end{array}$$
where 0≤ϵ≤1 and $S_{R},Z_{R}$ are independent with distribution CN(0,1). We further consider the CSIT $S_{T}={\left|S_{R}\right|}^{2}$.

The reverse model GMI again requires $\mathrm{Var}\left[U|Y,S_{R}\right]$, which can be computed by simulation; see Section C.4. However, as in Section 7.4 and Section 8.3, optimizing the powers seems difficult, and we instead focus on forward models. The expressions (Equation 235) and (236) are
(318)$$\begin{array}{c} \tilde{g}\left(s_{R}\right)=\bar{\epsilon }\;s_{T},\;{\tilde{\sigma }}^{2}\left(s_{R}\right)=\epsilon . \end{array}$$
The GMI (Equation 237) of Theorem 3 is
(319)$$\begin{array}{cc} I_{1}\left(X;Y|S_{R}\right) & =\int_{\lambda /\bar{\epsilon }}^{\infty }e^{-s_{T}}\log\left(1+\frac{\bar{\epsilon }\;s_{T}P\left(s_{T}\right)}{1+\epsilon \;P(s_{T})}\right)\;ds_{T} \end{array}$$
where the power control policy $P(s_{T})$ is given by (Equation 245). The parameter λ is chosen so that $E\left[P(S_{T})\right]=P$. For example, for ϵ→0 we recover the waterfilling solution (Equation 181). Figure 13 shows the quadratic and conventional waterfilling rates, which lie almost on top of each other. For example, the inset shows the rates for ϵ=0.2 and a small range of $E_{b}/N_{0}$.

## 9. Channels with In-Block Feedback

This section generalizes Shannon’s model described in Section 4.1 to include block fading with in-block feedback. For example, the model lets one include delay in the CSIT and permits many other generalizations for network models [22].

### 9.1. Model and Capacity

The problem is specified by the FDG in Figure 14. The model has a message *M*, and the channel input and output strings
$$\begin{array}{c} X_{i}^{L}=\left(X_{i1},\ldots ,X_{iL}\right),\;Y_{i}^{L}=\left(Y_{i1},\ldots ,Y_{iL}\right) \end{array}$$
for blocks i=1,…,n. The channel is specified by a string $S_{H}^{n}=\left(S_{H1},\ldots ,S_{Hn}\right)$ of i.i.d. hidden channel states. The CSIR $S_{Ri\ell }$ is a (possibly noisy) function of $S_{Hi}$ for all *i* and *ℓ*. The receiver sees the channel outputs (see (Equation 159))
(320)$$\begin{array}{cc} \; & \left(Y_{i\ell },S_{Ri\ell }\right)=\left(f_{\ell }\left(X_{i}^{\ell },S_{Hi},Z_{i}^{L}\right),S_{Ri\ell }\right) \end{array}$$
for some functions $f_{\ell }\left(\cdot \right)$, ℓ=1,…,L. Observe that the $X_{i}^{\ell }$ influence the $Y_{i\ell }$ in a causal fashion. The random variables $M,S_{H1},\ldots ,S_{Hn},Z_{1}^{L},\ldots ,Z_{n}^{L}$ are mutually independent.

We now permit past channel symbols to influence the CSIT; see Section 1.2. Suppose the CSIT has the form
(321)$$\begin{array}{c} S_{Ti\ell }=f_{T\ell }\left(S_{Hi},X_{i}^{\ell -1},Y_{i}^{\ell -1}\right) \end{array}$$
for some function $f_{T\ell }\left(.\right)$ and for all *i* and *ℓ*. The motivation for (Equation 321) is that useful CSIR may not be available until the end of a block or even much later. In the meantime, the receiver can, e.g., quantize the $Y_{i}^{\ell -1}$ and transmit the quantization bits via feedback. This lets one study fast power control and beamforming without precise knowledge of the channel coefficients.

Define the string of past and current states as
(322)$$\begin{array}{c} s_{T}^{i\ell }=\left(s_{T1}^{L},\ldots ,s_{T(i-1)}^{L},s_{Ti}^{\ell }\right). \end{array}$$
The channel input at time iℓ is $X(s_{T}^{i\ell })$ and the adaptive codeword $A^{nL}$ is defined by the ordered lists
(323)$$\begin{array}{cc} A_{i\ell } & =\left[X\left(s_{T}^{i\ell }\right),\;\forall \;s_{T}^{i\ell }\right] \end{array}$$
for 1≤i≤n and 1≤ℓ≤L. The adaptive codeword $A^{nL}$ is a function of *M* and is thus independent of $S_{H}^{n}$ and $S_{R}^{nL}$.

The model under consideration is a special case of the channels introduced in ([22], Section V). However, the model in [22] has transmission and reception begin at time ℓ=2 rather than ℓ=1. To compare the theory, one must thus shift the time indexes by 1 unit and increase *L* to L+1. The capacity for our model is given by ([22], Theorem 2) which we write as
(324)$$\begin{array}{c} C\overset{\left(a\right)}{=}\max_{A^{L}}\frac{1}{L}I\left(A^{L};Y^{L},S_{R}^{L}\right)\overset{\left(b\right)}{=}\max_{A^{L}}\frac{1}{L}I\left(A^{L};Y^{L}|S_{R}^{L}\right). \end{array}$$
where (a) follows by normalizing by *L* rather than L+1, and step (b) follows by the independence of $A^{L}$ and $S_{R}^{L}$.

### 9.2. GMI for Scalar Channels

We will study scalar block fading channels; extensions to vector channels follow as described in Section 4.4. Let $\underline{Y}={\left[Y_{1},\ldots ,Y_{L}\right]}^{T}$ be the vector form of $Y^{L}$ and similarly for other strings with *L* symbols. The GMI with parameter *s* is
(325)$$\begin{array}{c} I_{s}\left(A^{L};Y^{L}|S_{R}^{L}\right)=E\left[\log\frac{q{\left(\underline{Y}|\underline{A},{\underline{S}}_{R}\right)}^{s}}{q(\underline{Y}|{\underline{S}}_{R})}\right] \end{array}$$
*Reverse Model:* For the reverse model, let $\underline{A}$ be a column vector that stacks the $X_{\ell }\left(s_{T}^{\ell }\right)$ for all $s_{T}^{\ell }$ and *ℓ*. Consider a reverse density as in (Equation 105):$$\begin{array}{c} q\left(a^{L}|y^{L}\right)=\frac{\exp\left(-\underline{z}{\left(\underline{y},{\underline{s}}_{R}\right)}^{†}\;Q_{\underline{A}|\underline{Y}=\underline{y},{\underline{S}}_{R}={\underline{s}}_{R}}^{-1}\;\underline{z}\left(\underline{y},{\underline{s}}_{R}\right)\right)}{\pi^{N}\det Q_{\underline{A}|\underline{Y}=\underline{y},{\underline{S}}_{R}={\underline{s}}_{R}}} \end{array}$$
where
$$\begin{array}{c} \underline{z}\left(\underline{y},{\underline{s}}_{R}\right)=\underline{a}-E\left[\underline{A}|\underline{Y}=\underline{y},{\underline{S}}_{R}={\underline{s}}_{R}\right]. \end{array}$$
Using the forward model $q\left(y^{L}|a^{L}\right)=q\left(a^{L}|y^{L}\right)/p\left(a^{L}\right)$, the GMI with s=1 becomes
(326)$$\begin{array}{c} I_{1}\left(A^{L};Y^{L},S_{R}^{L}\right)=E\left[\log\frac{\det Q_{\underline{A}}}{\det Q_{\underline{A}|\underline{Y},{\underline{S}}_{R}}}\right]. \end{array}$$
To simplify, consider adaptive symbols as in (Equation 89) (cf. (Equation 107)):(327)$$\begin{array}{c} X_{\ell }\left(S_{T}^{\ell }\right)=\sqrt{P_{\ell }\left(S_{T}^{\ell }\right)}\;e^{j\phi_{\ell }\left(S_{T}^{\ell }\right)}\;U_{\ell } \end{array}$$
where $\underline{U}∼\mathrm{CN}\left(\underline{0},I\right)$. In other words, consider a conventional codebook represented by the $U_{\ell }$ and adapt the power and phase based on the available CSIT. The mutual information becomes $I\left(A^{L};Y^{L},S_{R}^{L}\right)=I\left(U^{L};Y^{L},S_{R}^{L}\right)$ (cf. (Equation 96)) and the GMI with s=1 is (cf. (Equation 108))
(328)$$\begin{array}{c} I_{1}\left(A^{L};Y^{L}|S_{R}^{L}\right)=E\left[-\log\det Q_{\underline{U}\left|\underline{Y},{\underline{S}}_{R}\right.}\right]. \end{array}$$
In fact, one may also consider choosing $U_{\ell }=U$ for all *ℓ* in which case we compute (cf. (Equation 139))
(329)$$\begin{array}{c} I_{1}\left(A^{L};Y^{L}|S_{R}^{L}\right)=E\left[-\log\mathrm{Var}\left[U|\underline{Y},{\underline{S}}_{R}\right]\right]. \end{array}$$
*Forward Model:* Consider the following forward model (cf. (Equation 111) and (Equation 141)):(330)$$\begin{array}{c} q\left(\underline{y}|\underline{a},{\underline{s}}_{R}\right)=\frac{\exp\left(-\underline{z}{\left({\underline{s}}_{R}\right)}^{†}Q_{\underline{Z}}{\left({\underline{s}}_{R}\right)}^{-1}\underline{z}\left({\underline{s}}_{R}\right)\right)}{\pi^{L}\det Q_{\underline{Z}}\left({\underline{s}}_{R}\right)}. \end{array}$$
with
$$\begin{array}{c} \underline{z}\left({\underline{s}}_{R}\right)=\underline{y}-H\left({\underline{s}}_{R}\right)\;\underline{\bar{x}}\left({\underline{s}}_{R}\right) \end{array}$$
and where similar to (Equation 142) we define
(331)$$\begin{array}{c} \underline{\bar{X}}\left({\underline{s}}_{R}\right)=\sum_{{\underline{s}}_{T}}W\left({\underline{s}}_{T},{\underline{s}}_{R}\right)\;\underline{X}\left({\underline{s}}_{T}\right) \end{array}$$
where the $W({\underline{s}}_{T},{\underline{s}}_{R})$ are L×L complex matrices. Note that
(332)$$\begin{array}{c} \underline{X}\left({\underline{s}}_{T}\right)={\left[X_{1}\left(s_{T1}\right),X_{2}\left(s_{T}^{2}\right),\ldots ,X_{2}\left(s_{T}^{L}\right)\right]}^{T} \end{array}$$
so $X_{\ell }$ is a function of $A^{L}$ and $S_{T}^{\ell }$, ℓ=1,…,L.

We have the following generalization of Lemma 4 (see also Theorem 1) where the novelty is that $S_{T}$ is replaced with ${\underline{S}}_{T}$. Define $\underline{U}\left({\underline{s}}_{T}\right)∼\mathrm{CN}\left(\underline{0},I\right)$ and $\underline{X}\left({\underline{s}}_{T}\right)=Q_{\underline{X}\left({\underline{s}}_{T}\right)}^{1/2}\;\underline{U}\left({\underline{s}}_{T}\right)$ for all ${\underline{s}}_{T}$.

**Theorem** **4.**
*A GMI (Equation 325) for the scalar block fading channel $p(y^{L}|a^{L},s_{R}^{L})$, an adaptive codeword $A^{L}$ with jointly CSCG entries, the auxiliary model (Equation 330), and with fixed $Q_{X({\underline{s}}_{T})}$ is*

(333)
$$\begin{array}{c} I_{1}\left(A^{L};Y^{L}|S_{R}^{L}\right)=E\left[\log\left(\frac{\det Q_{\underline{Y}}\left({\underline{S}}_{R}\right)}{\det\left(Q_{\underline{Y}}\left({\underline{S}}_{R}\right)-\tilde{D}\left({\underline{S}}_{R}\right)\;\tilde{D}{\left({\underline{S}}_{R}\right)}^{†}\right)}\right)\right]. \end{array}$$

*where*

(334)
$$\begin{array}{cc} Q_{\underline{Y}}\left({\underline{s}}_{R}\right) & =E\left[\left.\underline{Y}\;{\underline{Y}}^{†}\right|{\underline{S}}_{R}={\underline{s}}_{R}\right] \end{array}$$

*and for M×M unitary $V_{R}\left({\underline{s}}_{T},{\underline{s}}_{R}\right)$ the matrix $\tilde{D}\left({\underline{s}}_{R}\right)$ is*

(335)
$$\begin{array}{c} E\left[\left.U_{T}\left({\underline{S}}_{T},{\underline{s}}_{R}\right)\;\Sigma \left({\underline{S}}_{T},{\underline{s}}_{R}\right)\;V_{R}{\left({\underline{S}}_{T},{\underline{s}}_{R}\right)}^{†}\right|{\underline{S}}_{R}={\underline{s}}_{R}\right] \end{array}$$

*and $U_{T}\left({\underline{s}}_{T},{\underline{s}}_{R}\right)$ and $\Sigma ({\underline{s}}_{T},{\underline{s}}_{R})$ are N×N unitary and N×M rectangular diagonal matrices, respectively, of the SVD*

(336)
$$\begin{array}{c} E\left[\left.\underline{Y}\;{\underline{U}\left({\underline{s}}_{T}\right)}^{†}\right|{\underline{S}}_{T}={\underline{s}}_{T},{\underline{S}}_{R}={\underline{s}}_{R}\right]=U_{T}\left({\underline{s}}_{T},{\underline{s}}_{R}\right)\;\Sigma \left({\underline{s}}_{T},{\underline{s}}_{R}\right)\;V_{T}{\left({\underline{s}}_{T},{\underline{s}}_{R}\right)}^{†} \end{array}$$

*for all ${\underline{s}}_{T}$, ${\underline{s}}_{R}$ and the $V_{T}\left({\underline{s}}_{T},{\underline{s}}_{R}\right)$ are M×M unitary matrices. One may maximize (Equation 333) over the unitary $V_{R}\left({\underline{s}}_{T},{\underline{s}}_{R}\right)$.*


Suppose next that the actual channel is $\underline{Y}=H\underline{X}+\underline{Z}$ where $\underline{Z}∼\mathrm{CN}\left(\underline{0},I\right)$. The extension of (Equation 136) and (Equation 168) to block fading channels with CSIR is
(337)$$\begin{array}{c} I_{1}\left(A^{L};Y^{L}|S_{R}^{L}\right)=\sum_{\ell =1}^{L}E\left[\log\left(1+\frac{{\tilde{P}}_{\ell }\left({\underline{S}}_{R}\right)}{1+E\left[GP_{\ell }\left(S_{T}^{\ell }\right)|{\underline{S}}_{R}\right]-{\tilde{P}}_{\ell }\left({\underline{S}}_{R}\right)}\right)\right] \end{array}$$
where (cf. (Equation 166) and (167))
$$\begin{array}{cc} {\tilde{P}}_{\ell }\left({\underline{s}}_{R}\right) & =E{\left[\;\left|E\left[\left.H\sqrt{P_{\ell }\left(S_{T}^{\ell }\right)}\;\right|S_{T}^{\ell },{\underline{S}}_{R}={\underline{s}}_{R}\right]\right|\;\right]}^{2}\\ E\left[|Y_{\ell }{|}^{2}|{\underline{S}}_{R}={\underline{s}}_{R}\right]\; & =1+E\left[GP_{\ell }\left(S_{T}^{\ell }\right)|{\underline{S}}_{R}={\underline{s}}_{R}\right]. \end{array}$$

### 9.3. CSIT@ R

Continuing as in Section 5.2, suppose the CSIT in (Equation 321) can be written by replacing $S_{Hi}$ with $S_{Ri}^{\ell }$ for all *i* and *ℓ*:(338)$$\begin{array}{c} S_{Ti\ell }=f_{T\ell }\left(S_{Ri}^{\ell },X_{i}^{\ell -1},Y_{i}^{\ell -1}\right). \end{array}$$
The capacity (Equation 324) then simplifies to a directed information. To see this, expand the mutual information in (Equation 324) as
(339)$$\begin{array}{cc} I(A^{L};Y^{L}|S_{R}^{L})\; & \overset{\left(a\right)}{=}\sum_{\ell =1}^{L}I\left(A^{L},X^{\ell };Y_{\ell }|S_{R}^{L},Y^{\ell -1}\right)\\ \; & \overset{\left(b\right)}{=}\sum_{\ell =1}^{L}I\left(X^{\ell };Y_{\ell }|S_{R}^{L},Y^{\ell -1}\right) \end{array}$$
where step (a) follows because $X^{\ell }$ is a function of $A^{L}$ and $S_{T}^{\ell }$ in (Equation 338), and step (b) follows by the Markov chains
(340)$$\begin{array}{cc} \; & A^{L}-\left[S_{R}^{L},X^{\ell },Y^{\ell -1}\right]-Y_{\ell }. \end{array}$$
The capacity is therefore (see the definition (Equation 27))
(341)$$\begin{array}{c} C=\max_{X_{\ell }\left(S_{T}^{\ell }\right),\;\ell =1,\ldots ,L}\frac{1}{L}I\left(X^{L}\to Y^{L}|S_{R}^{L}\right). \end{array}$$
The maximization in (Equation 341) under a cost constraint becomes a constrained maximization for which $E\left[c(X^{L},Y^{L})\right]\leq LP$ for some cost function c(·).

**Remark** **75.**
*As outlined at the end of Section 9.1, the capacity (Equation 341) is a special case of the theory in ([22], Equation (Equation 48)). To see this, define the extended and time-shifted strings*

$$\begin{array}{c} {\hat{A}}^{L+1}=\left(0,A^{L}\right),\;\;{\hat{X}}^{L+1}=\left(0,X^{L}\right),\;\;{\hat{Y}}^{L+1}=\left(0,Y^{L}\right). \end{array}$$

*Since $A^{L}$ and $S_{R}^{L}$ are independent, one may expand (Equation 339) as*

(342)
$$\begin{array}{cc} I(A^{L};Y^{L}|S_{R}^{L}) & =I(A^{L}\;;\;\left(S_{R2},\ldots ,S_{RL},0\right),Y^{L}|S_{R1})\\ \; & \overset{\left(a\right)}{=}\sum_{\ell =1}^{L}I\left(A^{L},X^{\ell }\;;\;S_{R(\ell +1)},Y_{\ell }|S_{R}^{\ell },Y^{\ell -1}\right)\\ \; & \overset{\left(b\right)}{=}\sum_{\ell =1}^{L}I\left(X^{\ell }\;;\;S_{R(\ell +1)},Y_{\ell }|S_{R}^{\ell },Y^{\ell -1}\right)\\ \; & =\sum_{\ell =2}^{L+1}I\left({\hat{X}}^{\ell }\;;\;S_{R\ell },{\hat{Y}}_{\ell }|S_{R}^{\ell -1},{\hat{Y}}^{\ell -1}\right) \end{array}$$

*where step (a) follows because $X^{\ell }$ is a function of $A^{L}$ and $S_{T}^{\ell }$ in (Equation 338), and where $S_{R(L+1)}=0$, and step (b) follows by the Markov chains*

(343)
$$\begin{array}{cc} \; & A^{L}-\left[X^{\ell },Y^{\ell -1},S_{R}^{\ell }\right]-\left[Y_{\ell },S_{R(\ell +1)}\right]. \end{array}$$

*The expression (Equation 342) is the desired directed information*

(344)
$$\begin{array}{c} I\left(A^{L};Y^{L},S_{R}^{L}\right)=I\left({\hat{X}}^{L+1}\to {\hat{Y}}^{L+1},S_{R}^{L+1}\right). \end{array}$$



**Remark** **76.**
*Consider the basic CSIT model*

(345)
$$\begin{array}{c} S_{Ti\ell }=f_{T}\left(S_{Ri\ell }\right) \end{array}$$

*for some function $f_{T}\left(\cdot \right)$ and for ℓ=1,…,L and i=1,…,n. This model was studied in ([103], Section III.C) and its capacity is given as (see ([103], Equation (Equation 35) with Equation (Equation 13)))*

(346)
$$\begin{array}{cc} C & =\max_{X_{\ell }\left(S_{T}^{\ell }\right),\;\ell =1,\ldots ,L}\frac{1}{L}I\left(X^{L};Y^{L}|S_{R}^{L},S_{T}^{L}\right). \end{array}$$

*To see that (Equation 346) is a special case of (Equation 341), observe that*

(347)
$$\begin{array}{cc} I(X^{L}\to Y^{L}|S_{R}^{L}) & \overset{\left(a\right)}{=}\sum_{\ell =1}^{L}I\left(X^{\ell };Y_{\ell }|S_{R}^{L},S_{T}^{L},Y^{\ell -1}\right)\\ \; & \overset{\left(b\right)}{=}\sum_{\ell =1}^{L}I\left(X^{L};Y_{\ell }|S_{R}^{L},S_{T}^{L},Y^{\ell -1}\right) \end{array}$$

*where step (a) follows by (Equation 339), and step (b) follows by the Markov chains*

(348)
$$\begin{array}{c} \left[X_{\ell +1},\ldots ,X_{L}\right]-\left[S_{R}^{L},S_{T}^{L},Y^{\ell -1},X^{\ell }\right]-Y_{\ell }. \end{array}$$

*The expression (Equation 347) gives (Equation 346). Related results are available in ([10], Section III) and [104,105].*


**Remark** **77.**
*The capacity (Equation 341) has only $S_{R}^{L}$ in the conditioning while (Equation 346) has both $S_{R}^{L}$ and $S_{T}^{L}$ in the conditioning. This subtle difference is due to permitting $X^{\ell -1}$ to influence the $S_{T\ell }$ in (Equation 338), and it complicates the analysis. On the other hand, if we remove only $X^{\ell -1}$ from (Equation 338) then the receiver knows $S_{T\ell }$ at time ℓ and the capacity (Equation 341) can be written as (see the definition (28))*

(349)
$$\begin{array}{c} C=\max_{X_{\ell }\left(S_{T}^{\ell }\right),\;\ell =1,\ldots ,L}\frac{1}{L}I\left(X^{L}\to Y^{L}∥S_{T}^{L}|S_{R}^{L}\right). \end{array}$$

*We treat such a model in Section 9.7 below.*


### 9.4. Fading Channels with AWGN

The expression (Equation 341) is valid for general statistics. We next specialize to the block-fading AWGN model
(350)$$\begin{array}{c} Y_{\ell }=HX_{\ell }+Z_{\ell } \end{array}$$
where ℓ=1,…,L, $Z^{L}∼\mathrm{CN}\left(\underline{0},I\right)$, and $\left(H,S_{R}^{L}\right)$, $A^{L}$, $Z^{L}$ are mutually independent. Consider the power constraint
(351)$$\begin{array}{c} \sum_{\ell =1}^{L}E\left[P_{\ell }\left(S_{T}^{\ell }\right)\right]\leq LP \end{array}$$
where $P_{\ell }\left(s_{T}^{\ell }\right)=E\left[|X_{\ell }\left(s_{T}^{\ell }\right){|}^{2}\right]$. The optimization of (Equation 341) under the constraint (Equation 351) is usually intractable, and we again desire expressions with log(1+SNR) terms to obtain insight.

*Capacity Upper Bound:* Using similar steps as in (Equation 162), we have
(352)$$\begin{array}{cc} I(A^{L};Y^{L}|S_{R}^{L}) & \leq I(A^{L};Y^{L},H\;|\;S_{R}^{L})\\ \; & =\sum_{\ell =1}^{L}I\left(A^{L};Y_{\ell }\;|\;S_{R}^{L},H,Y^{\ell -1}\right)\\ \; & \leq \sum_{\ell =1}^{L}\left[h\left(Y_{\ell }|S_{R}^{L},H,Y^{\ell -1}\right)-h\left(Z_{\ell }\right)\right]\\ \; & \overset{\left(a\right)}{\leq }\sum_{\ell =1}^{L}E\left[\log\left(1+E\left[GP_{\ell }\left(S_{T}^{\ell }\right)|S_{R}^{L},H,Y^{\ell -1}\right]\;\right)\right] \end{array}$$
where $G={\left|H\right|}^{2}$ and step (a) follows by (Equation 163). However, CSCG inputs do not necessarily maximize the RHS of (Equation 352) because the inputs affect the CSIT.

**Remark** **78.**
*The expectation inside the logarithm in (Equation 352) becomes $GP_{\ell }\left(S_{T}^{\ell }\right)$ if $S_{T}^{\ell }$ is a function of $S_{R}^{L},H,Y^{\ell -1}$; see (Equation 161), Remark 77, and Proposition 3 below.*


*Achievable Rates:* Deriving achievable rates is more subtle than in Section 6. Consider the CSIT model (Equation 338) where for each block, we have
$$\begin{array}{c} S_{T\ell }=f_{T\ell }\left(H,X^{\ell -1},Y^{\ell -1}\right) \end{array}$$
for all *ℓ*. The capacity (Equation 341) is
(353)$$\begin{array}{cc} C(P) & =\max_{X_{\ell }\left(S_{T}^{\ell }\right),\;\ell =1,\ldots ,L}\frac{1}{L}I\left(X^{L}\to Y^{L}|H\right) \end{array}$$
(354)$$\begin{array}{cc} \; & =\max_{X_{\ell }\left(S_{T}^{\ell }\right),\;\ell =1,\ldots ,L}\left[\frac{1}{L}h\left(Y^{L}|H\right)\right]-\log\left(\pi e\right). \end{array}$$
However, CSCG inputs are not necessarily optimal since the inputs affect the CSIT.

Instead of trying to optimize the input, consider $X_{\ell }$ that are CSCG. We may write
(355)$$\begin{array}{c} I\left(X^{L}\to Y^{L}|H\right)=\sum_{\ell =1}^{L}E\left[\log\left(1+GP_{\ell }\left(S_{T}^{\ell }\right)\right)\right] \end{array}$$
and the Lagrangians to maximize (Equation 355) are
(356)$$\begin{array}{c} \sum_{\ell =1}^{L}E\left[\log\left(1+GP_{\ell }\left(S_{T}^{\ell }\right)\right)\right]+\lambda \left(LP-\sum_{\ell =1}^{L}E\left[P_{\ell }\left(S_{T}^{\ell }\right)\right]\right). \end{array}$$

Suppose the $S_{T\ell }$ are discrete random variables. Taking the derivative with respect to $P_{\ell }\left(s_{T}^{\ell }\right)$, we obtain
(357)$$\begin{array}{cc} \lambda \; & =\int_{0}^{\infty }p\left(g|s_{T}^{\ell }\right)\frac{g}{1+gP_{\ell }\left(s_{T}^{\ell }\right)}dg\\ \; & \;+\sum_{k=\ell +1}^{L}\sum_{s_{T}^{k}}\int_{0}^{\infty }p\left(g\right)\frac{dP_{S_{T}^{k}|G}\left(s_{T}^{k}|g\right)}{dP_{\ell }\left(s_{T}^{\ell }\right)}\frac{\log\left(1+gP_{k}\left(s_{T}^{k}\right)\right)}{P_{S_{T}^{\ell }}\left(s_{T}^{\ell }\right)}dg \end{array}$$
as long as $P_{\ell }\left(s_{T}^{\ell }\right)>0$. This expression is complicated because the choice of transmit powers $P_{\ell }\left(s_{T}^{\ell }\right)$ influences the statistics of the future CSIT $S_{T(\ell +1)},\ldots ,S_{TL}$. If (Equation 357) cannot be satisfied, choose $P_{\ell }\left(s_{T}^{\ell }\right)=0$. Finally, set λ so that $\sum_{\ell =1}^{L}E\left[P_{\ell }\left(S_{T}^{\ell }\right)\right]=LP$.

Instead of the above, consider the simpler CSIT model with $S_{T\ell }=f_{T\ell }\left(H\right)$ for all *ℓ*, cf. (Equation 345). The capacity (Equation 346) is now given by (Equation 355) with CSCG inputs and (Equation 357) simplifies because the derivatives with respect to $P_{\ell }\left(s_{T}^{\ell }\right)$ are zero, i.e., the double sum in (Equation 357) disappears and for all *ℓ* and $s_{T}^{\ell }$ we have
(358)$$\begin{array}{cc} \; & \lambda =\int_{0}^{\infty }p\left(g|s_{T}^{\ell }\right)\frac{g}{1+gP_{\ell }\left(s_{T}^{\ell }\right)}dg. \end{array}$$
We use (Equation 358) for (Equation 362)–(364) in Section 9.7 below.

### 9.5. Full CSIR, Partial CSIT

We next generalize Proposition 2 in Section 6.4 to the block-fading AWGN model (Equation 350) with the CSIR
(359)$$\begin{array}{c} S_{R\ell }=H\sqrt{P(S_{T}^{\ell })},\;\ell =1,\ldots ,L \end{array}$$
and where $S_{T\ell }=f_{T\ell }\left(S_{H}\right)$, i.e., we have discarded $X_{i}^{\ell -1}$ and $Y_{i}^{\ell -1}$ in (Equation 321). We then have the following capacity result that implies this CSIR is the best possible since one achieves a capacity upper bound similar to (Equation 161).

**Proposition** **3.**
*The capacity of the channel (Equation 350) with the CSIR (Equation 359) and $S_{T\ell }=f_{T\ell }\left(S_{H}\right)$ for ℓ=1,…,L is*

(360)
$$\begin{array}{cc} C(P)\; & =\max\frac{1}{L}\sum_{\ell =1}^{L}E\left[\log\left(1+GP_{\ell }\left(S_{T}^{\ell }\right)\right)\right] \end{array}$$

*where the maximization is over the power control policies $P_{\ell }\left(S_{T}^{\ell }\right)$ such that $\sum_{\ell =1}^{L}E\left[P_{\ell }\left(S_{T}^{\ell }\right)\right]\leq LP$. One may use (Equation 358) to compute the $P_{\ell }\left(S_{T}^{\ell }\right)$.*


**Proof.** For achievability, apply (Equation 337) with
$$\begin{array}{c} {\tilde{P}}_{\ell }\left({\underline{S}}_{R}\right)=GP_{\ell }\left(S_{T}^{\ell }\right)\;\mathrm{and}\;E\left[|Y_{\ell }{|}^{2}|{\underline{S}}_{R}\right]=1+{\tilde{P}}_{\ell }\left({\underline{S}}_{R}\right). \end{array}$$The converse follows by applying similar steps as in (Equation 162):
(361)$$\begin{array}{cc} I(A^{L};Y^{L}|S_{R}^{L}) & \leq I(A^{L};Y^{L},S_{T}^{L},H|S_{R}^{L})\\ \; & =\sum_{\ell =1}^{L}I\left(A^{L};Y_{\ell }\;|\;S_{R}^{L},S_{T}^{L},H,Y^{\ell -1}\right)\\ \; & \leq \sum_{\ell =1}^{L}\left[h\left(Y_{\ell }|S_{R}^{L},S_{T}^{L},H,Y^{\ell -1}\right)-h\left(Z_{\ell }\right)\right]\\ \; & \overset{\left(a\right)}{\leq }\sum_{\ell =1}^{L}E\left[\log\mathrm{Var}\left[Y_{\ell }|S_{R}^{L},S_{T}^{L},H,Y^{\ell -1}\right]\right]. \end{array}$$
Finally, insert $\mathrm{Var}\left[Y_{\ell }|S_{R}^{L},S_{T}^{L},H,Y^{\ell -1}\right]=1+GP_{\ell }\left(S_{T}^{\ell }\right)$. □

The RHS of (Equation 361) is at most the RHS of (Equation 352) and hence (Equation 361) gives a better bound. However, the bound (Equation 361) is valid only for particular CSIT, as in Remark 78.

### 9.6. On-Off Fading with Delayed CSIT

Consider on-off fading where the CSIT is delayed by *D* symbols, i.e., we have $S_{T\ell }=0$ for ℓ=1,…,D and $S_{T(D+1)}=H$. Define the transmit powers as $P_{\ell }\left(s_{T}^{\ell }\right)=E\left[|X\left(s_{T}^{\ell }\right){|}^{2}\right]$ for ℓ=1,…,L. The capacity is
$$\begin{array}{cc} C(P) & =\frac{D}{2L}\log\left(1+2P_{1}\right)+\frac{L-D}{2L}\log\left(1+2P_{D+1}\right) \end{array}$$
where we write $P_{D+1}=P_{D+1}\left(s_{T}^{D+1}\right)$. Optimizing the powers, we obtain
$$\begin{array}{cc} \; & \left\{\begin{array}{c} P_{1}=P-\frac{L-D}{4L}\\ P_{D+1}=2P+\frac{D}{2L} \end{array}\right\}\;\mathrm{if}\;P\geq \frac{L-D}{4L}\\ \; & \left\{\begin{array}{c} P_{1}=0\\ P_{D+1}=\frac{2LP}{L-D} \end{array}\right\}\;\mathrm{else}. \end{array}$$

For large *P*, we thus have $C\left(P\right)\approx \frac{1}{2}\log\left(P\right)$ for all 0≤D≤L. For small *P*, we have
$$\begin{array}{cc} C(P) & =\left\{\begin{array}{cc} \frac{L-D}{2L}\log\left(1+\frac{4LP}{L-D}\right), & \mathrm{if}\;0\leq D<L\\ \log(1+2P)/2, & \mathrm{if}\;D=L \end{array}\right.\\ \; & \approx \left\{\begin{array}{cc} \left(2P-\frac{4L}{L-D}P^{2}\right)\log\left(e\right), & \mathrm{if}\;0\leq D<L\\ \left(P-P^{2}\right)\log\left(e\right), & \mathrm{if}\;D=L. \end{array}\right. \end{array}$$
The CSIT thus gives a 3 dB power gain at low SNR since C(P)≈2Plog(e) for 0≤D<L and C(P)≈Plog(e) for D=L. Furthermore, using (Equation 37), the slope of the capacity versus $E_{b}/N_{0}$ in bits/s/Hz/(3 dB) is
$$\begin{array}{c} \begin{array}{cc} 1-D/L & \mathrm{if}\;0\leq D<L\\ 1 & \mathrm{if}\;D=L. \end{array} \end{array}$$

In other words, the delay reduces the low-SNR rate by a factor of 1−D/L for 0≤D<L.

### 9.7. Rayleigh Fading and One-Bit Feedback

Let $q_{u}\left(.\right)$ be the one-bit (B=1) quantizer in Section 2.9. We study Rayleigh fading for two scenarios with $S_{R}^{L}=H$, i.e., the receiver knows *H* after the *L* transmissions of each block.
For the CSIT (Equation 345), we study delayed feedback where $S_{T\ell }=0$ for ℓ=1,…,L−1 and $S_{TL}=q_{u}\left(G\right)$. The delay is thus D=L−1 in the sense of Section 9.6.For the CSIT (Equation 338), we study the case $S_{T1}=0$, $S_{T2}=q_{u}\left(\right|Y_{1}\left|\right)$, and $S_{T\ell }=0$ for ℓ=3,…,L. The delay is thus D=1 in the sense of Section 9.6.
*Delayed Quantized CSIR Feedback:* Consider $S_{T\ell }=0$ for ℓ=1,…,L−1 and $S_{TL}=q_{u}\left(G\right)$. CSCG inputs are optimal, and (Equation 347) has the same form as (Equation 360). The Lagrangians are given by (Equation 356), and we again obtain (Equation 358). For the case at hand, we have L+1 equations for λ, namely
(362)$$\begin{array}{cc} \lambda & =\int_{0}^{\infty }e^{-g}\frac{g}{1+gP_{\ell }}\;dg,\;\ell =1,\ldots ,L-1 \end{array}$$
(363)$$\begin{array}{cc} \lambda & =\int_{0}^{\Delta }\frac{e^{-g}}{1-e^{-\Delta }}\frac{g}{1+gP_{L}\left(\Delta /2\right)}\;dg \end{array}$$
(364)$$\begin{array}{cc} \lambda & =\int_{\Delta }^{\infty }\frac{e^{-g}}{e^{-\Delta }}\frac{g}{1+gP_{L}\left(3\Delta /2\right)}\;dg \end{array}$$
where we used (Equation 40) and (Equation 41) and abused notation by writing $P_{L}\left(s_{TL}\right)$ for $P_{L}\left(s_{T}^{L}\right)$. We thus have $P_{1}=\ldots =P_{L-1}$ and obtain three equations. We now search for λ such that
$$\begin{array}{c} \left(L-1\right)P_{1}+\sum_{s}P_{S_{TL}}\left(s\right)P_{L}\left(s\right)=LP \end{array}$$
and the capacity (Equation 353) is
(365)$$\begin{array}{c} C\left(P\right)=\frac{L-1}{L}e^{1/P_{1}}E_{1}\left(1/P_{1}\right)+\frac{1}{L}\sum_{s}\int_{I(s)}e^{-g}\log\left(1+gP_{L}\left(s\right)\right)dg \end{array}$$
where the sums are over s=Δ/2,3Δ/2 and
$$\begin{array}{c} I(\Delta /2)=[0,\Delta ),\;I(3\Delta /2)=[\Delta ,\infty ). \end{array}$$
We remark that, if $P_{1}=0$, then we set $e^{1/P_{1}}E_{1}\left(1/P_{1}\right)=0$ since $\lim_{x\to \infty }e^{x}E_{1}\left(x\right)=0$.

Figure 15 shows these capacities for L=1,2,3 and Δ=1. At low SNR (e.g., for L=3 below −2.97 dB) we have $P_{1}=0$ and $P_{L}\left(\Delta /2\right)=0$, i.e., the transmitter is silent unless $S_{TL}=3\Delta /2$ and it uses power at time ℓ=L only. Observe that, as in Section 9.6, a delay of *L* steps reduces the low-SNR slope, and therefore the low-SNR rates, by a factor of *L*. Delay can thus be costly at low SNR.

*Quantized Channel Output Feedback:* Consider $S_{T1}=0$, $S_{T2}=q_{u}\left(\right|Y_{1}\left|\right)$, and $S_{T\ell }=0$ for ℓ=3,…,L. As discussed in Remark 77, the capacity is given by the directed information expression (Equation 349). However, optimizing the input statistics seems difficult, i.e., CSCG inputs are not necessarily optimal. Instead, we compute achievable rates for a strategy where one symbol partially acts as a pilot.

Suppose the transmitter sends $X_{1}=\sqrt{P_{1}}e^{j\Phi }$ as the first symbol of each block, where Φ is uniformly distributed in [0,2π). The idea is that $|X_{1}|=\sqrt{P_{1}}$ is known at the receiver, and thus $X_{1}$ acts as a pilot to test the channel amplitude. Next, we choose a variation of flash signaling. Define the event $E=\left\{\right|Y_{1}\left|\geq \Delta \right\}=\left\{S_{T2}=3\Delta /2\right\}$. If this event does not occur, the transmitter sends $X_{\ell }=0$ for ℓ=2,…,L. Otherwise, the transmitter sends independent CSCG $X_{\ell }$ with variance $P_{2}/\mathrm{Pr}\left[E\right]$ for ℓ=2,…,L. Define $P_{\ell }\left(s_{T}^{\ell }\right)=E\left[|X\left(s_{T}^{\ell }\right){|}^{2}\right]$. We have $P_{\ell }=P_{2}$ for ℓ≥2 and the power constraint is $P_{1}+\left(L-1\right)P_{2}\leq LP$.

We use (Equation 347) to write
(366)$$\begin{array}{cc} \; & C\left(P\right)\geq \frac{1}{L}I\left(X_{1};Y_{1}|H\right)+\frac{L-1}{L}I\left(X^{2};Y_{2}|H,Y_{1}\right). \end{array}$$
The first mutual information in (Equation 366) is
$$\begin{array}{cc} I(X_{1};Y_{1}|H) & =h\left(Y_{1}|H\right)-\log\left(\pi e\right) \end{array}$$
and we compute (see ([52], Appendix A))
$$\begin{array}{c} p\left(y_{1}|h\right)=\frac{1}{\pi }e^{-(|y_{1}{|}^{2}+P_{1}|h{|}^{2})}I_{0}\left(2\;|y_{1}|\;|h|\;\sqrt{P_{1}}\right) \end{array}$$
where $I_{0}\left(.\right)$ is the modified Bessel function of the first kind of order zero. The Jacobian of the mapping from Cartesian coordinates $\left[ℜ\left(y_{1}\right),ℑ\left(y_{1}\right)\right]$ to polar coordinates $\left[|y_{1}|,\arg y_{1}\right]$ is $|y_{1}|$, so we have
$$\begin{array}{c} h\left(Y_{1}|H=h\right)=\int_{0}^{\infty }-\;p\left(y_{1}|h\right)\log\left(p\left(y_{1}|h\right)\right)\;2\pi |y_{1}\left|\;d\right|y_{1}|. \end{array}$$
We further compute
(367)$$\begin{array}{c} I\left(X^{2};Y_{2}|H,Y_{1}\right)=\int_{0}^{\infty }e^{-g}\mathrm{Pr}\left[E|G=g\right]\log\left(1+\frac{gP_{2}}{\mathrm{Pr}\left[E\right]}\right)dg. \end{array}$$
The conditional probability of a high-energy $Y_{1}$ is
$$\begin{array}{c} \mathrm{Pr}\left[E|G=g\right]=Q_{1}\left(\sqrt{2gP_{1}},\sqrt{2}\Delta \right) \end{array}$$
where $Q_{1}\left(.\right)$ is the Marcum Q-function of order 1; see (Equation 370) in Section A.1. For Rayleigh fading, we compute
$$\begin{array}{c} \mathrm{Pr}\left[E\right]=\mathrm{Pr}\left[{\left|H\sqrt{P_{1}}e^{j\Phi }+Z_{1}\right|}^{2}\geq \Delta^{2}\right]=e^{-\Delta^{2}/\left(P_{1}+1\right)}. \end{array}$$

The resulting rates are shown in Figure 16 for the block lengths L=10,20,100. Observe that each curve turns back on itself, which reflects the non-concavity of the directed information rates in *P*; see ([74], Section III). All rates below the curves are achievable by “time-wasting”, i.e., by transmitting for some fraction of the time only. This suggests that flash signaling [73] will improve the rates since one sends information by choosing whether to transmit energy.

## 10. Conclusions

This paper reviewed and derived achievable rates for channels with CSIR, CSIT, block fading, and in-block feedback. GMI expressions were developed for adaptive codewords and two classes of auxiliary channel models with AWGN and CSCG inputs: reverse and forward channel models. The forward model inputs were chosen as linear functions of the adaptive codeword’s symbols. We showed that, for scalar channels, an input distribution that maximizes the GMI generates a conventional codebook, where the codeword symbols are multiplied by a complex number that depends on the CSIT. The GMI increases by partitioning the channel output alphabet and modifying the auxiliary model parameters for each partition subset. The partitioning helps to determine the capacity scaling at high and low SNR. Power control policies were developed for full CSIT, including TMMSE policies. The theory was applied to channels with on-off fading and Rayleigh fading. The capacities with in-block feedback simplify to directed information expressions if the CSIT is a function of the CSIR and past channel inputs and outputs.

There are many possible applications and extensions of this work. For example, adaptive coding and modulation are important for all practical communication systems, including wireless, copper, and fiber-optic networks. Shannon’s adaptive codewords can improve current systems since the CSIT is usually a noisy version of the CSIR; see Remark 25. Moreover, the information theory for in-block feedback [22] applies to beamforming [106] and intelligent reflecting surfaces [107,108]. One may also apply GMI to multi-user channels with in-block feedback, such as multi-access and broadcast channels. Finally, it is important to develop improved capacity upper bounds. The standard approach here is the duality framework described in [97,109]; see also ([110], page 128).

## Figures and Tables

**Figure 1 entropy-25-00728-f001:**
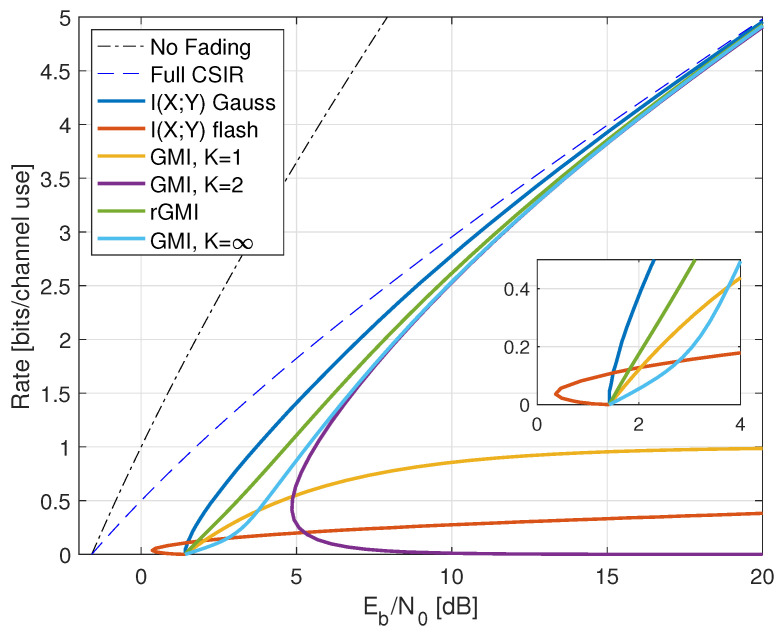
Rates for on-off fading with $S_{R}=0$. The curve “Full CSIR” refers to $S_{R}=H$ and is a capacity upper bound. Flash signaling uses p=0.05; the GMI for the K=2 partition uses the threshold $t_{R}=P^{0.4}+3$.

**Figure 2 entropy-25-00728-f002:**
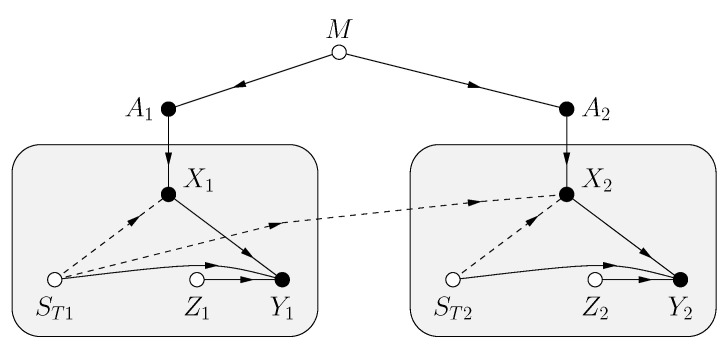
FDG for n=2 uses of a channel with CSIT. Open nodes represent statistically independent random variables, and filled nodes represent random variables that are functions of their parent variables. Dashed lines represent the CSIT influence on $X^{n}$.

**Figure 3 entropy-25-00728-f003:**
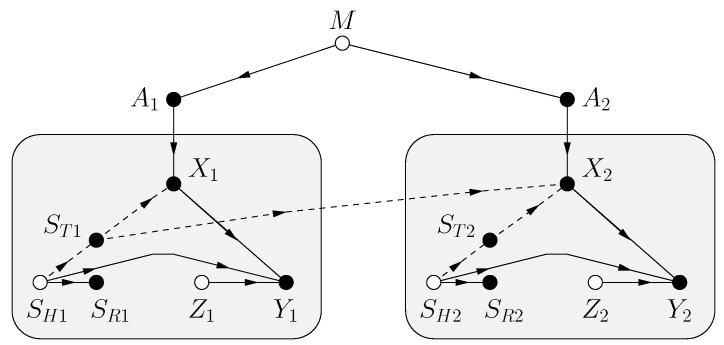
FDG for n=2 channel uses with different CSIT and CSIR. The hidden channel state $S_{Hi}$ permits dependent $S_{Ri}$ and $S_{Ti}$.

**Figure 4 entropy-25-00728-f004:**
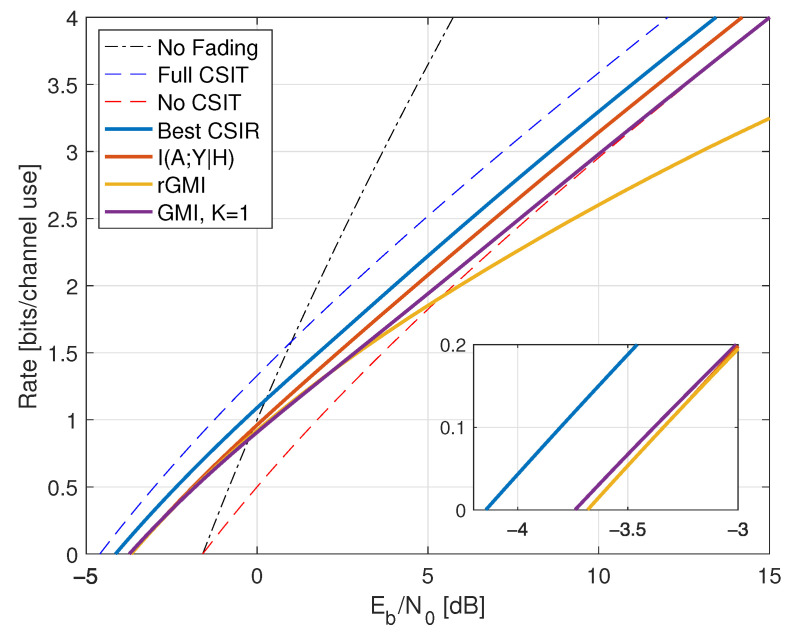
Rates for on-off fading with full CSIR and partial CSIT with noise parameter ϵ=0.1. The curve “Best CSIR” shows the capacity with $S_{R}=H\sqrt{P(S_{T})}$. The curves for I(A;Y|H), the reverse model GMI (rGMI), and the forward model GMI (GMI, *K* = 1) are for $S_{R}=H$ with CSCG inputs $X(s_{T})$. The I(A;Y|H) and rGMI curves are indistinguishable in the inset.

**Figure 5 entropy-25-00728-f005:**
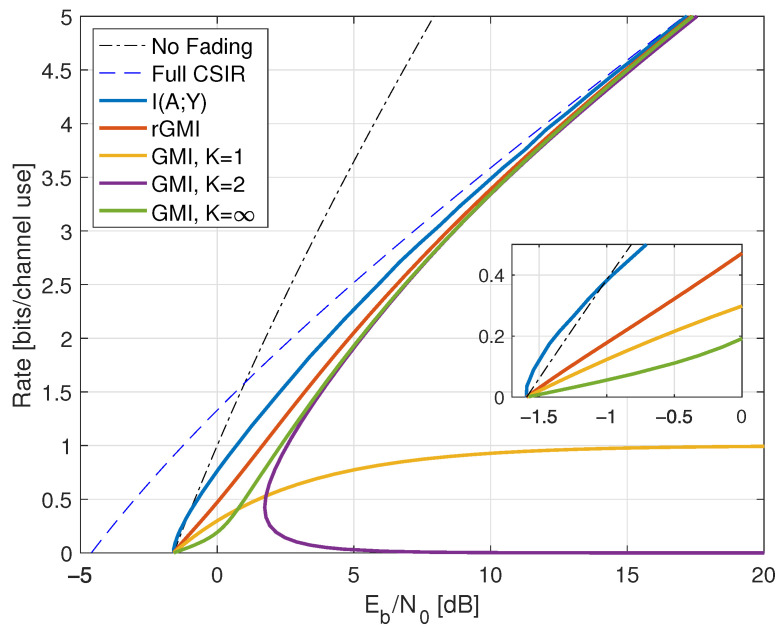
Rates for on-off fading with $S_{T}=H$ and $S_{R}=0$. The GMI for the K=2 partition uses the threshold $t_{R}=\sqrt{P}+3$.

**Figure 6 entropy-25-00728-f006:**
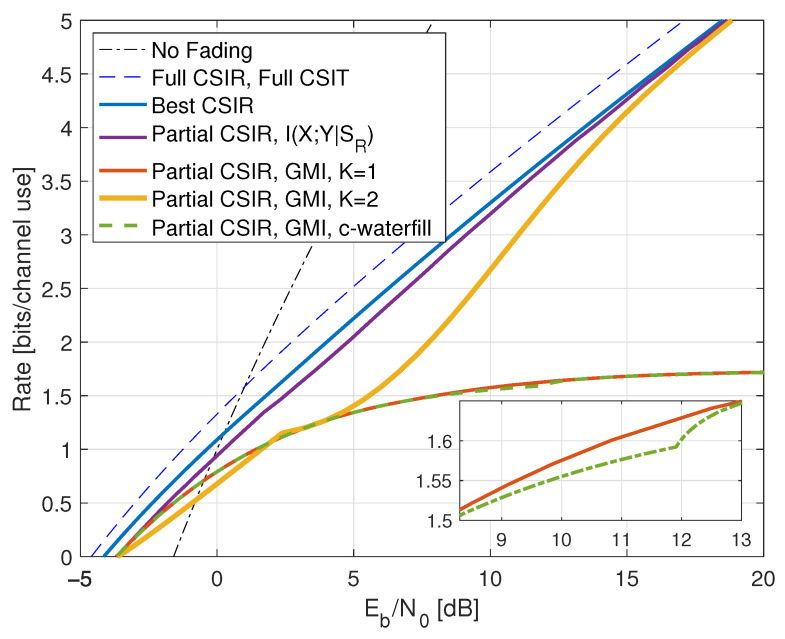
Rates for on-off fading with partial CSIR and CSIT@R. The curve “Best CSIR” shows the capacity with $S_{R}=H\sqrt{P(S_{T})}$. The mutual information $I(X;Y|S_{R})$ and the GMI are for $\mathrm{Pr}\left[S_{R}\neq H\right]=0.1$ and with CSCG inputs $X(s_{T})$. The GMI for the K=2 partition uses $t_{R}=P^{0.4}$. The curve labeled ‘c-waterfill’ shows the conventional waterfilling rates.

**Figure 7 entropy-25-00728-f007:**
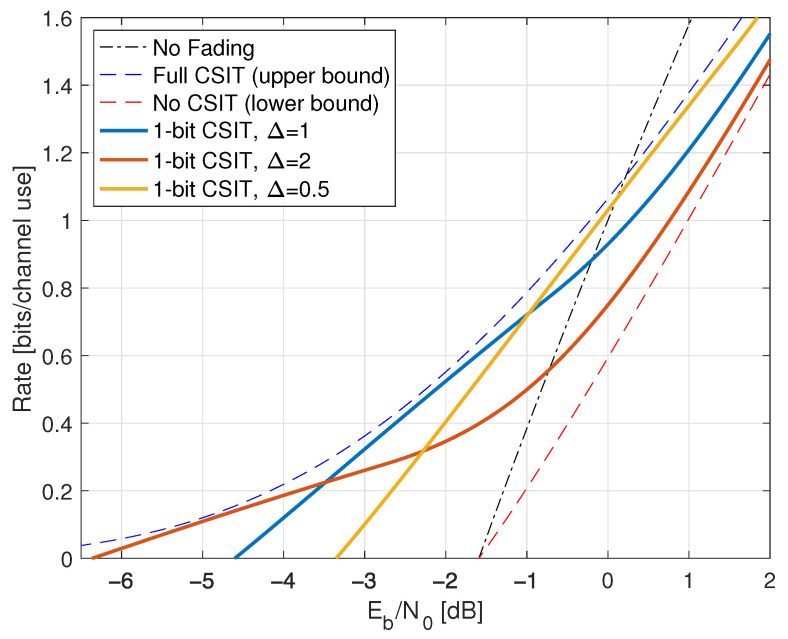
Capacities for Rayleigh fading with full CSIR, a one-bit quantizer with threshold Δ, and CSIT@R.

**Figure 8 entropy-25-00728-f008:**
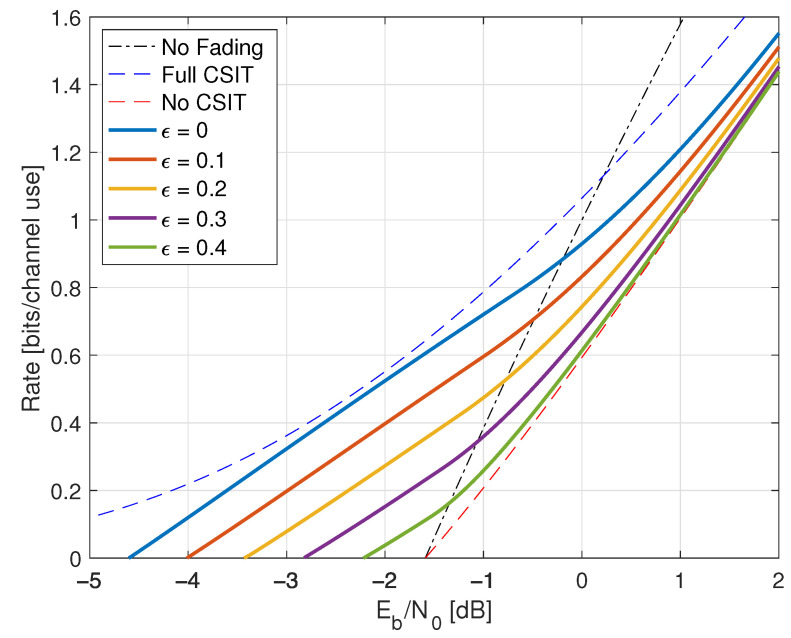
Capacities for Rayleigh fading, $S_{R}=\sqrt{P(S_{T})}H$, and a one-bit quantizer with threshold Δ=1, and various CSIT error probabilities ϵ.

**Figure 9 entropy-25-00728-f009:**
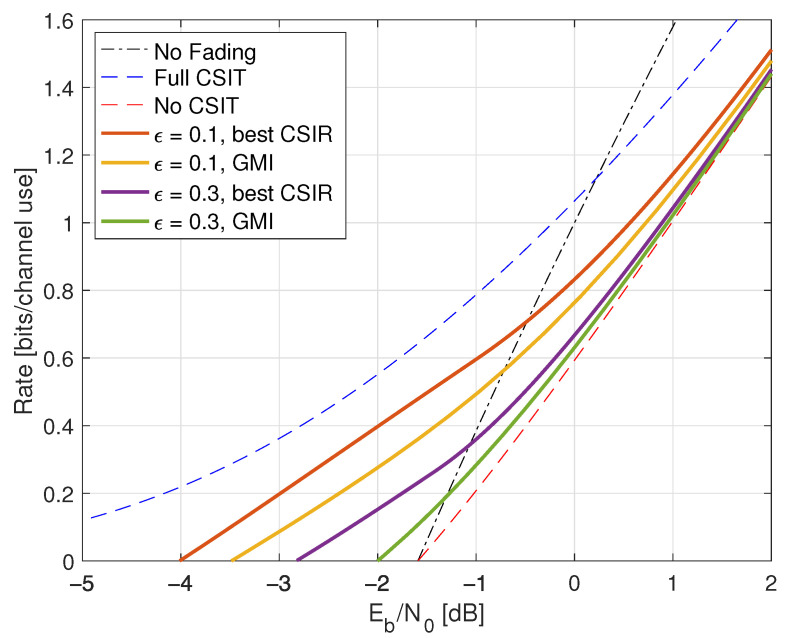
Rates for Rayleigh fading, $S_{R}=H$ and $S_{R}=H\sqrt{P(S_{T})}$, a one-bit quantizer with threshold Δ=1, and various ϵ. The curves labeled “best CSIR” show the capacities with $S_{R}=H\sqrt{P(S_{T})}$. The curves labeled “GMI” show the rates (Equation 285) for the optimal powers P(0) and P(1).

**Figure 10 entropy-25-00728-f010:**
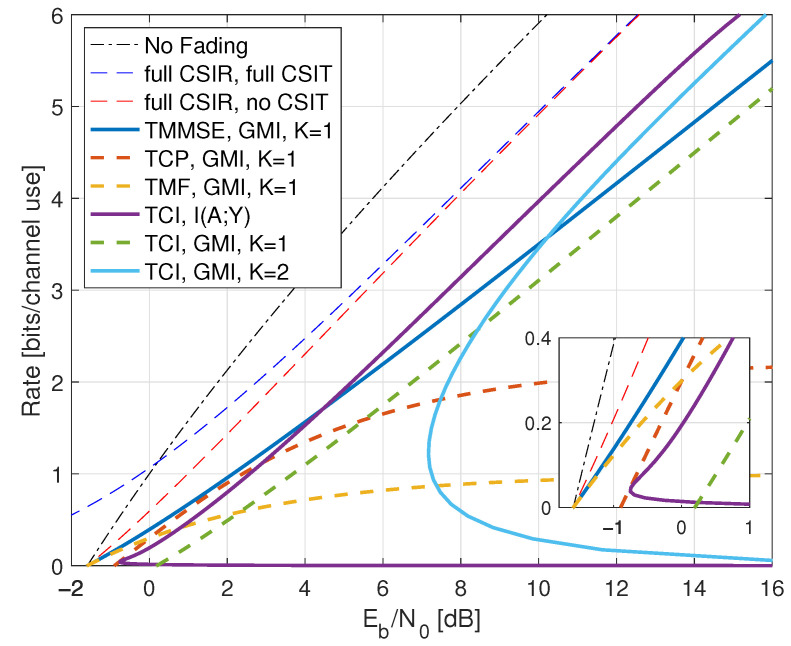
Rates for Rayleigh fading with $S_{T}=H$ and $S_{R}=0$. The threshold *t* was optimized for the K=1 curves, while $t=P^{-0.4}$ for the I(A;Y) and K=2 curves. The K=2 GMI uses $t_{R}=P^{0.4}$.

**Figure 12 entropy-25-00728-f012:**
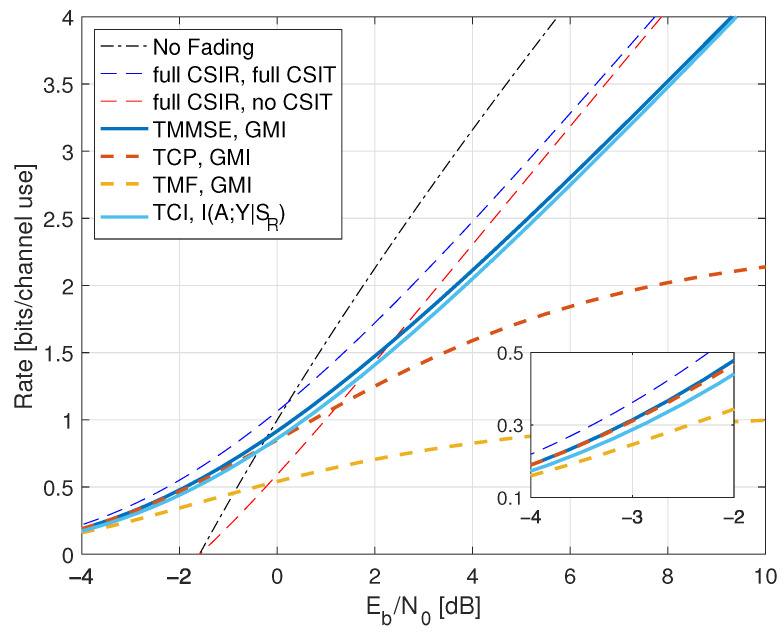
Rates for Rayleigh fading with full CSIT and $S_{R}=1\left(G\geq t\right)$.

**Figure 13 entropy-25-00728-f013:**
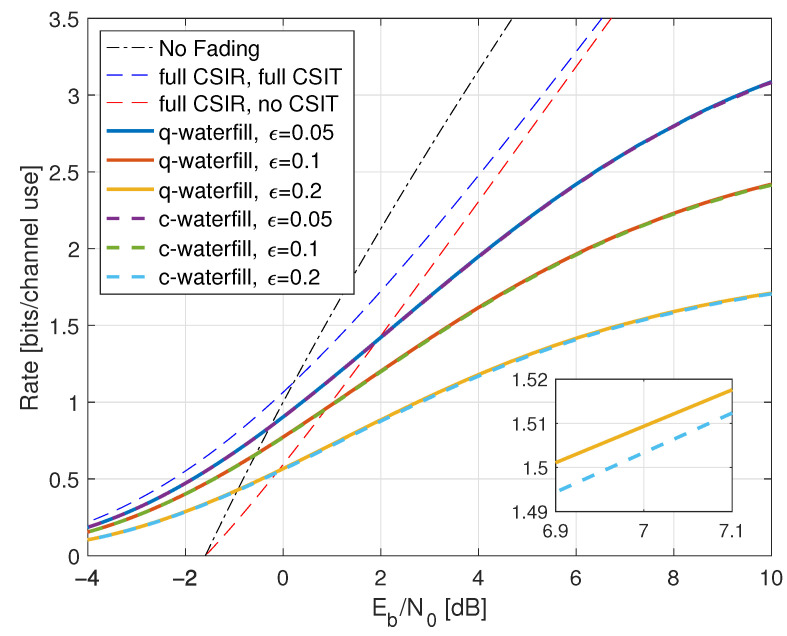
Rates for Rayleigh fading with partial CSIR and CSIT@R. The curves labeled ‘q-waterfill’ and ‘c-waterfill’ are the quadratic and conventional waterfilling rates, respectively.

**Figure 14 entropy-25-00728-f014:**
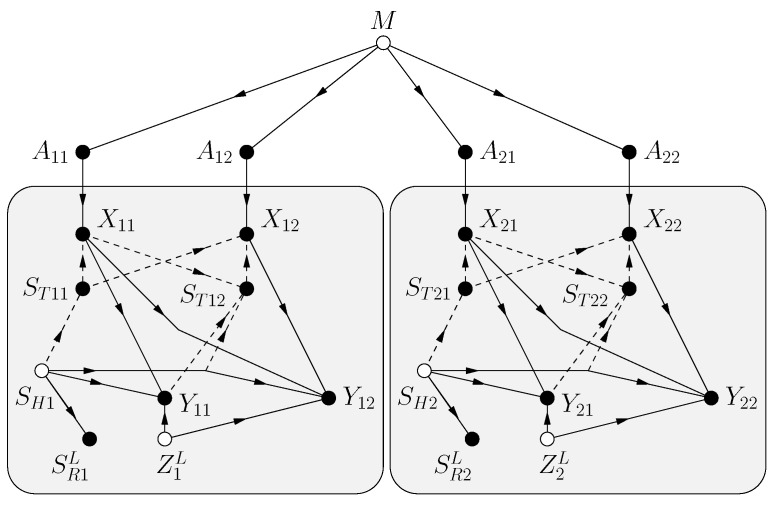
FDG for a block fading model with n=2 blocks of length L=2 and in-block feedback. Across-block dependence via past $S_{Ti\ell }$ is not shown.

**Figure 15 entropy-25-00728-f015:**
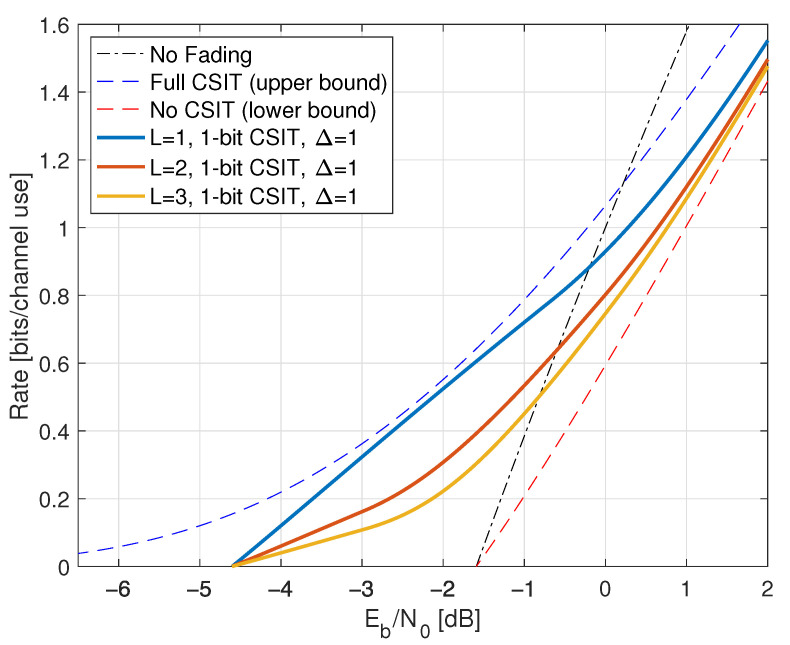
Capacities for Rayleigh block fading with L=1,2,3 and a CSIT delay of D=L−1. The CSIT at symbol *L* is $S_{TL}=q_{u}\left(G\right)$.

**Figure 16 entropy-25-00728-f016:**
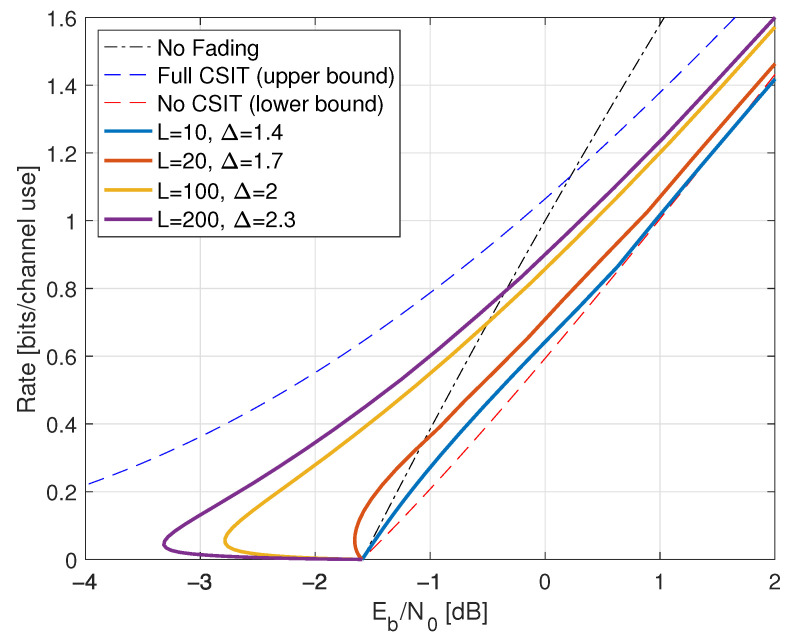
Rates for Rayleigh block fading with block lengths L=10,20,100. The CSIT at symbol 2 is $S_{T2}=q_{u}\left(\right|Y_{1}\left|\right)$.

**Table 1 entropy-25-00728-t001:** Models Studied in Section 6 (General Fading), Section 7 (On–Off Fading) and Section 8 (Rayleigh Fading).

		CSIR	
		Full	Partial/No
CSIT	Full	Section 6.3	Section 6.5
	@R	Section 6.3	Section 6.6
	Partial/No	Section 6.4	Section 6.2

**Table 2 entropy-25-00728-t002:** Power Control Policies and Minimal SNRs.

		CSIR	
		None: $S_{R}=0$	$S_{R}=1\left(G\geq t\right)$
Policy	TCP	Equation (Equation 221)	Equation (Equation 226)
	TMF	Equation (Equation 222)	Equation (Equation 227)
	TCI	Equation (Equation 223)	Equation (Equation 228)
	GMI-Optimal	See Theorem 2	
	TMMSE	See Remark 64	

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
