# Peer review of "Information Rates for Channels with Fading, Side Information and Adaptive Codewords"

_entropy, 2023, doi:10.3390/e25050728_

Round 1

Reviewer 1 Report

This paper presented the calculation of information rate for different channels with fading and in-block feedback. Authors used GMI to compute rates based on channel models. I would like to thank the author for attractive results. In my opinion, the applied approaches and calculations are correct. I only have some minor comments as follows.

After Lemma 4, author mentioned that optimizing the GMI is more difficult for N > 1. Please explain this point more clearly.

In fact, readers are very interested in the practical application of this work as well as its extensions. Could you please discuss these points further?

Reviewer 2 Report

The paper proposes lower bounds on the capacity of fading channels with various types of channel-state information at the transmitter and receiver that are based on the concept of generalized mutual information. These bounds are easy to evaluate and, in some cases, allow for an optimization of the transmit power (as a function of the channel-state information at the transmitter).

Fading channels are the go-to models for wireless communication problems, but their capacity is usually not known in closed form, except for some special cases, hence it must be assessed by means of upper and lower bounds. The concept of generalized mutual information is a powerful tool to obtain capacity lower bounds, since the choice of the auxiliary model offers a great degree of freedom that can be exploited to obtain tight bounds. This paper presents a number of variations of auxiliary models and demonstrates their usefulness by evaluating the corresponding generalized mutual informations for various settings of fading channels. The paper is a good introduction to the concept of generalized mutual information, and it is very well written. I have therefore no doubt that it will be a valuable contribution for this special after a minor revision.

On the negative side, with 92 pages, the paper is rather long, and it can sometimes a bit of a tedious read. The reason is that the same set of auxiliary models (reverse auxiliary model, forward auxiliary model, K-partitions) is applied to variations of fading channels, which can be somewhat monotonic. Furthermore, it is sometimes unclear how accurate the obtained lower bounds are, since the only upper bounds against which they are compared are trivial bounds, such as the capacity of the non-fading channel. It would be interesting to derive corresponding capacity upper bounds by means of the dual expression of channel capacity, as proposed in

A. Lapidoth, S. M. Moser, "Capacity bounds via duality with applications to multiple-antenna systems on flat-fading channels," IEEE Transactions on Information Theory, October 2003.

In any case, this is "complaining at a high comfort level", and the positive aspects of the paper predominate.

In the following, I provide a number of more detailed comments:

1. After (15), it is written that |f_y|^2 must be chosen as \pi e^{h_y} so that q(x|y) is a density. However, if I am not mistaken, then q(x|y) is need not be a density anyway. Perhaps a small comment would be helpful.

2. The author introduces the flash density p(x) in (36). However, [68] provides an exact definition of what flash signaling is; see [68, Definition 2]. It is unclear to me whether (36) is consistent with this definition. In any case, I am pretty sure that this is only the case if p -> 0 as the SNR tends to infinity. A brief discussion would be appreciated.

3. In (43), is q_{X|Y} always a density? I suppose it needs to be, since otherwise the relative entropy D(p_{X,Y} || p_Y q_{X|Y}) is not meaningful.

4. After (81), it is written that "at large SNR the receiver can estimate H accurately and one approaches the capacity with full CSIR." However, (81) only demonstrates that the dominating term of I_1(X;Y) is 1/2*log(1 + 2P). That is, any gap between I_1(X;Y) and 1/2*log(1 + 2P) must grow more slowly with P than 1/2*log(1 + 2P). However, this does not imply that this gap vanishes as P tends to infinity.

5. Does (83) not hold with equality?

6. Perhaps one could mention that the upper bound (92) corresponds to the case where transmitter and receiver have full channel-state information.

7. After (96), it is written that "an optimal p(a) is an extreme of P(A)." However, I believe this is only true if p(y) remains fixed. By changing p(a), do we not also change p(y)? Or would we need to intersect P(A) with the set of all distributions p(a) for which p(y) is fixed?

8. I did not understand the sentence before (134): "[...] where there is a hidden state S_H, the CSIR S_R such that [S_T,S_R] is a function of S_H [...]" Is there a typo?

9. Lemma 6: Is it not possible to solve the maximization on the right-hand side of (158) using Jensen's inequality? That is, is P(S_T) = P not optimal?

10. Remark 56 states that "the SNR(h) in (192) saturates unless we choose P(s_T) = P for all h". Is this really true? How about choosing P(s_T) = P + f(s_T) where f(S_T) grows sublinearly in P? In other words, is it not sufficient to choose P(s_T) such that the variance of \sqrt{P(S_T)} has a smaller order than P(S_T)?

11. After (208), the minimum energy per bit is compared against -1.59 dB. However, I believe this is the minimum energy per bit of fading channels when the transmitter does not have full channel-state information. At least when transmitter and receiver have full channel-state information, then the minimum energy per bit is -infinity, is it not? So comparing the minimum energy per bit of various policies against -1.59 dB may not be that meaningful.

12. In Remark 62, there is a typo in the last line: It should be "I(A;Y)" and not "I(A : Y)".

13. It is unclear whether (257) really implies that "one approaches the capacity with full CSI." See Comment 4.

14. Similarly, it is unclear whether (264) really implies that "the K=2 GMI can approach the capacity of Proposition 2". See Comment 4.

15. Last paragraph before Section 8 on p. 50: The author mentions a discontinuity at approximately 2.5 dB. Is this really a discontinuity (in the mathematical sense), or is the curve simply not smooth?

Reviewer 3 Report

This is a very solid work presenting important insights into channels with state information. It treats the topic extensively and thoroughly, focusing on analytically tractable solutions. The current version is already well written and the reviewer does not see any major flaw or concern.